# PD-1 antibody-bound progenitor-exhausted CD8+ T cells in lymph nodes boost PD-1-blockade anti-tumor immunity in gastrointestinal cancer

Yohei Nose [1,2,8], Yoshiaki Yasumizu [3,4,8], Takuro Saito [1,2] ✉, Yamami Nakamura[3], Koichi Jinushi[1,2], Kaoru Fujikawa[1,2], Kota Momose[1], Kotaro Yamashita[1], Koji Tanaka[1], Kazuyoshi Yamamoto [1], Tomoki Makino [1], Tsuyoshi Takahashi [1], Azumi Ueyama[2,5], Yukinori Kurokawa [1], Eiichi Sato [6], Naganari Ohkura[3,7], Shimon Sakaguchi [3], Hisashi Wada [1,2], Hidetoshi Eguchi[1] & Yuichiro Doki[1]

While progenitor-exhausted T cells (Tpex) expressing TCF1 and PD-1 are crucial for the therapeutic effect of immune checkpoint inhibitors (ICIs) with therapeutic anti-PD-1 antibodies (aPD-1), the dynamics of ICI-bound Tpex are not fully understood. In this study, we investigate ICI-bound T cells in detail using combined sequencing analysis at the single-cell level. By analyzing samples from gastrointestinal cancer patients with or without ICI treatment, we find that Tpex are enriched in proximal lymph nodes (LNs) and proliferate at a high rate after ICI treatment. Importantly, aPD-1 high-bound Tpex in LNs share T-cell receptor clonotypes with intratumoral exhausted CD8+ T cells (Tex), suggesting their migration to tumor sites after ICI treatment. This study thus provides new insights into how ICIs enhance anti-tumor immunity by acting on Tpex in LNs, deepening our understanding of the cellular mechanisms underlying ICI therapy.

The therapeutic applications of immune checkpoint inhibitors (ICIs) have expanded in several cancer types, including gastrointestinal cancers. However, their response rates are limited[1–9]. Given that preoperative ICI treatment is not currently recommended for gastrointestinal cancers, mechanistic analysis of the anti-tumor immune response with human samples after ICI treatment is of great importance, since elucidation of these mechanisms may improve therapeutic efficacy[10].

Cancer antigen-specific T cells were detected among PD-1+ T cells in tumors and peripheral blood of cancer patients[11,12], and response to ICI treatment is associated with the abundance of PD-1-expressing exhausted T cells (Tex) in tumors[13–16]. Accordingly, anti-PD-1 antibodies are thought to exert their therapeutic effects by binding to PD-1 molecules on immune cells. Recently, progenitor-exhausted T cells (Tpex) expressing TCF7 along with PD-1 have been identified in various

[1]Department of Gastroenterological Surgery, Graduate School of Medicine, The University of Osaka, Suita, Japan. [2]Department of Clinical Research in Tumor Immunology, Graduate School of Medicine, The University of Osaka, Suita, Japan. [3]Experimental Immunology, WPI Immunology Frontier Research Center, The University of Osaka, Suita, Japan. [4]Integrated Frontier Research for Medical Science Division, Institute for Open and Transdisciplinary Research Initiatives (OTRI), The University of Osaka, Suita, Japan. [5]Pharmaceutical Research Division, Shionogi & Co., Ltd., Toyonaka, Japan. [6]Department of Pathology, Institute of Medical Science (Medical Research Center), Tokyo Medical University, Tokyo, Japan. [7]Department of Basic Research in Tumor Immunology, Graduate School of Medicine, The University of Osaka, Osaka, Japan. [8]These authors contributed equally: Yohei Nose, Yoshiaki Yasumizu. ✉ e-mail: tsaito@gesurg.med.osaka-u.ac.jp

organs, which have the potential to be bound by anti-PD-1 therapeutic antibodies (aPD-1). Indeed, these populations also correlated with a response to ICI treatment[17–21]. However, it remains unclear which population of Tex or Tpex is activated by ICIs and whether the induction of tumor-reactive CD8[+] T cell responses occurs primarily in the tumor microenvironment (TME) or in the tumor-draining lymph nodes (TDLNs). Therefore, this study addressed the questions of which cell populations are tumor-reactive T cells activated by the binding of aPD-1, and where these cells are located.

A problem is that aPD-1 bind to surface PD-1 molecules on immune cells, making it impossible to detect these cells with conventional PD-1 antibodies and necessitating a new method to detect these cells ex vivo. We therefore developed a method to detect ICI-bound T cells using an anti-IgG4 antibody and reported the importance of increased CD103 expression on blood ICI-bound CD8[+] T cells with a favorable prognosis in gastric cancer patients[22]. Similar methods were used in some reports with the prognostic significance of increased proliferative activity of blood ICI-bound T cells after ICI treatment[23,24]. However, there are no reports demonstrating the dynamics of ICI-bound T cells in organs other than the blood.

Paired single-cell RNA and T-cell receptor sequencing (scRNA/TCR-seq) has enabled deep profiling of T cells in the context of their clonal lineage, phenotypic heterogeneity, and tissue distribution[25]. We combined cellular indexing of transcriptomes and epitopes sequencing (CITE-seq) with scRNA/TCR-seq (scRNA/TCR/CITE-seq) to analyze the dynamics of ICI-bound T cells in more detail, by detecting IgG4 antibody at the single-cell level. Indeed, this analysis of ICI-bound T cells allowed us to reveal region-dependent transcriptional programs of T-cell clonotypes. In addition, we systematically profiled the regional distribution and region-dependent cell states of ICI high- and ICI low-bound T cells.

Here, we demonstrate that ICIs preferentially engage CD8-Tpex cells in LNs, driving their proliferation and clonal expansion, followed by their migration to tumors and differentiation into CD8-Tex cells. This finding provides a clonally resolved framework for understanding how aPD-1 orchestrates tumor-specific T-cell responses across tissues.

## Results

### Multi-regional scRNA-seq of T cells in gastrointestinal cancer samples after ICI treatment

To profile the characteristics of infiltrating T cells in various organs after anti-PD-1 therapy, we first analyzed a discovery cohort of 2 patients with metastatic esophagogastric junction (EGJ) cancer who received aPD-1 combined with chemotherapy and underwent surgical resection within a month after completion of the therapy (Fig. 1a, Supplementary Fig. 1). We obtained 4 samples of tumor, metastatic LN, metastasis-free LN, and PBMC from patient 1, and 3 samples of liver metastases, metastasis-free LN, and PBMC from patient 2 (Supplementary Table 1a). CD3[+] T cells in these samples were sorted for scRNA-seq analysis. We then performed scRNA-seq to characterize their molecular profiles. After quality control, 19,388 CD3[+] T cells were retained.

Clustering analysis of the scRNA-seq data identified 23 clusters, which we segregated into CD8[+] (12 clusters), conventional CD4[+] (8 clusters), regulatory CD4[+] (Treg cells) (2 clusters), and doublet (1 cluster) cells (Fig. 1b). These clusters were annotated with reference to differentially expressed genes and marker genes (Fig. 1c, Supplementary Fig. 2a, b)[21]. In these clusters, exhausted T cells (Tex; *TCF7[−]*, *PDCD1[+]*, *TOX[+]*), progenitor-exhausted T cells (Tpex; *TCF7[+]*, *PDCD1[+]*), follicular helper T cells (Tfh; *TCF7[+]*, *PD-1[+]*, *CXCR5[+]*, *BCL6[+]*), effector T cells (*GZMK[+]*, *GNLY[+]*, *FGFBP2[+]*), regulatory T cells (Treg; *FOXP3[+]*, *CTLA4[+]*, *CCR8[+]*), naive T cells (*CCR7[+]*, *TCF7[+]*), central memory T cells (Tcm; *CCR7[+]*), and mucosal-associated invariant T cells (MAIT; *SLC4A10[+]*) were identified (Fig. 1b, c, Supplementary Fig. 2a). Tissue specificity was also detected in the clusters (Fig. 1d, e, Supplementary

Fig. 3a–c). Among CD8[+] T cell clusters, tumor and liver metastases were enriched for CD8-Tex (C04, C06.0, C17.0, C17.1) and CD8-Tpex (C12) cells, while LNs were enriched for CD8-Tpex (C05) and CD8-naive (C11) cells. Blood was enriched for CD8-naive (C10) and CD8-effector (C08) cells. Likewise, among CD4[+] T cell clusters, tumor and liver metastases were enriched for CD4-Tex (C06.1, C17.2) and CD4-Treg effector (C07) cells, while LNs were enriched for CD4-Tfh (C09), CD4-naive (C00, C15), CD4-central memory (C03), and CD4-Treg (C13) cells. Blood was enriched for CD4-naive (C02) and CD4-central memory (C01) cells. The transcriptomic similarity of clusters was observed mainly according to tissue, whereas CD8-Tpex (C05) in LNs was similar to CD8-Tex (C04, C06.0) in tumor and liver metastases, and CD8-effector (C08) in PBMCs, despite these clusters exhibiting differential expression of multiple genes (Fig. 1e, Supplementary Fig. 3d). CD4-Tfh (C09) in LNs was also similar to CD4-Tex (C06.1) in tumor and liver metastases (Fig. 1e). The canonical gating strategy of *TCF7[+]PDCD1[+]CD8[+]* for CD8-Tpex overlapped with C05[20,26–29], and *TCF7[+]PDCD1[+]CD4[+]* for CD4-Tfh overlapped with C09, which is consistent with previous studies[29,30] (Fig. 1f). Importantly, C05 cells expressed T follicular marker *CXCR5*. As CXCR5[+]CD8[+] T cells are reported to act on Tfh and exhibit anti-tumor effects, C05 cells may interact with C09 cells[31]. In summary, the regional distribution of T-cell subtypes was similar based on each tissue, and Tex and Tpex were coordinately enriched in tumor and LNs.

### The C05 cluster (CD8-Tpex) was correlated with the therapeutic effect of ICI treatment

It has been reported that CD8-Tpex cells proliferate following ICI treatment[17–20], that CD8-Tpex cells in tumors correlate with the efficacy of ICI treatment[21], and that LNs act as reservoirs for these cells[32–34]. Therefore, we hypothesized that CD8-Tpex (C05) cells play a critical role in the efficacy of ICI treatment. To test this hypothesis, we analyzed a previously reported scRNA-seq dataset of tumor-infiltrating lymphocytes from 15 surgically resected tumors after ICI treatment[35] to identify cell populations that affect the response to ICI treatment. The single-cell data on mutation-associated neoantigens were projected to our dataset using Symphony, in which gene expression profiles showed a good projection of this gene set and cluster-level similarities between the datasets were observed in tissue-specific comparative analyses (Supplementary Figs. 4a, b, 5). Comparison of the proportion of cells in each cluster between patients with the major pathologic response (MPR) and the non-major pathologic response (non-MPR) showed a significantly higher proportion of CD8-Tpex (C05) and CD8-effector (C08) in the MPR group than in the non-MPR group, indicating that abundant intratumoral CD8-Tpex and CD8-effector were associated with a favorable therapeutic effect of ICI (Supplementary Fig. 4c). Following these results, we focused further on the CD8-Tpex (C05) cells that were enriched in LNs in our scRNA-seq data.

### Transcriptional signature of CD8-Tpex and CD4-Tfh

To examine the tissue-specific differences among CD8-Tpex cells, we investigated the characteristic genes of CD8-Tpex (C05) cells in LNs by comparing the gene expression of CD8-Tpex (C05) cells with that of CD8-Tex (C04, C06.0, C17.0, C17.1) and CD8-Tpex (C12) cells in tumor and liver metastases, and with that of CD8-naive (C11) cells in LNs (Fig. 1g, h). CD8-Tpex cells in LNs expressed naive/progenitor-associated genes (*TCF7, KLF2, IL7R, CCR7, GPR183, SELL*) at high levels compared with CD8-Tex/Tpex cells in tumor and metastasis. CD8-Tpex cells in LNs also had high expression of genes of effector transcripts (*GZMA, GZMK*), co-stimulatory molecules (*TIGIT, KLRG1, LAG3, SLAMF7, KLRD1*), IFN response (*IFNG, TNF*), and chemokine ligands and receptors (*CCL4, CCL5, CXCR3, CXCR4*) compared with CD8-naive cells. These data suggest that CD8-Tpex cells have already acquired migratory and effector functions in LNs while maintaining progenitor cell characteristics. We also investigated the characteristically expressed genes of

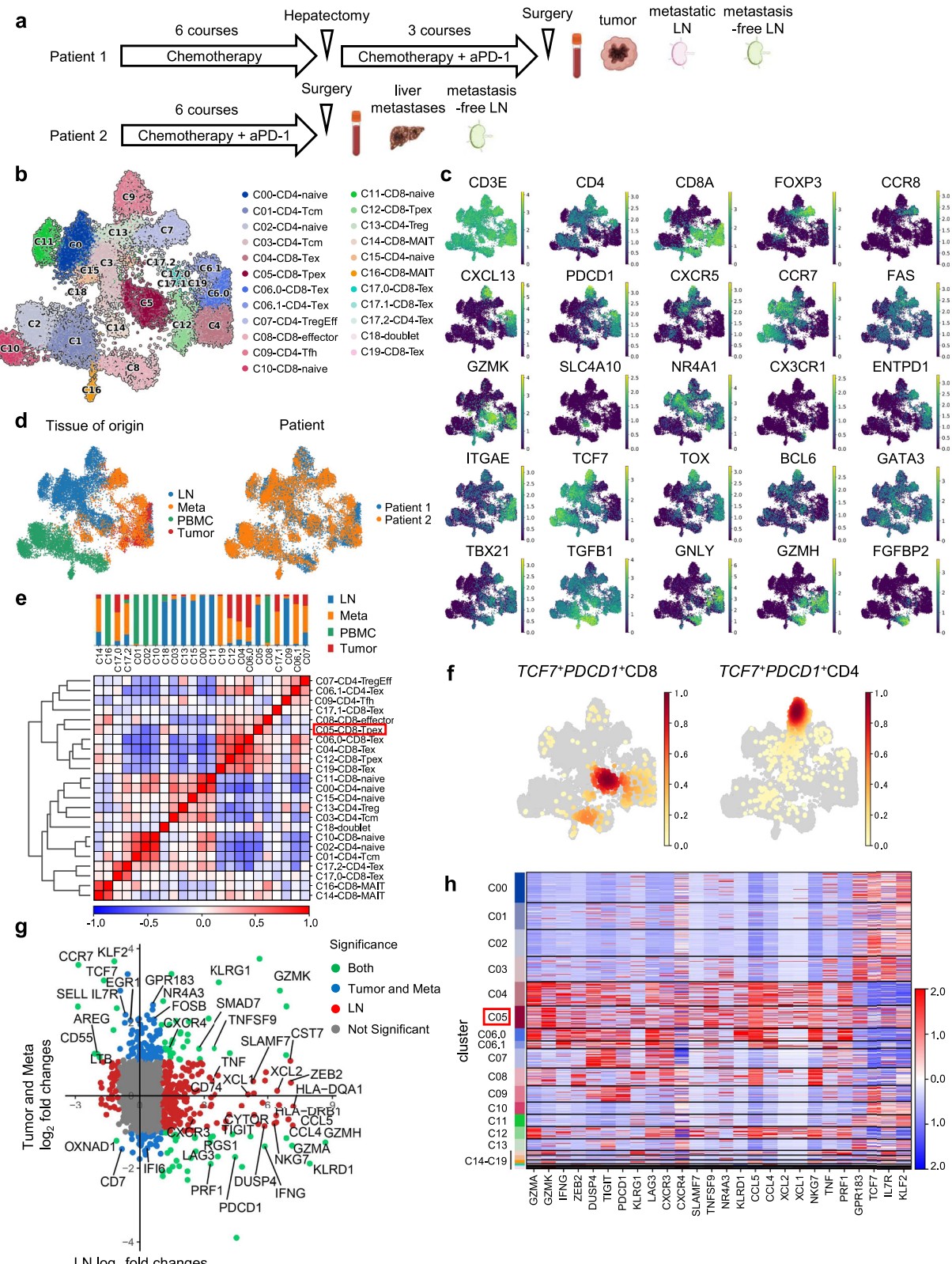

CD4-Tfh (C09) cells in LNs by comparing them with the gene expression of CD4-Tex (C06.1, C17.2) cells in tumor and liver metastases, and with that of CD4-naive (C00) and CD4-Tcm (C03) cells in LNs (Supplementary Fig. 6). CD4-Tfh cells in LNs had high expression of naive/progenitor-associated genes (*TCF7, IL7R, CCR7, LEF1, KLF2, SELL, GPR183*) and chemokine receptors (*CXCR4, CXCR5*) compared with CD4-Tex cells in tumor and liver metastases. CD4-Tfh cells in LNs also had high expression of genes encoding co-stimulatory molecules (*TIGIT, LAG3, TNFRSF4, TNFRSF18*), a gene related to T-cell exhaustion (*TOX*), proliferation genes (*DUSP4, DUSP6*), and chemokine receptor genes (*CXCL13, CXCR3*) compared with CD4-naive/Tcm cells in LNs. These data suggest that CD4-Tfh cells have proliferative functions in LNs, possibly interacting with CD8-Tpex cells, while still maintaining progenitor cell characteristics[31].

**Fig. 1 | Overview of single-cell RNA-seq of tissue-infiltrating T cells from two surgically resected esophagogastric junction cancers after ICI treatment.** **a** Schematic of treatment timeline and sample collection across two patients (patients 1, 2) who underwent surgical resection after ICI treatment (anti-PD-1 therapeutic antibodies: aPD-1). Four samples of tumor, metastatic LN, metastasis-free LN, and blood from patient 1 (esophagogastric junction cancer of adenocarcinoma), and three samples of liver metastases, metastasis-free LN, and blood from patient 2 (esophagogastric junction cancer of squamous cell carcinoma) were collected. Created in BioRender. Nose, Y. (2026) https://BioRender.com/x46k1n3. **b**–**d** Uniform manifold approximation and projection (UMAP) plot of 19388 CD3+ T cells in single-cell RNA-seq colored by Leiden cluster (**b**) marker genes (**c**) tissue of origin (**d** left), and patient (**d** right). The number of cells per cluster was evaluated by projecting the previously reported single-cell RNA-seq data on the self-administered data using Symphony. **e** Heatmap showing the similarity of clusters.

The tissue distribution for each cluster is shown at the top of the heatmap. **d**, **e** Liver metastases were defined as 'Meta' and metastatic LN and metastasis-free LN together as 'LN'. **f** UMAP plot of $TCF7^+PDCD1^+CD8^+$ (left) and $TCF7^+PDCD1^+CD4^+$ T cells (right). **g** Scatter plot of differentially expressed genes between CD8-Tpex (C05) cells in LNs and those of CD8-Tex (C04, C06.0, C17.0, C17.1) and CD8-Tpex (C12) cells in tumor and liver metastases (represented by blue dots, y-axis), and with those of CD8-naïve (C11) cells in LNs (represented by red dots, x-axis). Differentially expressed genes in both comparisons are shown in green dots. Genes with normalized expression >0.3 in C05 are shown. Log2 fold change >1 and adjusted P-value < 0.05 were considered significant. Only genes with mean expression in C05 cells >0.3 were used for the analysis. Two-sided tests were used for differential expression analysis. **h** Heatmap of differentially expressed genes. **e**, **h** CD8-Tpex (C05) is highlighted in red boxes.

## Relationship between intratumoral immune cell clusters and prognosis in ICI-naïve gastric cancer patients from the Cancer Genome Atlas

We next examined whether intratumoral CD8-Tpex cells correlate with prognosis in ICI-naïve patients, using the Cancer Genome Atlas (TCGA) dataset of gastric adenocarcinoma. Traditional clustering methods often categorize cells into discrete groups. These may fail to capture the continuum of cellular states, especially in dynamic or transitional populations, where many subsets exhibit overlapping gene expression profiles and functional states. Therefore, we applied non-negative matrix factorization (NMF), which is useful for profiling T cells with ambiguous boundaries[36,37]. We analyzed our scRNA-seq dataset to extract ten gene programs (Fig. 2a), in which the top 8 genes of each cluster by NMF are presented (Fig. 2b). In these ten clusters, NMF8 corresponded with CD8-Tpex, NMF3 with CD8-Tex, NMF2 with CD4-Tfh, and NMF0 with CD4-Treg (Supplementary Table 2). The pathway analysis of the top 100 characterized genes for NMF of CD8-Tpex revealed that the featured pathways of NMF8 were "Interferon signaling", "Translocation of ZAP-70 to immunological synapse", and "PD-1 signaling" (Fig. 2c, Supplementary Fig. 7). We projected these gene programs to the TCGA dataset of gastric adenocarcinoma. When the association between the abundance of each cluster and overall survival was examined, only NMF0, a cluster related to CD4-Treg, correlated with prognosis, while no significant associations were observed for the other clusters, including NMF8/NMF3, which reflected the Tpex/Tex ratio (Fig. 2d, e). TCGA evaluates gene expression of tumors treated without ICI. Thus, the result indicated that the intratumoral accumulation of CD8-Tpex and Tpex/Tex ratio would not affect the prognosis of ICI-naïve gastric cancer.

## The abundance of CD8-Tpex in LNs was correlated with better prognosis in patients with ICI-naïve gastric cancer

Since there are few published data on the function and prognostic impact of Tpex cells in LNs of human samples, we next collected surgically dissected tissue samples of tumor ($n = 39$), metastatic LNs ($n = 24$), and metastasis-free LNs ($n = 193$: proximal 134, distal 59) from 55 patients with ICI-naïve gastric cancer, and analyzed immune cells by flow cytometry (Fig. 3a, Supplementary Fig. 8, Supplementary Table 3). The LNs were defined by location as proximal LNs (pLN) and distal LNs (dLN), based on the Japanese Gastric Cancer Treatment Guidelines[38]. First, the expression frequency of various checkpoint molecules, such as PD-1, Tim-3, and CD103 among CD8+ T cells in each tissue was analyzed (Fig. 3b). The expression frequency of each molecule was high on intratumoral CD8+ T cells, and it was higher in pLNs than in dLNs within metastasis-free LNs, suggesting that anti-tumor immunity was induced in LNs close to the tumor. Although no difference in cytokine production between T cells in dLN and pLN was observed, we noted higher cytokine production by PD-1+ T cells than by PD-1− T cells in metastasis-free LNs, both in CD8+ and CD4+ T cells (Supplementary

Fig. 89a, b). PD-1+ T cells are further divided into three subpopulations by the presence or absence of TCF1 expression based on previous reports: Tpex (TCF1+PD-1+ T cells), Tex (TCF1−PD-1+ T cells), and PD-1− T cells (Fig. 3c)[20,27,28]. Among CD8+ T cells, the frequency of CD8-Tpex was higher in pLN than in the other three regions, whereas the frequency of CD8-Tex was highest in tumors[39] (Fig. 3d). The frequency of CD8-Tpex in metastasis-free LNs was lower in advanced tumor stages (Fig. 3d). Although CD8-Tpex in metastasis-free LNs did not express Ki-67 and GZMB at high levels (Fig. 3e), a higher CD8-Tpex/Tex ratio in metastasis-free LNs was associated with a better prognosis (Fig. 3f, Supplementary Fig. 10a, b). Multivariate analysis revealed that a high CD8-Tpex/Tex ratio in metastasis-free LNs tended to be a statistically significant prognostic factor (Supplementary Table 4). Regarding CD4+ T cells, the expression frequency of PD-1 and Tim-3 was high in intratumoral CD4+ T cells, and it was higher in pLNs than in dLNs within metastasis-free LNs. Although intratumoral CD103+CD4+ T cells have been reported to exhibit high cytokine production and are associated with a good prognosis[40,41], CD103 was barely detected in CD4+ T cells of LNs (Fig. 3g). CD4-Tfh cells were abundant in pLN than dLNs (Fig. 3h), but CD4-Tfh cells in metastasis-free LNs did not express high Ki-67 and GZMB (Fig. 3i). In contrast, CD4-Tex cells in metastasis-free LNs had relatively high GZMB expression among CD4+ T cells, although the GZMB expression in CD4+ T cells was considerably lower than that in CD8+ T cells (Fig. 3i). A higher CD4-Tfh/Tex ratio in metastasis-free LNs was not associated with a better prognosis (Fig. 3j). In summary, CD8-Tpex cells in metastasis-free LNs are less proliferative and less cytotoxic, but are associated with a favorable prognosis, suggesting that they may act as reservoirs of tumor-reactive T cells.

## The flow cytometry profiling of tissue-infiltrating T cells after ICI treatment

As it was suggested that Tpex cells in LNs may act as reservoirs of tumor-reactive T cells in ICI-naïve cancer patients, we then proceeded to evaluate tissue-infiltrating T cells after ICI treatment, using samples from 14 ICI-treated patients with gastrointestinal cancer. Since PD-1+ T cells could not be detected by conventional anti-PD-1 antibodies after ICI treatment due to the binding of aPD-1 to T-cell surface PD-1 molecules, we developed methods to identify ICI-bound T cells using anti-IgG4 antibodies, which detect T-cell-bound aPD-1 (Fig. 4a)[22]. This approach was validated in vitro to accurately detect ICI-bound T cells (Supplementary Fig. 11). The method detected ICI-bound T cells in tumor, LNs, and blood, consistent with previous reports of ICI-bound T cells in blood (Fig. 4b)[22–24]. Here, we encountered a case of gastric cancer treated with ICI followed by gastrectomy. The patient initially had multiple liver and LN metastases and had been treated with three regimens of chemotherapy for 18 months. After progression of the primary tumor, LNs, and liver metastases after chemotherapy, the patient received anti-PD-1 treatment as fourth-line, which resulted in shrinkage of the tumor and metastases. However, the primary tumor

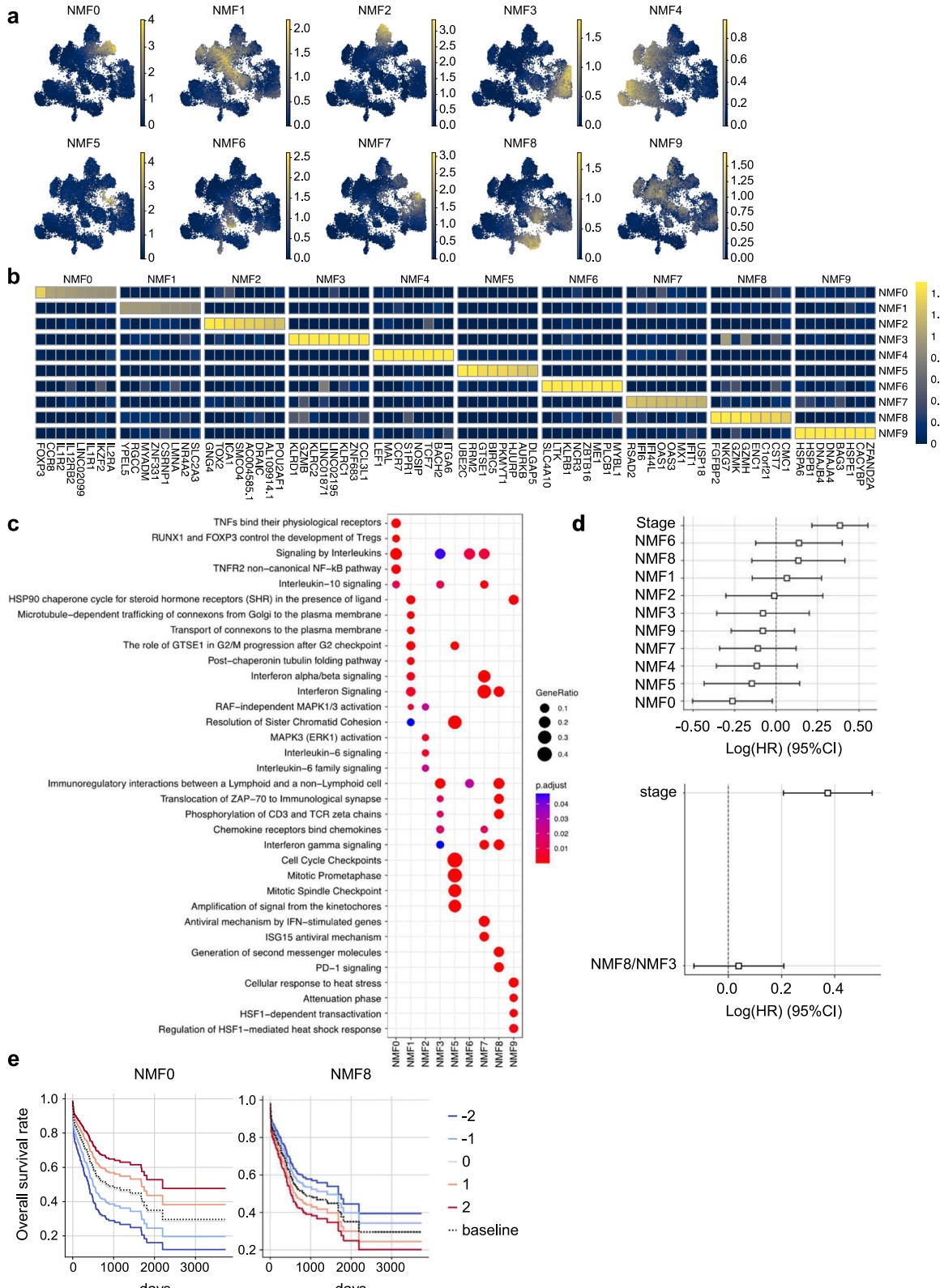

**Fig. 2 | The prognostic significance of intratumoral T-cell clusters determined by non-negative matrix factorization (NMF) analysis in the TCGA dataset of gastric cancer. a** UMAP plot showing the NMF cell feature with the single-cell RNA-seq dataset analyzed in Fig. 1. **b** Gene features for each NMF component. The top 8 genes for each feature were selected. **c** Significantly enriched REACTOME pathways in each NMF component. The node size represents the number of genes included in each pathway, and the color represents the adjusted *P*-value of the enrichment. NMF4 was excluded because no significant pathway could be detected. **d** Forest plot of the Cox hazards analysis for survival duration in the TCGA dataset of gastric adenocarcinoma, comparing each NMF group (upper) and NMF8/NMF3 (lower) based on NMF analysis (*n* = 385). The analysis incorporates NMFs and cancer stages (quantified as integers from 1 to 4) as covariates. Dots indicate hazard ratios estimated by Cox proportional hazards models, and error bars represent 95% confidence intervals. **e** Kaplan–Meier plots for varying covariates of NMF0 (left) and NMF8 (right) (*n* = 385).

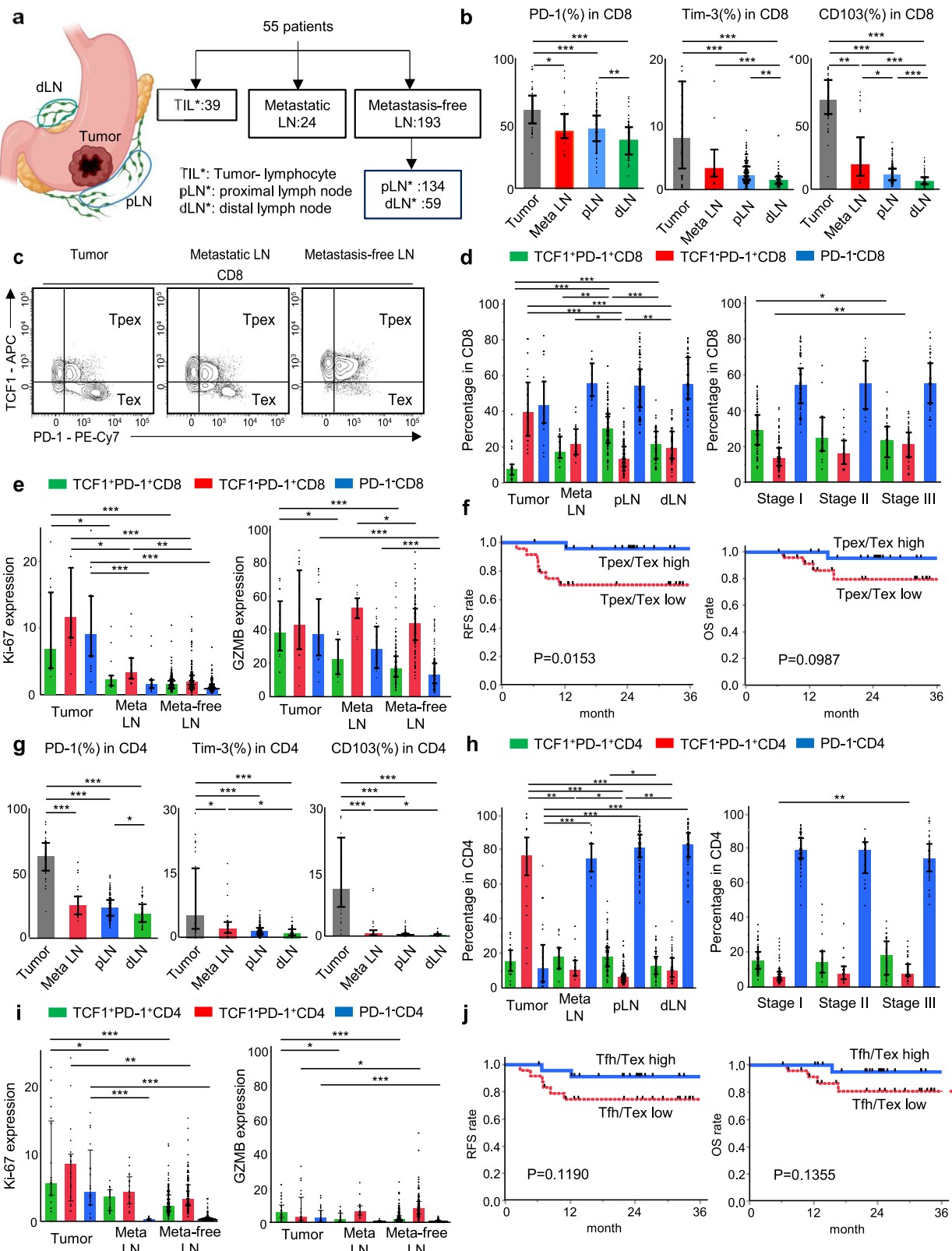

subsequently regrew, with the liver metastases diagnosed as complete response by imaging, and a gastrectomy was performed (Fig. 4c). The flow cytometry analysis demonstrated high proliferative activity of ICI-bound CD8+ T cells compared with ICI-non-bound CD8+ T cells in all metastasis-free LNs (Fig. 4d). These results prompted us to further investigate the dynamics of ICI-bound T cells in ICI treatment.

## ICI-bound Tpex in metastasis-free LNs were highly proliferative after ICI treatment

To evaluate the effect of ICI treatment on the function of Tpex in LNs, we collected surgically dissected tissue samples of tumor ($n = 13$), metastatic LNs ($n = 6$), metastasis-free LNs ($n = 55$), PBMC ($n = 12$), and liver metastases ($n = 2$) from 14 patients with gastrointestinal cancers (3 gastric, 7 esophageal, 3 EGJ, and 1 colorectal) treated with ICI, in

**Fig. 3 | The higher Tpex/Tex cell ratio in metastasis-free lymph nodes was associated with better prognosis in ICI-naive gastric cancer. a** Schematic of sample collection across 55 patients with ICI-naive gastric cancer who underwent surgical resection (39 tumors, 24 metastatic LNs, and 193 metastasis-free LNs of which 134 were proximal and 59 distal). Created in BioRender. Nose, Y. (2026) https://BioRender.com/x46k1n3. **b** Quantification of the frequency of PD-1$^+$, Tim-3$^+$, and CD103$^+$ cells among CD8$^+$ T cells from the tumor ($n = 38$), metastatic LNs ($n = 24$), proximal LNs ($n = 134$), and distal LNs ($n = 59$) by flow cytometry analysis. **c** Representative flow cytometry plots for TCF1 and PD-1 staining of CD8$^+$ T cells in each tissue (Tpex: TCF1$^+$PD-1$^+$CD8, Tex: TCF1$^-$PD-1$^+$CD8). **d** Percentage of TCF1$^+$PD-1$^+$, TCF1$^-$PD-1$^+$, and PD-1$^-$ cell population among CD8$^+$ T cells in each tissue (left) and in metastasis-free LNs according to tumor stage (right; Stage I: $n = 74$, Stage II: $n = 22$, Stage III: $n = 37$). **e** Quantification of the frequency of Ki-67 (left) and GZMB (right) among TCF1$^+$PD-1$^+$CD8$^+$, TCF1$^-$PD-1$^+$CD8$^+$, and PD-1$^-$CD8$^+$ T cells in each tissue. **f** Kaplan–Meier curves for recurrence-free survival (RFS: left) and overall survival (OS: right) were compared by classifying each group into the value of Tpex/Tex high or low in metastasis-free LNs (Tpex/Tex high: $n = 25$, Tpex/Tex low: $n = 24$). The value of Tpex/Tex was calculated as the ratio of TCF1$^+$PD-1$^+$ to TCF1$^-$PD-1$^+$ among CD8$^+$ T cells and divided into two groups, above or below the median,

respectively. When multiple metastasis-free LNs exist per case, the median Tpex/Tex was used. **g** Quantification of the frequency of PD-1$^+$, Tim-3$^+$, and CD103$^+$ cells among CD4$^+$ T cells from the tumor ($n = 37$), metastatic LNs ($n = 24$), and proximal ($n = 132$) and distal LNs ($n = 57$) by flow cytometry analysis. **h** Percentages of TCF1$^+$PD-1$^+$, TCF1$^-$PD-1$^+$, and PD-1$^-$ cells among CD4$^+$ T cells in each tissue (left) and in metastasis-free LNs according to tumor stage (right; Stage I: $n = 72$, Stage II: $n = 22$, Stage III: $n = 37$). **i** Quantification of the frequency of Ki-67 (left) and GZMB (right) expression among TCF1$^+$PD-1$^+$CD4$^+$, TCF1$^-$PD-1$^+$CD4$^+$, and PD-1$^-$CD4$^+$ T cells in each tissue. **j** Kaplan–Meier curves for recurrence-free survival (RFS: left) and overall survival (OS: right) were compared by classifying each group as Tfh/Tex high or low in metastasis-free LNs ($n = 49$). The value of Tfh/Tex was calculated as the ratio of TCF1$^+$PD-1$^+$ to TCF1$^-$PD-1$^+$ among CD4$^+$ T cells and divided into two groups, above or below the median, respectively. **b, d, e, g, h, i** Error bars indicate median ± interquartile range. The significance of differences was calculated using the Kruskal–Wallis test followed by Dunn's post hoc test (*$P < 0.05$, **$P < 0.01$, ***$P < 0.001$). All tests were two-sided. **d, e, h, i** Differences were tested between tissue types in each cell population (Tumor: $n = 23$, Metastatic LNs: $n = 16$, Metastasis-free LNs: $n = 133$ [proximal: $n = 85$, distal: $n = 48$]). **f, j** $P$-values were calculated by using the log-rank test.

which infiltrating immune cells were analyzed by flow cytometry (Supplementary Table 5). Since preoperative ICI treatment for gastrointestinal cancer is not approved worldwide, all subjects had initially unresectable cancer which then became resectable. First, the comparison between ICI-bound and ICI-non-bound CD8$^+$ T cells revealed that ICI-bound CD8$^+$ T cells exhibited higher CD103 expression and lower TCF1 expression than did ICI-non-bound CD8$^+$ T cells across all tissues, suggesting that ICI-bound CD8$^+$ T cells possess a memory phenotype, whereas ICI-non-bound CD8$^+$ T cells retain a naive phenotype (Supplementary Fig. 12a, b). Next, the expression frequencies of Ki-67 and GZMB in Tpex, Tex, and PD-1$^-$CD8$^+$ T cells were evaluated by flow cytometry in tumor and metastatic and metastasis-free LNs, and compared with those in ICI-naive gastric cancer. The results showed that the proliferative activity of Tpex in metastasis-free LNs significantly increased after ICI treatment, along with increased proliferative activity of other subgroups (Tex, PD-1$^-$ T cells) in metastasis-free LNs after ICI treatment (Fig. 4e). On the other hand, no increase in cytotoxicity, as evaluated by GZMB expression, was observed in any T cell subpopulation after ICI treatment (Fig. 4f). Regarding CD4$^+$ T cells, the proliferative activity of CD4$^+$ T cells in metastatic and metastasis-free LNs significantly increased after ICI treatment (Fig. 4g), and GZMB expression in Tex cells from metastasis-free LNs increased after ICI treatment, although the GZMB expression levels in CD4$^+$ T cells remained much lower than those in CD8$^+$ T cells (Fig. 4h). Although not statistically significant, patients with higher Ki-67 expression in CD8-Tpex within metastasis-free LNs tended to show better survival (Supplementary Fig. 13). Thus, ICI treatment primarily enhanced the proliferative potential of CD8$^+$ and CD4$^+$ T cells in LNs rather than their cytotoxicity. Notably, ICI treatment greatly increases the proliferative potential of CD8-Tpex cells in metastasis-free LNs.

## scRNA/TCR/CITE-seq analysis identified and characterized ICI-bound T cells

We confirmed that the proliferative activity of ICI-bound Tpex cells was boosted in metastasis-free LNs after ICI treatment, but it was still unknown whether these cell populations exert anti-tumor immunity at the tumor site. To investigate the impact of ICI treatment on the distribution of cell populations, we performed integrated scRNA-seq on five patients, incorporating one additional ICI-treated (patient 3) and two ICI-naive cases (patients 4, 5) as a validation cohort. In the analysis, CD8-Tpex cells in LNs (C5.0, C5.1), CD4-Tfh cells (C7), and intratumoral CD8-Tex cells (C6, C10) were identified. Notably, the distribution of these clusters across tissues did not differ substantially between ICI-treated and ICI-naive groups, suggesting that the effects of ICI treatment may depend more on the functional state of the cells rather than

on changes in the size or distribution of cell populations (Supplementary Fig. 14). To uncover the dynamics of T cells after ICI treatment, we investigated the TCR repertoire at the single-cell level in combination with CITE-seq detecting anti-IgG4 antibody with the scRNA-seq dataset of 19 samples from 3 ICI-treated patients and 2 ICI-naive patients (Fig. 5a, Supplementary Table 1a, b). We thought this analysis might allow us to detect ICI-bound T cells at the single-cell level and simultaneously evaluate the TCRs on the ICI-bound T cells. Indeed, the comparative analysis between ICI-treated and ICI-naive patients demonstrated that the anti-IgG4 antibody in CITE-seq identified ICI-bound T cells in samples from ICI-treated patients, while these cells were hardly detected in samples from ICI-naive patients (Fig. 5b). According to the median value of IgG4 binding levels, clusters with more than half of the cells being IgG4-positive cells were classified as IgG4-high clusters (Fig. 5c, Supplementary Table 6). Consequently, we observed abundant binding of aPD-1 to Tex and Tpex in both tumor and LNs (Fig. 5b–d). These ICI-bound T cells showed moderate to high levels of *PDCD1* mRNA expression, and IgG4 binding levels were proportional to *PDCD1* mRNA levels, meaning that all PD-1-expressing T cells at any site in the body were bound by aPD-1, depending on the level of surface PD-1 expression (Fig. 5b, Supplementary Fig. 15). In the comparison of gene expression between ICI-bound PD-1$^+$CD8$^+$ T cells from ICI-treated patients and PD-1$^+$CD8$^+$ T cells from ICI-naive patients, genes associated with antigen recognition and T-cell activation, such as *HLA-DRB1* and *HLA-DQA1*, were upregulated in tumor of ICI-treated patients, whereas genes related to cell migration and interferon signaling, including *CCL3L1*, *CCL4L2*, and *GBP5* were elevated in LNs. In contrast, no significant differences in exhaustion-related genes were observed in the blood, suggesting that the effects of ICIs are most pronounced in LNs and tumors (Supplementary Fig. 16a). In a comparison of gene expression between ICI-bound and ICI-non-bound CD8$^+$ T cells in tumors from ICI-treated patients, ICI-bound CD8$^+$ T cells expressed *CXCL13*, *PDCD1*, *TIGIT*, and *LAG3*, while ICI-non-bound CD8$^+$ T cells expressed *IL7R* and *LTB* (Supplementary Fig. 16b), indicating that ICI-bound CD8$^+$ T cells exhibit an exhausted phenotype, whereas ICI-non-bound CD8$^+$ T cells retain a naive phenotype. In addition, paired TCR data with phenotypic analysis showed that these ICI-bound T cells expanded clonally, with the top 20 overlapping clones between LNs and tumor or liver metastases occupied by these ICI-bound T cells (Fig. 5e, f).

## STARTRAC analysis unveiled the clonal link between ICI-bound CD8-Tpex in LNs and intratumoral ICI-bound CD8-Tex

To analyze the dynamic relationships among T-cell clusters with distinct gene signatures and clonalities, we analyzed the scRNA-seq data

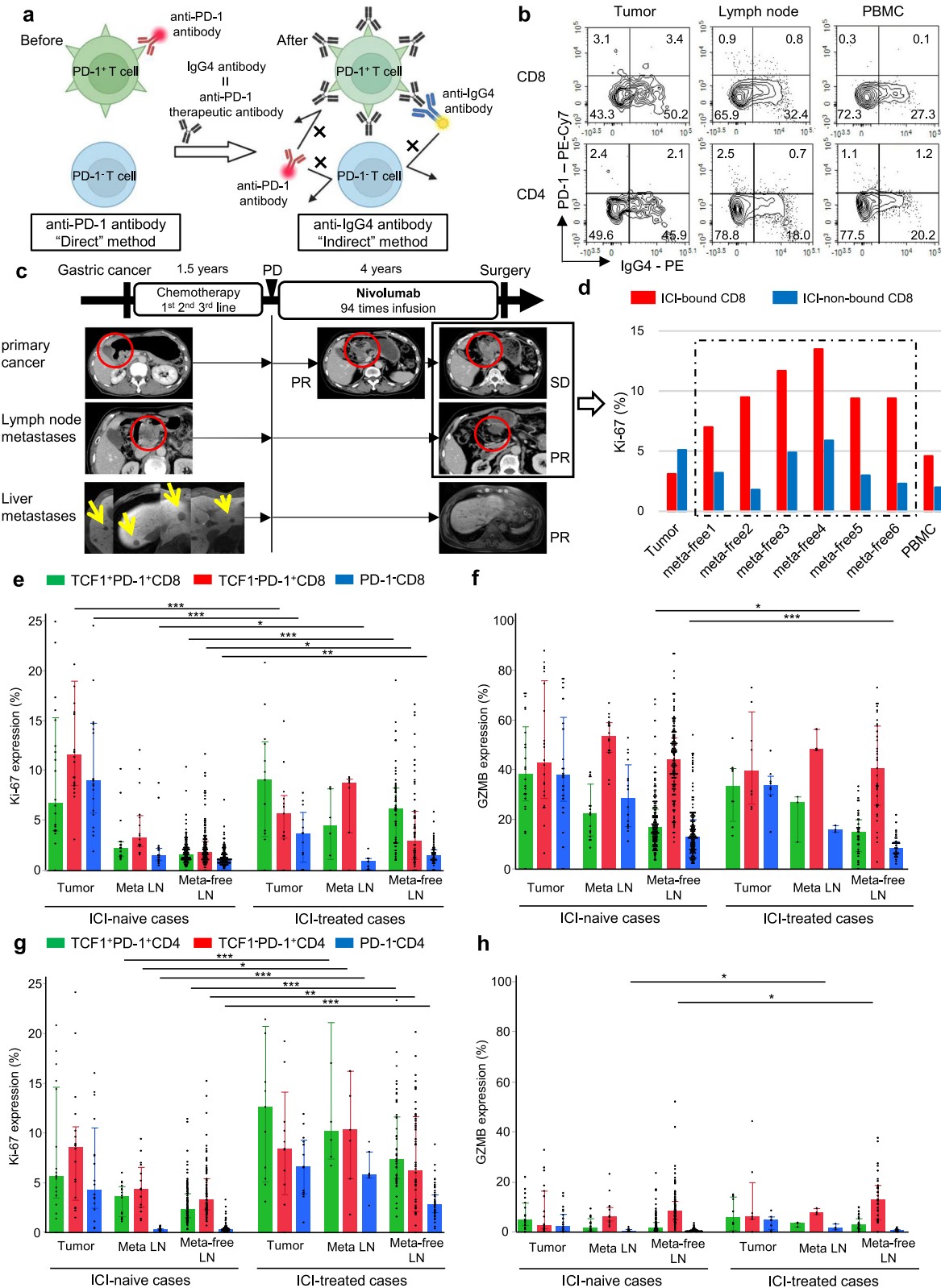

from 3 ICI-treated (patients 1, 2, 3) and 2 ICI-naive patients (patients 4, 5) using the RNA sequencing and TCR tracking (STARTRAC) algorithm[42] (Supplementary Fig. 15). This algorithm incorporates several indices, including distribution, expansion, migration, and transition, to quantitatively describe tissue distribution, clonal expansion, migration, and developmental transition or differentiation, respectively. First, the comparison of STARTRAC results between ICI-treated and ICI-naive patients demonstrated that the expansion score was increased in CD8-Tex (C04, C17.0), CD8-Tpex (C05, C12), and CD4-Tex (C06.1) cells after ICI treatment (Fig. 6a). The cross-tissue migration score was increased in CD8-Tex (C04, C06.0, C17.0), CD8-Tpex (C05), CD8-effector (C08), and CD4-Tex (C06.1) cells. Conversely, the transition score was decreased in CD8-Tex (C04, C06.0, C17.0), CD8-Tpex (C12), and CD8-effector (C08) cells. Thus, increased expansion and migration scores, along with decreased transition score after ICI treatment, were predominantly observed in IgG4-high clusters. In the

**Fig. 4 | The proliferation of Tpex in lymph nodes, but not the cytotoxicity, was greatly increased after ICI treatment. a** Scheme of the method to identify ICI-bound T cells using anti-IgG4 antibodies. Therapeutic anti-PD-1 IgG4 antibodies bound to surface PD-1 molecules on T cells can be detected when they are bound by anti-IgG4 antibodies. Created in BioRender. Nose, Y. (2026) https://BioRender.com/x46k1n3. **b** Representative flow cytometry plots for PD-1 and IgG4 staining of CD8[+] (top) and CD4[+] (bottom) T cells after ICI treatment in tumor, LNs, and PBMC. **c** Scheme of the clinical course in a patient with gastric cancer treated with ICI treatment (Nivolumab) followed by surgical resection. Red circles indicate the primary tumor and metastatic lymph nodes, and yellow arrows denote liver metastases. **d** Quantification of the frequency of Ki-67 among ICI-bound and non-bound CD8[+] T cells in each tissue. The data from metastasis-free LNs (meta-free) are highlighted within the dotted line. **e, f** Quantification of the frequency of Ki-67 (**e**) and GZMB (**f**) expression among TCF1[+]PD-1[+]CD8[+], TCF1[-]PD-1[+]CD8[+], and PD-1[-]CD8[+]

T cells of tumor, metastatic LNs, and metastasis-free LNs in ICI-naive and ICI-treated cases. **g, h** Quantification of the frequency of Ki-67 (**g**) and GZMB (**h**) expression among TCF1[+]PD-1[+]CD4[+], TCF1[-]PD-1[+]CD4[+], and PD-1[-]CD4[+] T cells of tumor, metastatic LNs, and metastasis-free LNs in ICI-naive and ICI-treated cases. The number of cases included tumor ($n = 22$), metastatic LNs ($n = 16$), and metastasis-free LNs ($n = 133$) in ICI-naive cases in (**e**–**h**); tumor ($n = 13$), metastatic LNs ($n = 6$), and metastasis-free LNs ($n = 55$) in ICI-treated cases in (**e**) and (**g**); and tumor ($n = 8$), metastatic LNs ($n = 3$), and metastasis-free LNs ($n = 35$) in ICI-treated cases in (**f**) and (**h**). **e**–**h** Error bars indicate the median ± interquartile range. Differences were tested between the same cell populations derived from the same tissue types with and without ICI treatment. The significance of differences was calculated using the non-parametric two-sided Wilcoxon rank-sum test comparing ICI-naive and ICI-treated patients within each tissue and cell population (*$P < 0.05$, **$P < 0.01$, ***$P < 0.001$).

analysis with 3 ICI-treated patients, a high expansion score was observed in CD8-Tex (C04, C06.0, C19), CD8-Tpex (C05, C12), and CD8-effector (C08) cells (Supplementary Fig. 15b). A high cross-tissue migration score was observed in CD8-Tex (C04, C06.0, C17.0), CD8-Tpex (C05, C12) and CD8-effector (C08) cells. A high transition score was observed in CD8-Tex (C04, C06.0, C17.0, C17.1) cells. Thus, CD8-Tpex cells in LNs and tumors had a high potential to expand and migrate, and intratumoral CD8-Tex cells had a high potential to expand, migrate, and differentiate. All of these T cells with high expansion, migration, and transition scores were derived from IgG4-high clusters, except for C08. The high scores in the C08 cluster may be attributed to the presence of PD-1-negative recently activated effector memory (TEMRA) T cells within C08 cells, which have been reported to exhibit high proliferation, migration, and transition scores, as well as strong migratory potential between blood and organs, even without ICI treatment[42]. In contrast, CD4-Tfh in LNs showed limited potential for clonal expansion after ICI treatment (Fig. 5e, Supplementary Table 7). Moreover, STARTRAC analysis revealed that these cells exhibit low levels of migration and transition (Fig. 6a, b). Overall, these findings suggest that ICI-bound T cells play a leading role after ICI administration, possibly by the enhanced expansion and migration of PD-1[+]CD8[+] T cells.

Regarding migration, the most observed route of migration was between LN and tumor (Supplementary Fig. 15c). The LN-enriched CD8-Tpex (C05) had a moderate expansion score and a high cross-tissue migration score (Supplementary Fig. 15b), and exhibited the highest migration potential between LN and tumor among all clusters (Fig. 5f, Supplementary Fig. 15c). Notably, this migration potential of C05 between LN and tumor increased after ICI treatment (Fig. 6b). Importantly, in ICI-treated cases, TCR clonotypes within CD8-Tex (C04, C06.0) in tumor were highly shared with CD8-Tpex (C05, C12) in LNs or tumor, in contrast to ICI-naive cases (Fig. 6c, Supplementary Fig. 17). Conversely, TCR clonotypes within the C05 cluster were highly shared with CD8-Tex (C04, C06.0) and CD8-Tpex (C12) in tumor and CD8-effector (C08) in blood, with 89.8% in patient 1, 97.9% in patient 2, and 53.1% in patient 3 (Supplementary Fig. 18a). Moreover, the C05 consisted of two subgroups with high and low IgG4 binding. After categorizing clonotypes based on high and low IgG4 binding, ICI high-bound T cells in the C05 cluster shared clones with CD8-Tex (C04, C06.0) and CD8-Tpex (C12) cells. Whereas, ICI low-bound T cells in the C05 cluster shared clones with CD8-effector (C08) cells in blood (Fig. 6d, Supplementary Fig. 18b). On the other hand, compared with C05 cells, ICI high- and low-bound T cells in the C08 cluster shared fewer clones with CD8-Tex (C04, C06.0) and CD8-Tpex (C12) cells in tumor (Supplementary Fig. 18c). Interestingly, ICI low-bound T cells in the C08 cluster exhibited higher expansion, migration, and transition scores than ICI high-bound T cells (Supplementary Fig. 19). This may be because TEMRA cells with low PD-1 expression were included in ICI low-bound T cells within the C08 cluster. Indeed, it has been reported

that the clonotypes of blood TEMRA cells were mutually exclusive with those of intratumoral Tex cells[42], which may explain why the clonal similarity of C08 cells to intratumoral CD8-Tex cells was relatively lower than that of C05 cells. Additionally, pseudotime analysis within C05, C12, and C04 clusters across organs revealed a differentiation flow from C05 to C12 and then to C04, suggesting that CD8[+] T cells differentiate sequentially from CD8-Tpex in LNs to CD8-Tpex in tumor, and ultimately to CD8-Tex in tumor (Supplementary Fig. 20). Regarding CD4[+] T cells, TCR similarity between CD4-Tfh in LN (C09) with intratumoral CD4-Tex (C06.1) appears to be relatively low compared with that of CD8[+] T cells (Fig. 6c, Supplementary Fig. 21). Overall, these findings suggest that ICI high-bound Tpex cells in LNs serve as a source of Tex cells in tumor, possibly containing cancer antigen-specific T cells, whereas ICI low-bound Tpex cells in LNs may have a different antigen specificity.

Finally, the comparison of gene expression between the ICI high- and low-bound CD8-Tpex in LNs showed that the ICI low-bound T cells expressed high levels of naive/progenitor-associated genes (*TCF7, KLF2, IL7R, CCR7, SELL*) (Fig. 6e). On the other hand, the ICI high-bound T cells expressed high levels of genes encoding effectors (*GZMA, GZMK*), co-stimulatory molecules (*TIGIT, LAG3, SLAMF6, ICOS, PRDM1, TNFSF9, CD82, CD27*), an IFN response gene (*IFNG*), chemokine ligands and receptors (*CCL4*), and differentiating molecules (*DUSP4, DUSP5*). Taken together, these data support the hypothesis that ICIs target Tpex in LNs, leading to their proliferation and migration to tumor, where these cells differentiate into Tex exerting anti-tumor immunity.

## Discussion

Since the binding of aPD-1 to T cells is a possible mechanism of treatment efficacy with ICIs, mechanistic analysis for the dynamics of ICI-bound T cells has great importance. In this study, we deeply explored the function of ICI-bound T cells by scRNA-seq and flow cytometry analysis. We first identified *PDCD1[+]TCF7[+]* Tpex primarily within LNs and tumors. Interestingly, abundant CD8-Tpex presence in metastasis-free LNs was linked to a favorable prognosis in ICI-naive gastric cancer, whereas that in tumors or metastatic LNs showed no such association. Regarding ICI-bound T cells after ICI treatment, we found that PD-1[+] T cells are bound by aPD-1 in tissues throughout the body. The proliferative activity of Tpex in LNs was significantly increased after ICI treatment. scRNA/TCR/CITE-seq revealed the increased expansion and migration potential of PD-1[+] T cells after ICI treatment. Importantly, ICI high-bound CD8-Tpex in LNs shared clones with intratumoral ICI-bound CD8-Tex, while ICI low-bound CD8-Tpex shared clones with blood CD8[+] T cells. The results indicated that ICIs target Tpex in LNs, leading to their proliferation and migration to the tumor, where ICI high-bound Tpex differentiate into Tex and actively engage in tumor destruction.

Possible mechanisms of efficacy by ICI treatment are the reinvigoration of intratumoral terminally exhausted T cells, local expansion

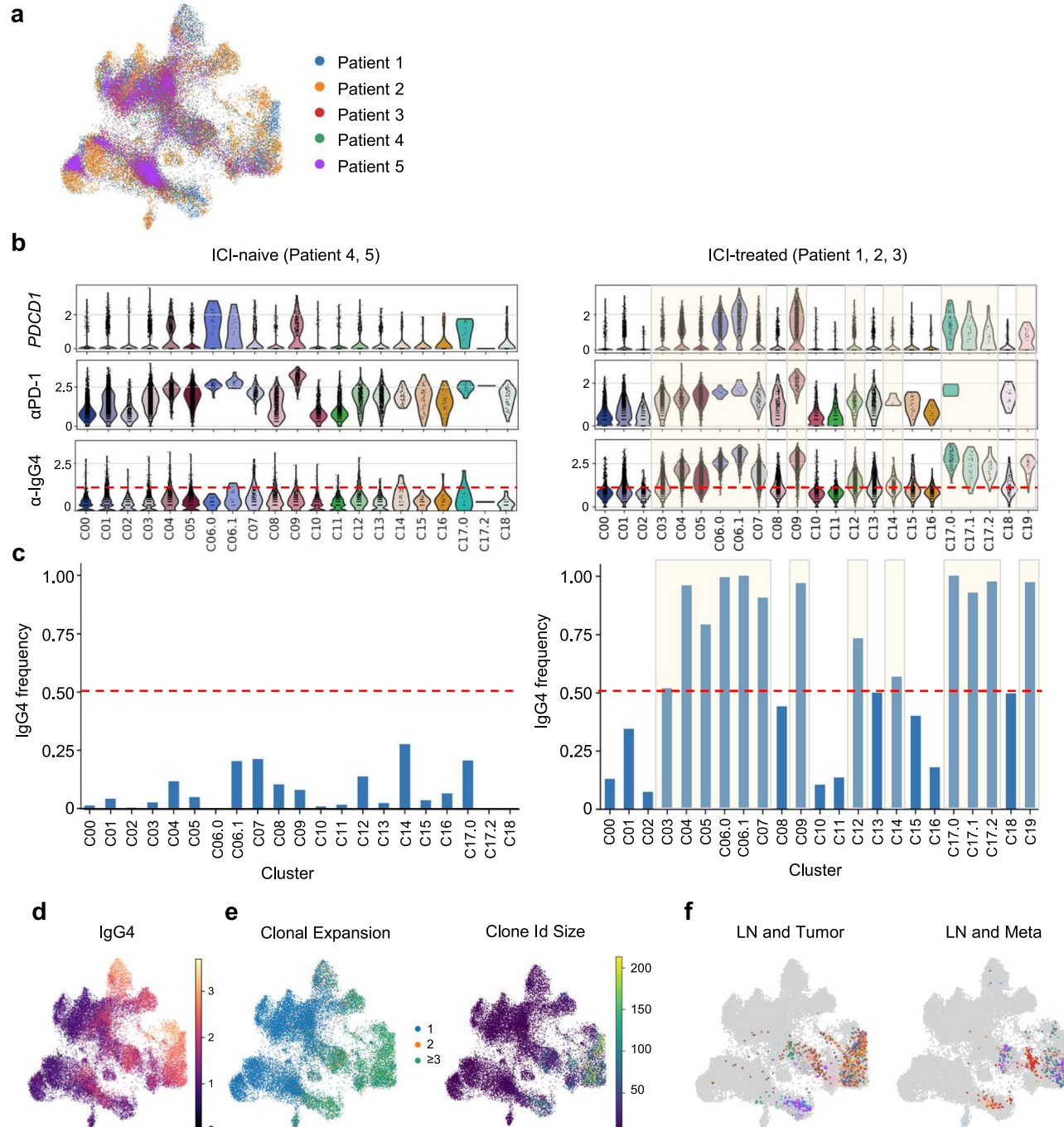

**Fig. 5 | The frequency of IgG4 binding in ICI-naive and ICI-treated patients in single-cell RNA/TCR/CITE-seq analysis and single-cell RNA/TCR/CITE-seq analysis identified clonal expansion of ICI-bound CD8-Tpex after ICI treatment.** **a** UMAP plot colored by the patient identity in all five patients. **b** Violin plots showing PDCD1 mRNA expression (top), anti-PD-1 expression frequency (middle), and anti-IgG4 expression frequency (bottom) in ICI-naive (patients 4, 5) and ICI-treated patients (patients 1, 2, 3). The red line indicates the cutoff value for anti-IgG4 expression, with the positive IgG4 binding levels highlighted in orange boxes (bottom). **c** The frequency of IgG4-bound T cells in each cluster in ICI-naive and ICI-treated patients. The red line indicates the cutoff value for anti-IgG4 expression. **b**, **c** Clusters with high-IgG4 binding are highlighted in orange boxes. **d** UMAP plot of bound IgG4 in single-cell RNA/TCR/CITE-seq from 27257 CD3+ T cells of three ICI-treated cancer patients (patients 1, 2, 3). **e** UMAP plot showing clonotype sizes in low-size categories (left) and actual numbers (right). The color indicates the number of cells that share the same T-cell receptor (TCR) sequences. In the left plot, clone sizes are specifically labeled to represent categories with clone sizes 1, 2, and greater than 3. **f** UMAP projection of top 20 clones overlapping between LN and tumor (left), and LN and liver metastases (right). Liver metastases were defined as 'Meta', and metastatic and metastasis-free LN together as 'LN'.

---

of intratumoral progenitor-exhausted T cells, or replenishment by peripheral T cells with both new and pre-existing clonotypes[21,43,44]. It was initially thought that ICI reinvigorates pre-existing cells within the TME[45], but investigations have shown that the terminally exhausted phenotype is epigenetically locked and difficult to modify[46]. Recent

works suggest that T-cell responses to ICIs may originate outside the tumor and rely on the recruitment of peripheral T cells[14,43,44,47,48], and TDLNs may act as a reservoir of ICI treatment-responsive, tumor-reactive T cells[25,32–34,49,50]. More recent mouse work on the dynamics of tumor-specific CD8+ T cells proposed two phases of tumor-specific

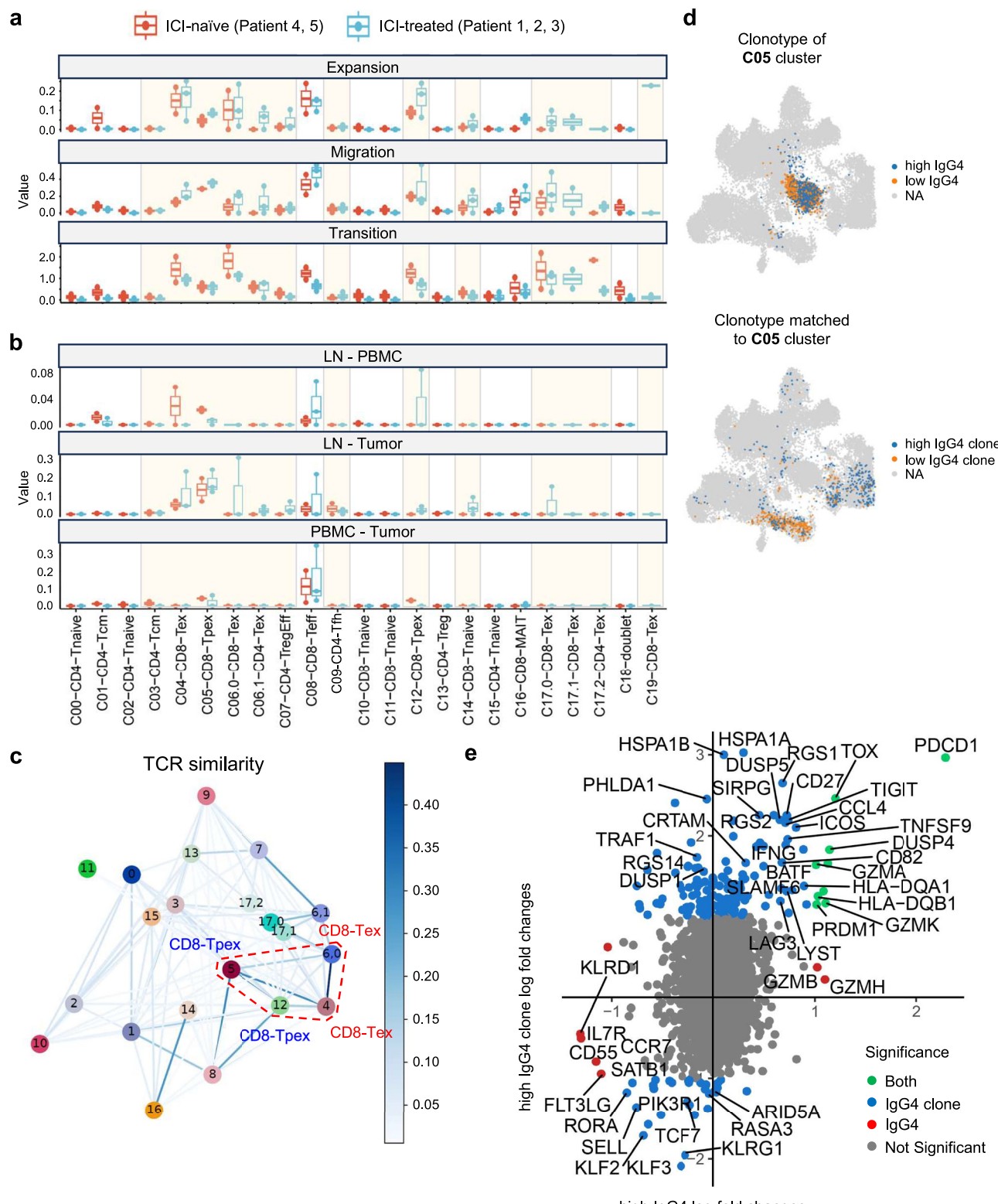

CD8+ T-cell activation, consisting of initial activation in TDLNs, followed by subsequent acquisition of effector programs within the tumor after additional co-stimulation[51]. Consistent with these findings, our data showed that ICI administration promoted the proliferation of ICI-bound Tpex in LNs. Furthermore, ICI high-bound CD8+ Tpex in LNs were TCR-matched to ICI-bound Tex in the tumor, implying that ICI high-bound CD8+ Tpex migrate to the tumor. These data indicate that ICI treatment is not based on the reversal of T-cell exhaustion programs but on the proliferation of a progenitor T cells.

PD-1+ T cells in tumors and blood have been shown to be cancer antigen-specific T cells[11,12]. Indeed, response to ICI treatment has been associated with the abundance of tumor-infiltrating PD-1+CD8+ T cells in several types of cancer[13–16]. Thus, a detailed understanding of the dynamics of PD-1+ T cells is crucial to understanding the mechanism of the therapeutic effect of ICIs. We proved that the anti-IgG4 antibody detection method using CITE-seq is powerful in evaluating the dynamics of ICI-bound T cells at the single-cell level. The method unveiled the TCR sharing between ICI high-bound Tpex in LNs and

**Fig. 6 | Comparison of STARTRAC analysis results between ICI-treated and ICI-naive cancer patients. a** Potentials of clonal expansion (top), tissue migration (middle), and transition (bottom) of T cells in each cluster qualified by overall STARTRAC-expansion, migration, and transition indices from two ICI-naive (patients 4, 5) and three ICI-treated cancer patients (patients 1, 2, 3). **b** Comparison of migration potentials of T cells in each cluster by pairwise STARTRAC-migration indices from two ICI-naive and three ICI-treated patients. **a, b** Clusters with high-IgG4 binding are highlighted in orange boxes. Red boxes indicate parameters from ICI-naive patients. Blue boxes indicate parameters from ICI-treated patients. The box plots show the median (center line), the interquartile range (25th–75th percentiles; box), and the minimum to maximum values (whiskers). **c** TCR similarity networks for each cluster in ICI-treated cases. TCR similarities were calculated for clones observed in Tumor, Meta, or LN. Jaccard Index is used for the similarity

metric (see "Methods"). Edges with TCR similarity >0.005 are shown. CD8-Tpex and CD8-Tex, which showed high similarity, are highlighted in red dot boxes. **d** UMAP plot showing the CD8-Tpex clonotype in LNs (C05) classified by high and low levels of IgG4 binding (upper), and T cells sharing the clonotype with these CD8-Tpex subpopulations in the other clusters (lower). **e** Scatter plot of genes expressed differentially between ICI high- and low-bound CD8-Tpex in the C05 (represented by red dots, x-axis), and between ICI high- or low-bound CD8-Tpex in the C05 and each T-cell population sharing the clonotype with these CD8-Tpex as shown in Fig. 6d (represented by blue dots, y-axis). Differentially expressed genes in both comparisons are shown in green dots. Log2 fold change >1 and adjusted $P$-value < 0.05 were considered significant. Genes with normalized expression >0.3 in C05 are shown. Two-sided tests were used for differential expression analysis.

intratumoral ICI-bound Tex, and between ICI low-bound Tpex in LNs and T cells in blood. These results suggest that high *PDCD1* mRNA-expressing Tpex in LNs may respond to cancer antigens, while low *PDCD1* mRNA-expressing Tpex in LNs recognize the other antigens. The intensity of antigen stimulation in LNs may explain these differences in *PDCD1* mRNA expression levels. Considering the mechanism of anti-tumor immunity by ICIs, the method of increasing PD-1⁺ T cells in LNs can be one of the therapeutic strategies to enhance anti-tumor immunity. In our data, ICI high-bound T cells in LNs had high levels of gene expression of effector transcripts (*GZMA, GZMK*), co-stimulatory molecules (*TIGIT, LAG3, ICOS, PRDM1, TNFSF9*), IFN response (*IFNG*), chemokine ligands and receptors (*CCL4*), and differentiation molecules (*DUSP4, DUSP5*) compared with ICI low-bound T cells. These genes were slightly different from the previously reported genes such as *SELL* and *SPRY1*, but these differences may be derived from the methods to profile ICI-bound T cells with CITE-seq[25,52]. Since our study is the first to analyze Tpex in paired samples of human tumors, LNs, and PBMC after ICI treatment, focusing directly on ICI-bound T cells, methodological variations may explain these discrepancies. Therefore, the inhibitors of the other co-stimulatory molecules, such as anti-TIGIT, LAG3, and ICOS antibodies, in combination with anti-PD-1 therapy could enhance anti-tumor immunity through regulation of Tpex in LNs.

Good therapeutic effects and prognosis of ICI treatment have been reported to be associated with abundant intratumoral Tex, intratumoral Tpex, and blood Tpex[16,19,27,28,47,53,54]. The lack of consistency in the significance of these cells on the therapeutic effect and prognosis may cause confusion. In our data, Tpex in LNs highly proliferated after ICI treatment, which may imply the association of abundant Tpex in LNs with good therapeutic effects and prognosis of ICI treatment. A recent mouse study showed that Tpex are activated in LNs and reactivated in tumors[51], and another study showed that Tpex are then activated in the tumor niche composed of CXCL13⁺ helper T cells and mregDC[29]. Taken together, Tpex are activated in the TDLNs, migrate to tumors, and differentiate to Tex in tumors. Thus, all of the above hypotheses between the T-cell subset and prognosis can be considered correct.

There were some limitations to this study, including the absence of experimental animal models with which to further explore causality and the inability to assess the difference between responders and non-responders. We can not sample non-responders, as preoperative ICI treatment is not currently indicated in gastrointestinal cancers. As NMF linearly decomposes gene expression, it is well-suited to capture terminal and well-defined gene programs such as those of Treg, Th1, or Th17 cells[36]. In contrast, Tpex cells exhibit an intermediate phenotype, and their representation in this framework may diverge from expectations, potentially reflecting their transitional nature. Additionally, the flow cytometry dataset included many discrete samples, but the scRNA-seq dataset was derived from only a few patients with gastrointestinal cancer. Thus, it is not yet clear whether our findings are generalizable to other tumor types after ICI treatment. Furthermore,

investigating the prognostic impact of the Tpex/Tex ratio in LNs of ICI-treated patients remains challenging due to the limited number of patients. Future larger prospectively designed studies will help to overcome these limitations. Finally, we used pseudotime analysis to infer the differentiation of CD8⁺ T cells; however, functional validation has not yet been performed. Nonetheless, our results suggest that ICI treatment exerts therapeutic efficacy by aPD-1 binding to CD8-Tpex in LNs, which could be detected using an anti-IgG4 antibody combined with single-cell sequencing.

## Methods

### Patients and samples

Fifty-five patients with pathologically diagnosed ICI-naive gastric cancer, who underwent surgery at Osaka University Hospital from November 2019 to February 2022, were included for flow cytometry analysis (Supplementary Table 3). Survival analysis and univariate and multivariate analysis were performed in 49 of these cases with data on metastasis-free LNs (Fig. 3f, j, Supplementary Table 4). Fourteen patients with gastrointestinal cancer (3 gastric, 7 esophageal, 3 EGJ, and 1 colorectal) who underwent surgery within one month of ICI treatment from September 2020 to October 2024 were included for flow cytometry analysis (Supplementary Table 5). TNM status was recorded according to the *Union for International Cancer Control TNM Classification of Malignant Tumors (7th edition)*[55]. Two ICI-treated patients (patients 1, 2) with EGJ cancer (1 adenocarcinoma, and 1 squamous cell carcinoma) were included for initial single-cell analysis, and a total of 7 samples (patient 1: PBMC, tumor, metastatic LN, metastasis-free LN; patient 2: PBMC, liver metastases, metastasis-free LN) were collected from 2 patients (Fig. 1, Supplementary Table 1a). One additional ICI-treated patient (patient 3) with gastric adenocarcinoma and two ICI-naive patients (patients 4, 5) with gastric cancer were included for scRNA/TCR/CITE-seq, and a total 12 samples (patient 3: tumor, two metastasis-free LNs, PBMC; patient 4: tumor, metastatic LN, metastasis-free LN, PBMC; patient 5: tumor, two metastasis-free LNs, PBMC) were collected from 3 patients (Figs. 5 and 6, Supplementary Table 1b). Demultiplexing was performed using HashSolo[56] with the default parameters. Next, the data were integrated with the discovery cohort reference using symphonypy. Tumor tissues, LN tissues, and peripheral blood were collected at the time of surgery. Multiple LN samples were taken from a single patient. LNs were divided into halves, and one was subjected to flow cytometry analysis while the other was submitted to histopathological examination to determine the presence or absence of metastasis. The collected samples were subdivided by gentleMACS Dissociator (Miltenyi Biotec, Bergisch Gladbach, Germany, #130-093-235) and purified by cell strainer (BD Biosciences, San Jose, USA, #352340), and mononuclear cells were extracted. Blood samples were centrifuged to extract mononuclear cells using Ficoll-Paque (Cytiva, #17144002). PBMCs were isolated from blood samples and stored in the nitrogen bank. To collect tissue-infiltrating lymphocytes, fresh tumor and LN tissues were minced and treated with the gentleMACS Dissociator[57]. Extracted lymphocytes

were stored in the nitrogen bank and analyzed with an LSR Fortessa cytometer (BD Biosciences) after thawing and washing. All donors provided written informed consent before sampling in accordance with the Declaration of Helsinki. This study was approved by the institutional ethics committees of Osaka University (approval number #13266, Osaka, Japan).

## Antibodies and reagents

The fluorescence-labeled antibodies used in the present study were the following: anti-CD3 (1/100, clone UCHT1; BioLegend, San Diego, CA, #300424), CD4 (1/100, clone OKT4; BioLegend, #317440), CD8 (1/100, clone RPA-T8; BioLegend, #301048/clone SK1; BD Biosciences, #565310), CD45RA (1/100, clone HI100; BioLegend, #304148), PD-1 (1/100, clone EH12.1; BD Biosciences, #561272), T-cell immunoglobulin mucin domain 3 (1/100, Tim-3, clone F38-2E2; BioLegend, #345012), CD25 (1/100, clone BC96; BioLegend, #302638), CD103 (1/100, clone Ber-ACT8; BioLegend, #350218), Ki-67 (1/50, clone Ki-67; BioLegend, #350506), TCF1/TCF7 (1/100, clone C63D9; Cell Signaling Technology, #37636), Granzyme B (1/100, clone QA16A02; BioLegend, #372212), IFN-γ (1/100, clone 4S.B3; BioLegend, #502506), TNFα (1/100, clone MAb11; BioLegend, #502950). ICI-bound T cells were detected by use of a biotinylated anti-human IgG4 antibody (1/4000, clone HP6025; Invitrogen, #A-10663) followed by secondary staining with streptavidin (1/200, BioLegend, #405204). Dead cells were identified using the Live/Dead Fixable Red Dead Cell Stain Kit (1/1000, Invitrogen, #L34993). Corresponding isotype control antibodies were purchased from the same manufacturer.

## Surface marker and intracellular staining

Thawed cells were incubated with antibodies at 4 °C for 30 min after washing. Cells washed with phosphate-buffered saline with 2% fetal calf serum were stained with CD3, CD4, CD8, CD25, and CD45RA antibodies, and fixable viability dye (Invitrogen). After washing, cells were analyzed with an LSR Fortessa (BD Biosciences), and the data obtained were analyzed using BD FACSDiva software. For intracellular staining, cells were washed and stained for surface markers. Cells were permeabilized with fix/perm solution (BD Biosciences, #554714) at 4 °C for 15 min. Cells were then stained with antibodies for intracellular molecules such as transcription factors (TCF1), Ki-67, and cytotoxic granules (GZMB) at 4 °C for 30 min. After washing, the cells were analyzed with an LSR Fortessa instrument.

## IgG4 staining method

We have previously established a method to identify PD-1$^+$ T cells to which aPD-1 bind by binding a secondary antibody, anti-IgG4, which binds directly to aPD-1 (Fig. 4a)[22]. Briefly, cells were stained with biotinylated anti-IgG4 antibody at 4 °C for 30 min to detect administered aPD-1 (human IgG4) bound to PD-1 molecules on the T-cell surface. The cells were washed twice and then incubated with streptavidin (PE) at room temperature for 15 min, followed by flow cytometry analysis. To confirm this method was detecting ICI-bound T cells correctly, the proportion of anti-IgG4 antibody in T cells after in vitro culture with aPD-1 was compared with that of anti-PD-1 antibody in T cells prior to culture, using PBMC samples from 29 ICI-naive gastric cancer patients (Supplementary Fig. 11). Following in vitro culture with aPD-1 (1 μg/ml) at 4 °C for 30 min, the anti-PD-1 antibody could barely detect PD-1$^+$ T cells (<1%), and there was a strong correlation between the proportion of anti-PD-1 and anti-IgG4 antibodies before and after in vitro culture of the aPD-1, for both CD8$^+$ and CD4$^+$ T cells, respectively (R$^2$ = 0.79 [CD8$^+$ T cells] and R$^2$ = 0.81 [CD4$^+$ T cells]).

## Intracellular cytokine staining assay

Cells were stimulated with 50 ng/ml PMA (Sigma-Aldrich, Saint Louis, MO, #P1585), 1 μg/ml ionomycin (Sigma-Aldrich, #IO634), and Golgi Plug reagent (BD Biosciences, #555029) at 37 °C under 5% $CO_2$ for 5 h.

Harvested cells were washed and stained for surface markers (mAbs specific for CD3, CD4, CD8, and CD45RA, and fixable viability dye). Cells were permeabilized with fix/perm solution (BD Biosciences) at 4 °C for 15 min, and stained with IFN-γ and TNF-α antibodies at 4 °C for 30 min. After washing, the cells were analyzed with an LSR Fortessa instrument.

## Cell preparation and sequencing of scRNA-seq

Mononuclear cells were extracted from samples collected. Cells were incubated with FcBlock and LiveDead for 15 min at room temperature, then a biotinylated anti-IgG4 antibody was added and incubated for 30 min at 4 °C. Streptavidin for CITE-Seq (BioLegend, Barcode sequence: AACCTTTGCCACTGC, #405261) was added and incubated at room temperature for 15 min. Anti-CD3, anti-CD45, and TotalSeqC antibodies (PD-1 [clone EH12.2H7; BioLegend, Barcode sequence: ACAGCGCCGTATTTA, #329963) were then added, incubated at 4 °C for 30 min, and cell sorting was performed by fluorescence-activated cell sorting. The sorted cells were loaded onto a Chromium Next GEM Chip G on a Chromium Controller (both from 10x Genomics) for barcoding and cDNA synthesis. Amplification of the cDNA and the library construction were performed using a Chromium Next GEM Single-Cell 5′ Kit v2, Chromium Single-Cell Human TCR Amplification Kit, and 5′ Feature Barcode Kit (all 10x Genomics) for 5′, VDJ, and antibody profiling according to the manufacturer's protocol. The libraries were sequenced on NovaSeq6000 (Illumina) and DNBSEQ (BGI).

## Single-cell immune profiling for gene expression and TCR repertoire

The sorted cells were mixed with reagents of the Chromium Next GEM Single-Cell 5′ Library & Gel Bead Kit v1.1 and loaded onto a Chromium Next GEM Chip G on a Chromium Controller (all 10x Genomics) for barcoding and cDNA synthesis. Amplification of the cDNA, targeted enrichment of TCR sequences, and library construction were performed using a Chromium Single-Cell V(D)J Enrichment Kit, Human T Cell and Chromium Single-Cell 5′ Library Construction Kit (both 10x Genomics) according to the manufacturer's protocol. The libraries were sequenced on a NovaSeq 6000 (Illumina).

## Bioinformatics analysis of scRNA-seq

Gene expression was quantified by Cell Ranger (v6.0.0) count with pre-built reference refdata-gex-GRCh38-2020-A downloaded from the 10x Genomics website. TCRs were called by Cell Ranger (v6.0.0) vdj with pre-built reference refdata-cellranger-vdj-GRCh38-alts-ensembl-4.0.0 downloaded from the 10x Genomics website. Quantified gene expression levels were preprocessed and visualized using Scanpy v1.9.1.[58] and Python 3.8.0. Demultiplexing was performed using HashSolo[56] with the default parameters. Inferred singlets with TCR chains were retained. The TRAV, TRBV, TRAJ, and TRBJ genes were removed for the clustering and embedding analysis in order to remove the effect by clonal expansion. The data were preprocessed by sc.pp.normalize_per_cell (counts_per_cell_after = 1e4), sc.pp.log1p, sc.pp.highly_variable_genes (min_mean = 0.0125, max_mean = 3, min_disp = 0.5), retaining highly variable genes. The inference of the cell cycle was performed using the sc.tl.score_genes_cell_cycle function following the tutorial (https://nbviewer.jupyter.org/github/theislab/scanpy_usage/blob/master/180209_cell_cycle/cell_cycle.ipynb).

Unwanted variations ('total_counts', 'pct_counts_mt', 'S_score', 'G2M_score') were regressed out using sc.pp.regress_out, and the expression was scaled using sc.tl.scale. Then, principal components were computed using sc.tl.pca. The batch effect of samples on principal components was removed by use of the Harmony algorithm[59]. Lastly, cells were embedded in two-dimensional space by uniform manifold approximation and projection (UMAP) using sc.tl.umap, clustered using sc.tl.leiden, and manually annotated using the following criteria. Pseudotime was calculated using sc.tl.dpt.

## Bioinformatics analysis of non-negative matrix factorization (NMF)

To extract gene programs from the single-cell data, we applied NMF following the tutorial (https://github.com/yyoshiaki/NMFprojection/blob/main/PBMC.ipynb). For each factor in NMF, we calculated the enriched pathways for the top 100 genes with the highest feature values using the enrichPathway function from the R package ReactomePA[60], and visualized them using the cnetplot and dotplot functions. NMF is extracted from the highly expressed variation by factor. As the correlation plots are influenced by the large variation, local features such as NMF4 can be underestimated. The gene expression data of tumors and clinical information for TCGA-STAD were downloaded using TCGAbiolinks (2.24.3). The NMF scores were calculated using NMFproj (0.0.1_220610, https://github.com/yyoshiaki/NMFprojection). A Cox proportional-hazards model was applied to infer the influences of the NMF features and the stage using lifelines[61] (0.27.4). For this analysis, the stage was defined as a number in the range 1–4. All values were standardized.

## Integration of published dataset with our dataset

To assess the effect of ICI, we integrated a published dataset (GSE173351) with our dataset using the Symphony algorithm[62] implemented in Symphonypy (https://github.com/potulabe/symphonypy). Cell abundances were statistically assessed using single-cell compositional data analysis (scCODA)[63] with the default parameters. In scCODA, Bayesian estimation is used to calculate the posterior probability of cell frequencies, and a qualitative assessment is made as to whether the difference is greater than the significance level.

## TCR analysis

TCR analysis was performed using Scirpy[64] (0.10.1). Clonotypes were defined using ir.tl.define_clonotypes (receptor_arms = "all", dual_ir = "any"). A same clone was defined as one in which the DNA sequence of CDR3 of VD and VDJ matched. For the assigned clones, STARTRAC[42] (0.1.0) analysis was performed to infer the expansion, migration, and transition status. For the clone network construction, we used the ir.tl.repertoire_overlap function to calculate the Jaccard index, which we then employed as a measure of clonotype similarity (Fig. 6c). There are clones in PBMCs that exhibit strong clonal expansion, which can disproportionately affect visualization. Therefore, to reduce their influence and to focus on clones present in the tumor, metastasis, and LN, the Jaccard index was calculated using only the clones found in those three compartments.

## Statistics

Differences between the two groups in each experiment were analyzed using the Student's t-test or Mann–Whitney U test, where appropriate. Categorical variables and continuous variables were compared as indicated in the figure captions. Non-parametric comparisons involving three or more groups were performed using the two-sided Kruskal–Wallis test followed by Dunn's post hoc test. Univariate and multivariate Cox regression analyses were performed. Hazard ratios (HRs) and 95% confidence intervals (CIs) were calculated for each factor. The variables with univariate regression $P$-value < 0.1 were included in multivariate regression analysis. Survival curves were calculated using the Kaplan–Meier method and differences were assessed using the log-rank test. The high and low groups were divided by the median values of each factor. A value of $P < 0.05$ was considered to be significant. All statistical analyses were performed using JMP Pro 17 Discovery (SAS Institute Inc., Cary, NC), R (4.3.0) and Python (3.9.16).

## Reporting summary

Further information on research design is available in the Nature Portfolio Reporting Summary linked to this article.

## Data availability

The raw sequence data for single-cell RNA/TCR/CITE-seq data generated in this study have been deposited at JGA via the NBDC human database (Accession Number: Study: JGAS000720, Dataset: JGAD000853) under controlled access (https://humandbs.dbcls.jp/en/hum0522-v1). Researchers can obtain access by submitting a data access request to the NBDC Data Access Committee (DAC) through the NBDC application system. The request requires submission of a research plan and institutional approval. After review and approval by the DAC, authorized users can download the data from the NBDC secure data archive. The analysis of single-cell RNA/TCR/CITE-seq data from five cancer patients have been deposited in the CELLxGENE database [https://cellxgene.cziscience.com/collections/0540ee09-5b45-43ec-9fc1-b9a7fa7f8f53; https://datasets.cellxgene.cziscience.com/2de07a23-f061-4582-9eae-2507edfd6cf7.h5ad]. All other data that support the findings of this study are available from the lead contact upon reasonable request. Source data are provided with this paper.

## Code availability

All source codes were deposited in the GitHub repository (https://github.com/yyoshiaki/2026_ICI_Nose_et_al) and have been archived at Zenodo with a DOI (https://doi.org/10.5281/zenodo.18436192).

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

## Acknowledgements

This study was supported by JSPS KAKENHI Grant Number 23K15472 (Y.N.), 20K17685 (T.S.), 23K08194 (T.S.), Takeda Science Foundation Medical research grant (T.S.), Chubei Itoh Foundation Medical research grant (T.S.). We created some figures with BioRender.com.

## Author contributions

Y.Nose, T.S., H.W., and Y.D. conceived the project. Y.Nose, Y.Y., T.S., E.S., N.O., H.W., and Y.D. designed the experimental framework. Y.Nose, Y.Y., T.S., Y.Nakamura, and E.S. performed the experiments. Y.Nose, K.J., K.F., A.U., and T.S. collected samples and obtained clinical data. Y.Nose., Y.Y., T.S., N.O., and H.W. analyzed data. K.M., K.Yamashita, K.T., K.Yamamoto, T.M., T.T., Y.K., N.O., S.S., H.W., H.E., and Y.D. supervised the research. Y.Nose, Y.Y., T.S., and N.O. wrote the manuscript and created the figures and tables. All authors approved the final version of the article, including the authorship list.

## Competing interests

The authors declare no competing interests. Yoshiaki Yasumizu, Yamami Nakamura, Naganari Ohkura, and Shimon Sakaguchi belong to the Department of Clinical Research in Tumor Immunology, Graduate School of Medicine, The University of Osaka, which is a joint research laboratory with Shionogi & Co., Ltd.
