## [Transparent Peer Review file · Nature Communications]

PD-1 antibody-bound progenitor-exhausted CD8+ T cells in lymph nodes boost PD-1-blockade anti-tumor immunity in gastrointestinal cancer

Corresponding Author: Dr Takuro Saito

Version 0:

Reviewer comments:

Reviewer #1

(Remarks to the Author)

This study aims to understand the dynamics of progenitor-exhausted T cells (Tpex) in cancer patients treated with immune checkpoint inhibitors (ICIs), addressing a fundamental question since Tpex are crucial for the effectiveness of ICIs. The key findings of the manuscript demonstrate that ICIs primarily target Tpex in lymph nodes, prompting their proliferation and migration to tumors. This has been suggested in recent studies in mice (https://www.science.org/doi/10.1126/sciimmunol.abg7836?url_ver=Z39.88-2003&rfr_id=ori:rid:crossref.org&rfr_dat=cr_pub) and human ([https://www.cell.com/cell/fulltext/S0092-8674\(23\)00164-2](https://www.cell.com/cell/fulltext/S0092-8674(23)00164-2)). However, overall, clarity needs enhancement, transitions should be smoother for better flow, and a more thorough analysis is necessary to provide additional context.

Major concerns:

While the paper reads nicely and is well written, it lacks the depth and detail required to provide the reader with a complete understanding of the experimental setup and the significance of the results. Additionally, the logic behind each cohort and experiment needs to be clarified, as well as the connections between the figures, to ensure the reader understands how each part of the study contributes to the overall findings.

The authors seek to answer how ICI-bound CD8 respond in comparison to non-ICI bound CD8, a really important question that will be critical for understanding how to better improve/target new therapies. Nonetheless, the authors provide inadequate details about the method. Furthermore, there is no validation within the paper that confirms this method performs in a specific manner. It would be nice to have (1) more controls that corroborates the effectiveness of the method and (2) explanation of ICI-non-bound CD8 – the implication is that these cells are exhausted and ICI therapy is just not working for them, it would help to determine whether these cells are tumor specific, obviously this is challenging not knowing the specific antigen but at least marker of exhaustion could be helpful (e.g. CD39, TOX, TCF7, PD-1, Tim3)?

The major finding of the paper concludes that C5 Tpex in the LN have higher expansion, migration, transition upon ICI treatment. Yet, we see in Figure 5e that C12 appear to have higher expansion, migration, transition but no ICI-bound CD8 T cells (according to text). That said, figure 5H shows high IgG4 clones and they appear to be located in cluster 12 Tpex, so it remains unclear how the authors claim with certainty that ICI treatment is expanding Tpex in LN and not meta-LN. Also, cluster 12 has most Tpex within the meta-LN or tumor, where one would expect Tpex to respond better. This reviewer feels that there is not evidence provided to determine if ICI truly expands Tpex within LN instead of meta-LN. This point needs further expansion. Do the authors have non-ICI treated patients they could compare cluster 12 to, to see whether the increased expansion, migration, transition pattern seen in cluster 12 Tpex is truly independent of ICI?

Minor concerns:

Figure 1: needs better explanation for cluster assignments. For example, authors assigned effectors based on (GrzmK)

expression, but in ext fig 1a – highest level of Grzmk was in Tpex. This should be explained in more depth and more markers should be added to the text.

Why did the authors not exclude doublet clusters from their analysis? This could compromise the interpretation of the results.

Ext data fig 1A: This representation is quite confusing and not intuitive. Also missing important markers like Tcf7

Fig2. it would significantly help if the authors provided more details about the bioinformatics analysis and for example why they choose non-negative matrix factorization (NMF) analysis for their analysis rather than saying Non-negative matrix factorization (NMF) is useful for profiling T cells with ambiguous boundaries... what does it mean ambiguous boundaries here ?

Figure 2C: why is NMF4 missing?

Figure 3: Include gating tree

Figure 4E-H: It would help significantly if we had number of samples used in 5E-H (especially for Tumor and Meta LN samples that look like there are only 2 samples there)

Figure 5C: what is 1, 2, 3? Is this TCR sequences that were found more than 3 times?

Figure 5G-H: explain better. Is this TCR sequences (alpha, beta chain), or is this amino acid sequence specifying similar binding properties? How do they define a clonotype – matching TCR sequence? How do they define clonotype matched to C5 cluster – T cells that share the exact same TCR sequence? Or similar amino acid sequence?

Authors may comment the result btw different figure for example why there is no significant of Tpex with better prognosis in gastric cancer (Fig2) and why there is better prognosis in gastric in fig 3f and J, this is because of they include LN...

Reviewer #2

(Remarks to the Author)

In this manuscript, the authors aim to characterize the anti-tumor immunity of progenitor exhausted T cells (CD8-Tpex) in gastrointestinal cancers. They also used anti-IgG4 antibody combined with epitopes sequencing (CITE-seq) with scRNA/TCR-seq (scRNA/TCR/CITE-seq) to study the ICI-bound T cells in tumors, lymph nodes (LNs), and blood from one ICI-treated gastric cancer patient. They showed that there was a clonal link between ICI-bound CD8-Tpex in LNs and intratumoral ICI-bound CD8-Tex.

Although the mechanistic analysis of ICI-bound T cells in ICI-treated cancer patients is interesting and important, I am confused by the flow of the manuscript. The manuscript starts by analyzing the T cells from tumors, metastatic and metastasis-free LNs, and blood from two esophagogastric junction (EGJ) cancer patients who had received chemotherapy and anti-PD-1 antibody therapy. They then projected published scRNAseq data of tumors from 15 ICI-treated lung cancer patients on to their EGJ dataset and claimed that a CD8-Tpex T cell subset from the LNs of the two EGJ patients is associated with better therapy response. They redefined the T cell subsets from the two EGJ patients using non-negative matrix factorization (NMF) and tested their association with survival in the TCGA dataset of gastric adenocarcinoma, finding no association between Tpex abundance and patient survival.

Subsequently, they analyzed the LN Tpex from 55 ICI-naive gastric cancer patients using flow cytometry and showed that the ratio of Tpex/Tex in metastasis-free LNs was associated with better survival outcomes. They then compared the LN Tpex from 12 ICI-treated gastrointestinal cancer patients with the ICI-naive gastric cancer patients and found that ICI treatment potentially increases the proliferative activity of the LN Tpex but not their cytotoxicity. Finally, they characterized the ICI-bound T cells in two(?) ICI-treated gastric cancer patient and found that ICI-bound CD8-Tpex had a higher cross-tissue migration potential and high clone overlaps with exhausted CD8 T cells (CD8-Tex) in the tumor.

Overall, the observations of the scRNAseq and flow cytometry data from the ICI-naive and ICI-treated gastric cancer patients are diverse and, many times, do not converge on concrete conclusions. Please find the specific comments below.

Comments:

1. As the authors noted, Figure 1b shows that all the CD3+ T cells sampled from various locations/organs exhibit significant organ specificity. However, the authors provided the same annotation for these cells from distinct clusters originating from different organs (Fig 1b). This is confusing and raises doubts about the correctness of the classification. For instance, the authors defined Tpex cells by the high expression of genes TCF7, PDCD1, and CXCL13. However, among the two annotated CD8-Tpex clusters (C05 and C12), C12 clearly lacks the expression of CXCL13.

2. The authors later mentioned that "The transcriptomic similarity of clusters was observed mainly according to tissue". Is this an actual difference (i.e. is it true that the naive T cell populations in LN and PBMC are very distinct)? The UMAP and the correlation analysis show that C2 and C10 are separated from C0 and C11 - I am not sure if this is true difference or there is some batch effect that was not accounted for here. Also, C05 and C08 are correlated in Fig 1E but they are much further in the UMAP. Why is that?

3. C09 is the population of CD4 T follicular helper which is known to express CXCR5, PD1, TOX, and BCL6 - as the authors mentioned later in Fig 1F. There is no indication that these are CD4 Tpex.
A part of C5 "Tpex" may have CD8+ CXCR5+ population reported previously (PMID: 31028278) - this population resembles the CD4 Tfh in their markers and these seems to be found in the LN.
4. The authors stated that "a significantly higher proportion of CD8-Tpex (C05) and CD8-effector (C08) in the MPR group than in the non-MPR group, indicating that abundant intratumoral CD8-Tpex and CD8-effector were associated with a favorable therapeutic effect of ICI (Fig. 1G)."
However, neither C05 nor C08 were tumor-infiltrating T cells in Fig 1B dataset: C05 were found mainly in lymph nodes (LNs) and C08 in blood.
Are the projected T cells from the lymph nodes of the lung cancer patients (in GSE173351)?
If they are not from the LN but from the tumor, how did the authors correlate these non-tumor T cell clusters (C5 and C8) to the tumor-infiltrating T cells from lung cancers and draw conclusions on their prognostic power? Furthermore, in Figure 1g, no statistical significance of the comparison was mentioned. What is the meaning of "Credible"?
5. What is the data supporting this statement (page 5, line 154-155) "CD4-Tpex cells have acquired migratory and proliferative functions in LNs, possibly working on CD8-Tpex cells, while still maintaining progenitor cell characteristics"?
The reviewer still thinks that the CD4 Tpex is a population of Tfh that may or may not be related to the CD4 Tex. Did the author check if the TCR of CD4 Tpex and CD4 Tex highly overlap?
6. The NMF analysis of their scRNAseq dataset (Fig 2) appears inconsistent with their clustering analysis (Fig 1b) and the similarity analysis of these clusters (Fig 1e). For example, NMF4 includes C0, C11, C1, C2, and C10 (Fig 2a, Fig 1b). C0 and C11 are from lymph nodes (LNs), which do not share high similarity with C1, C2, and C10, which are from blood (Fig 1e). This raises doubts about the robustness of the NMF analysis or the initial clustering analysis in Fig 1.
7. Furthermore, it is unclear how the biological processes represented by NMF1 in Fig 2c are relevant to Tpex cell state and ICI response.
8. The absence of any significant correlation between the NMF1 or NMF8 gene program enrichment in TCGA does not imply that the Tpex accumulation is not important in the prognosis of ICI-naive gastric cancer. The authors has not established that the enrichment of NMF1 or NMF8 onto a bulk RNAseq dataset of TCGA is specific to the T cell and not other cell types in the tumor.
9. On Page 5, lines 184-188, why did the authors think that Extended Figures 5a and 5b support their hypothesis that "anti-tumor immunity was induced in LNs close to the tumor"? Extended Figure 5 compares the frequency of IFN- γ + and TNF- α + CD8 T cells among CD8+PD1+ vs. CD8+PD1- cells across pathological tumor stages. It was expected that CD8+PD1+ T cells would have higher IFN- γ and TNF- α production than CD8+PD1- T cells, since PD1 itself is a T cell activation marker. However, I do not understand how this data is related to Figure 3b and the authors' claim that "anti-tumor immunity was induced in LNs close to the tumor." The authors should show that IFN γ and TNF expression data in the same grouping as Fig 3b and not by pathological tumor stage.
10. "The frequency of CD8-Tpex in metastasis-free LNs was lower in advanced tumor stages (Fig. 3d)." Is this analysis only done in the proximal metastasis free LN?
11. "The expression frequency of molecules among CD4 + T cells in each tissue was similar to that of CD8 + T cells (Fig. 3g)." The reviewer is unsure of the relevance of Tim3 and CD103 in CD4 T helper populations. Furthermore, Fig 3i also show an increase in GZMB expressing CD4 Tex - what is this population?
12. In Figure 3d, metastasis-free LNs from Stage III seem to have a lower Tpex/Tex ratio compared to Stages I and II. I suggest that the authors perform a multivariate Cox hazard analysis (as they did in Figure 2d) to determine if Tpex/Tex is an independent prognostic marker or if it is just reflecting tumor stages (the ratio correlates with stages in Fig 3d and 3h). Additionally, did the authors test the survival association of Tpex, Tex frequency, or the Tpex/Tex ratio in tumors and metastatic LNs?
13. "On the other hand, cytotoxicity evaluated by GZMB was not significantly different between each T-cell subset in patients with ICI-naive and ICI-treated cancers (Fig. 4f)" - there seems to be one statistically significant difference in Fig 4f. More puzzlingly, there is a uniform, statistically significant upregulation of GZMB across all CD4+ T cell groups in mets free LNs in Fig 4h but the authors stated: "These results were consistent with CD4 + T cells (Fig. 4g,h). Thus, ICI treatment does not affect the cytotoxicity of Tpex in metastasis-free LNs, but greatly increases their proliferative potential".
14. Is it correct that both CD4 Tpex (of Tfh) and Tex are not clonally expanding as shown in Fig 5c? The authors have been drawing parallel conclusions across the CD8 and CD4 Tpex and Tex populations but most of the data, including this TCR data, suggest that they are different.
15. The reviewer noted that most of the analysis in Fig 5 is based on only two patients, which limits the ability to generalize any conclusions. Additionally, the authors should provide some supporting data (maybe as supplementary) that the clonal expansion, migration and transition across the organs and/or T cell clusters are consistent between the two patients to ensure we are not looking at a pattern dominated by a single patient.

16. "A high cross-tissue migration score was observed in CD8-Tpex (C05, C12) and CD8-effector (C08) cells" - are the T cells in C08 clonally related to C05 and C12? What is the authors opinion on the sequence of the migration of tumor infiltrating Tex T cell clones? Is it starting in the proximal LN?
In line 263, the authors stated "Regarding migration, the most observed route of migration was between LN and tumor (C04, C05, C06.0, C08) (Fig. 5f)." It is noted that C08 is from PBMC and not from LN. The reviewer is also curious that the PBMC C08 clones are not discussed as much as LN Tpex C05 despite a strong clonal overlap with tumors and a high expansion, migration and transition indices.
17. The Tex clusters have high transition scores and the authors stated "intratumoral CD8- Tex cells had a high potential to differentiate and expand" - To the reviewer's understanding, the Tex are exhausted T cells that are not undergoing further state transition.
18. "All of these T cells were bound by ICIs except for C12, suggesting that ICI-bound T cells play a leading role after ICI administration." I am not sure if this is what I observe from Fig 5a - there are some IgG4 signal in C12 and the mRNA expression in C12 is similar to C5 (Fig 5b). This seems to be a rather incoherent description of the data. The implication "suggesting that ICI-bound T cells play a leading role after ICI administration" is even more confusing. What is the basis of this statement?
19. "Although some cells in the C05 migrated between LNs and PBMCs, most cells migrated between LNs and tumor, and clonotypes within this cluster were highly shared with the tumor-enriched CD8- Tex clusters (C04, C06.0) (Fig. 5f,g)." This qualitative approach comparing "some cells" with "most cells" without quantification is concerning.
20. "Importantly, clonotypes with high IgG4 binding (ICI high-bound T cells) shared clones with the tumor-enriched CD8- Tex clusters (C04, C06.0), whereas clonotypes with low IgG4 binding (ICI low-bound T cells) shared clones with the PBMC-enriched CD8-effector (C08) (Fig. 5h), indicating the possibility that ICI high-bound Tpex in LNs contains cancer antigen-specific T cells, whereas ICI low-bound Tpex in LNs may have a different antigen specificity."
This statement needs to be supported by a quantification of clone overlap between IgG4 high clones in C05 with the tumor (and mets) compared to the overlap with C8 PBMC clones.
21. Based on Fig 5f, C08 has a strong pairwise migration index from PBMC to the tumor (almost as strong as C05 in LN - Tumor comparison). Can the authors generate the same analysis focusing the clones in C08 and separating them by IgG4 high vs. low? This is important to confirm the source of Tex in the tumor (whether they all come from Tpex in LN or not). Either way these observations are important to be reported.
22. "We also discovered that PDCD1+ TCF7+ Tpex presence within tumors correlates with the successful outcome of ICI treatment. Interestingly, this correlation did not extend to the prognosis of ICI-naive gastric cancers. In contrast, abundant Tpex presence in LNs was linked to favorable prognosis in ICI-naive gastric cancer, and the proliferation activity of Tpex in LNs significantly increased after ICI treatment."
I am not sure the projection of lung tumors on the scRNAseq of gastric cancer can reliably define that the PDCD1+ TCF7+ T cell in the TIL is the same as the Tpex in the LN. This convoluted sentence is probably unnecessary and is confusing to the reader. It is adequate to state the positive correlation in gastrointestinal cancers.
23. In line 339, the authors stated "In our data, abundant intratumoral Tpex were associated with better response to ICI treatment". Is this pointing to the lung cancer data? It is confusing. I really suggest the authors to remove the lung cancer data analysis altogether (GSE173351) - it is hard to see why it is related to this study.

Reviewer #3

(Remarks to the Author)

The study conducted by Takuro Saito's team investigates the impact of therapeutic PD-1 antibodies on TCF1 and PD1-expressing progenitor exhausted CD8 T cells (Tpex) by analyzing tumor tissues, peripheral blood, and lymph node samples from gastrointestinal cancer patients both before and after ICI treatment. The authors found that Tpex cells are enriched in the proximal lymph nodes of non-metastatic tumors and then proliferate and migrate to the tumor site post-ICI treatment, possibly differentiating into exhausted CD8 T cells (Tex) to exert anti-tumor effects. However, I have several major concerns listed as below.

1. The authors mention that several studies have already demonstrated the association of TCF1 and PD1-expressing Tpex in tumors and peripheral blood with the response and prognosis of cancer patients to ICI treatment. A study published in Cell 2023 titled "Dynamic CD8+ T cell responses to cancer immunotherapy in human regional lymph nodes are disrupted in metastatic lymph nodes" also reported a high presence of TCF1+PD1+Tpex in the uninvolved lymph nodes of head and neck squamous cell carcinoma patients, and verified the clonal correlation between these Tpex and Tex in the tumor microenvironment. This study already provided a potential explanation for the anti-tumor mechanism of Tpex post-ICI treatment. Additionally, in another Cancer Cell paper published in 2023 entitled "Progenitor-like exhausted SPRY1+CD8+ T cells potentiate responsiveness to neoadjuvant PD-1 blockade in esophageal squamous cell carcinoma." revealed that the abundance of SPRY1+ CD8+ Tpex in the digestive tract tumor microenvironment can serve as a marker for patient treatment response and prognosis to immunotherapy. Given the existing research discussing the presence and functions of Tpex in improving ICI treatment, the novelty and translational value of this study are limited.

2. The authors performed multi-sample single-cell sequencing on two cancer patients who underwent chemotherapy combined with anti-PD-1 treatment, and multi-sample flow cytometry analysis on 12 gastrointestinal tumor patients who received ICI treatment. However, a larger number of samples from more patients is required to provide more convincing evidence.

3. The authors utilized CITE-seq, TCR sequencing, and TCR tracking algorithms (STARTRAC) to analyze the clonal correlation between lymph node CD8+ Tpex and tumor CD8+ Tex, and their potential migration pathways at the single-cell level. However, showing the analysis of bioinformatics is not enough and indirect. This study lacks dynamic monitoring of lymph node Tpex migration in animal models, failing to clearly elucidate the migration and differentiation process of Tpex. Therefore, the study does not provide strong evidence for the hypothesis that lymph node Tpex migrate to the tumor and differentiate into Tex.

4. The authors analyzed the relationship between the Tpex/Tex ratio in the lymph nodes of 12 patients without ICI treatment and prognosis, finding that a high Tpex/Tex ratio correlates with better prognosis. This finding suggests that Tpex could serve as a prognostic predictor and a novel cell therapy approach, holding significant clinical translational value. However, the authors need to conduct validation in large external cohorts to enhance the reliability of this finding.

Version 1:

Reviewer comments:

Reviewer #1

(Remarks to the Author)

The authors have completed a thorough revisions and have provided a detailed response to my comments. This has significantly improved the manuscript. There are just a few minor points I would suggest adding:

Tpex → Tex transition: The pseudotime and clonal data are of high quality, but lack functional validation. Acknowledging this point as a limitation in the discussion would be beneficial.

TOX in CD4-Tpex: These cells display both TOX and progenitor markers such as TCF7 and IL7R. It might be useful to add a line in the discussion addressing the potential plasticity of these cells, considering their dual marker expression.

Reviewer #2

(Remarks to the Author)

The revised manuscript by Nose et al is not addressing most of the concern of the reviewer. In fact, Supp Fig 1 shows how complicated the flow of this paper is. Fig 1 start with scRNAseq analysis of T cells in blood, LN and tumor of ICI treated EGJ cancer and the author applied the NMF-derived PBMC and blood gene signatures on ICI naive TCGA gastric cancer (why?) and say that these signatures does not predict survival (can it be the problem of the signature?), then they show using flow cytometry that the Tpex/Tex ratio in LN (a value not related to the NMF signatures) is associated with RFS and OS. Then, the authors show that they can identify anti-PD1-bound T cells using anti-IgG4 (the isotype of the therapeutic anti-PD1) and suggest that the anti-PD-1 bound T cells in the LN is more proliferative. Then, they went back into their original scRNAseq analysis and added additional samples with scTCRseq and CITEseq (1 ICI-treated and 2 ICI naive) and this time added the layer of scTCRseq and CITEseq analysis. The finding was that anti-PD-1 bound CD8 Tpex cells are clonally expanding and share the same clones as the Tex in the tumor.

Pai et al (ref 25) and others have reported the clonal linkage between Tpex in tumor draining LN and the intratumoral CD8 T cell. The novel result that the Authors reported here is the analysis of the anti-PD-1 bound T cells. The Authors show the anti-PD-1 bound T cells seem to be proliferating more in the LN and have mostly exhausted phenotype in the tumor. Thus, instead of first showing scRNAseq analysis of 2 ICI-treated patients, the authors should directly start with analysis of all 5 scRNAseq/scTCRseq/CITEseq samples (3 ICI treated and 2 ICI naive). The Authros should present the phenotype(s) of the anti-PD-1-bound T cells and how they are different from the similar T cells from ICI naive patients. The Authors should separate analyses on intratumoral T cells vs T cells in LN vs. T cells in PBMC unless they can be integrated into unified phenotypes across organs. The Authors highlighted the patterns of T cell clone sharing in the anti-PD-1 bound T cells vs non-bound T cells (in ICI treated patients) vs. T cells from ICI-naive patients across the tumor, LN and PBMC, which is novel and important. Is there any clonal sharing between TFH in LN with the CD4 in the tumor?

Any intratumoral anti-PD-1 bound T cell population identified in the ICI treated gastric tumor/mets samples can be compared to those found in lung tumor/mets sample from GSE173351 to see if there are any similarity. T cell populations from the LN of gastric tumor patients should be compared with those from LN of the lung tumor patients in the GSE173351 dataset (instead of what is done now where the authors are mixing the clusters from blood, LN and tumor and compare them against a mixture of T cells from lung tumor, adjacent normal lung, and LN).

The flow cytometry analysis should be focused on that in Figure 4 where the authors compare the effect of ICI in the Tpex and other T cell populations. The effect of anti-PD-1 treatment in increasing the proliferation of TCF1+ PD1+ Tpex in LN. The Authors should demonstrate if the fraction of Ki67+ TCF1+ PD1+ Tpex in LN is associated with survival in patients with gastric cancer. The importance of anti-PD-1 bound TFH in LN should also be supported with a similar analysis. Finally, the Reviewer could not see any significant implication or conclusion from the TCGA analysis.

The Reviewer also has other specific concerns with the analysis as listed below.

Specific comments:

1. The single cell RNAseq analysis in Fig 1b shows a strong organ-driven batch effect (not patient driven batch effect). The Authors should really try to integrate the data using data integration method like CCA, RPCA, bbknn etc and observe whether such organ effect can be mitigated. Looking at the method description of the scRNAseq analysis in line 579-580 of the main text, the Reviewer noticed that the Authors did not integrate the profiles from different samples (different organs and different patients).

For an example of multi sample integration using scanpy, please see <https://scanpy-tutorials.readthedocs.io/en/latest/integrating-data-using-ingest.html> or https://nbisweden.github.io/workshop-archive/workshop-scRNAseq/2020-01-27/labs/compiled/scanpy/scanpy_03_integration.html.

For instance, the integration may be able to cluster together the naive T cell populations in the LN (C0 and C11) and PBMC (C2 and C10).

2. The scRNAseq analysis should be done on all 5 scRNAseq samples right from the start rather than analyzing two sample first (pt 1 and 2) and projecting the scRNAseq of Pt 3,4,5 on to the clustering of Pt 1 and 2 using symphony.

3. If the authors still find separate CD4 and CD8 T cell populations after integration step, then the Authors must ensure that analysis of different T cell populations are grouped by organ. For instance, to analyze the correlation between intratumoral T cell populations in TCGA gastric cancer dataset, the Authors should only test gene signatures representing intratumoral T cell populations (e.g. C4, C6.0, C6.1, C7, C12, C14, C17.0, C17.1, C19 - based on Fig 1d). If the authors must do NMF analysis, then please only test NMF signatures that included intratumoral T cells (NMF0,3,5,6,7). NMF8 is clearly not strongly related to intratumoral T cells clusters (see Fig 1d and Fig 2b).

4. (related to response to Reviewer 1 minor comment #5, Reviewer 2 comment #6 #7 #8) The gene signatures of each NMF group need not be enriching for some reactome gene set (reactome is a pathway database with groups of proteins in a pathway, not a gene signature database with list of genes changing in expression in response to perturbation like the MsigDB). NMF0 clearly enriches for Treg markers (Fig 2b). NMF4 enriches naive or central memory T cell genes (Fig 2b). Looking at the genes in NMF1 and NMF8 (Fig 2b), there is no indication that they are related to CD8 T_{pex}. Please provide paper reference(s) showing that SLC2A3 (GLUT2), NR4A2, GZMK, FGFBP2, CST7 and interferon signaling are in any way related to T_{pex}. Also, are TCF7, SELL, TOX, PDCD1 included in the gene signatures of NMF1 and 8?

5. (related to response to Reviewer 2 comment #4) In Suppl Fig 4, the Authors tests if any of the T cell clusters in Fig 1b is enriched in the T cells in a neoadjuvant PD-1 inhibitor-treated lung cancer patients. The lung cancer patient-derived T cells are coming from lung tumor/mets or adjacent normal lung or the LN (this is described in the Figure caption). The authors are lumping all of them together into their existing clustering, which is also a mixture of T cells from tumor, LN and PBMC of gastric cancer patients. The Authors then reported that higher proportion of T cells in C5 (T_{pex} in LN) and C8 (T_{eff} in PBMC) is associated with major pathological response (MPR) in the lung cancer patients receiving neoadjuv anti-PD1. Are the T cells projected to C5 from the lung cancer dataset from the LN T cells (there are 3 patients with T cells from LN in the dataset)? Are the T cells projected to C8 from the lung cancer dataset from the blood T cells (there is only 1 patient with T cells from blood)?

In their response, the Authors stated that "there is currently no scRNA-seq dataset that directly evaluates T_{pex} in LNs after ICI treatment in human samples and examines its association with ICI treatment efficacy or prognosis." This does not justify checking the T_{pex} cluster in LN across T cells from multiple organs. What the Authors can do is to test if any of the intratumoral T cell population in Fig 1b is enriched in the intratumoral T cell population of lung cancer patients with MPR after neoadjuvant anti-PD1.

6. (related to response to Reviewer 2 comment #3 and #5) The whole discussion of "CD4 T_{pex}" in this paper totally ignores the known fact that LNs enriches for CD4 T_{fh} and that C09 matches the phenotype of T_{fh} very clearly (including the expression of TCF7, TOX, TOX2, PDCD1, TIGIT, CXCL13, CXCR5, BCL6 shown in Supp Fig 5) and that C09 is the ONLY cluster matching CD4 T_{fh} in this whole dataset. So C09 is T_{fh} and there is no need to invent a new CD4 T_{pex} population here. What is NOT known is whether aPD-1 binds to T_{fh} in the tumor draining LN (answer: yes based on Fig 5a), whether they are activated by aPD-1 (yes, their proliferation is induced in Fig 4g) and whether they migrate into the tumor (answer: probably no based on Fig 5d,e,g; more specific analysis is needed).

7. Analysis in Fig 3 is to support the importance of CD8 T_{pex} and T_{fh} fraction in proximal (presumably tumor draining) LNs. The Authors used a ratio between T_{pex} and T_{ex} to correlate with relapse free or overall survival. Does the fraction of T_{pex} in the LN not correlated with RFS/OS? Also, the difference in p-value significance between CD8 T_{pex}/T_{ex} ratio in LN vs. CD8 T_{pex}/T_{ex} ratio in tumor may simply reflect the difference in the sample numbers (LN n=49, tumor n=23) - the KM plots look

quite comparable between the two.

8. (related to response to Reviewer 2 comment #9) "First, The comparison between ICI-bound and ICI-non-bound CD8+ T cells revealed that ICI-bound CD8+ T cells exhibited higher CD103 expression and lower TCF1 expression than did ICI-non-bound CD8+ T cells across all tissues, suggesting that ICI-bound CD8+ T cells possess an exhausted phenotype, whereas ICI-non-bound CD8+ T cells retain a naive phenotype (Supplementary Fig. 11a, b)." CD103 is not a known marker of exhaustion - it is reported as a marker of tissue resident T cells that can also be expressed on intratumoral T cells (see for example <https://rupress.org/jem/article/218/4/e20201605/211911/Tissue-resident-memory-T-cells-in-tumor-immunity>). The authors should include citation of supporting papers defining CD103 as marker of exhaustion, especially for Tpex in the LN?

9. (Fig 4e-h) are the samples from ICI-treated T cells stained for PD-1 using the indirect method (i.e. using anti-IgG4) while the ICI-naive stained using direct method (i.e. using anti-PD1)? If yes, cells from the same T cell subsets from the same organ in the ICI-treated and ICI-naive may not be comparable. This is because the antibodies used to define the subsets are not exactly the same and hence may bind the PD1+ T cells at different rates. Indeed, Supp Fig 10 shows that although the correlation between PD-1 staining and IgG4 staining is highly positive, the slope of the fitted linear curve is not equal to 1 (the %IgG4+ is smaller than %PD-1+).

What the authors can do is comparing the relative abundance of TCF1+ within PD1+ subset (for ICI naive) or IgG4+ subset (for ICI-treated).

10. The STARTRAC analysis in the main Figure 5 should highlight the differences between the ICI-naive vs. ICI-treated patients (now Supp Fig 14). The Authors should highlight the subsets with differences. The important observations include ICI increases expansion in Tpex (C5 and C12) and increased migration from LN to tumor only in C5 (which may make sense since the Tpex in C12 is already in the tumor). It is important to highlight that with ICI, the transition of intratumor Tex and Tpex are lower -- does this mean they slow their transition to terminally exhausted phenotype? These are important points to highlight. Finally, the STARTRAC analysis should be performed only on the IgG4+ subset of the ICI-treated samples to show the patterns of expansion, migration and transition in the anti-PD-1 bound T cells.

To aid the visualization of the TCR overlap between different pairs of clusters, the author should visualize the percentage overlap in a heatmap format rather than listing them in the text. This should be done in ICI treated vs ICI naive (and perhaps in IgG4+ T cell subset).

Reviewer #3

(Remarks to the Author)

After carefully evaluating the author's revised manuscript and supplementary data in response to the review comments, I find that the author has provided detailed and substantive responses to the key concerns, including research innovation, sample size, limitations in animal model validation and prognostic analysis. While certain constraints remain due to practical limitations, the author has made notable improvements to the paper, addressing several critical issues and enhancing its overall rigor.

The authors have integrated matched tumor, lymph node, and blood samples from patients after ICI treatment and utilized scRNA/TCR/CITE-seq technologies to distinguish ICI-bound and non-bound T cells, demonstrating a certain level of technical innovation. Furthermore, the study focuses on the activation of Tpex cells in non-metastatic LNs, their migration into tumors, and their association with prognosis, providing unique insights into the dynamic migration mechanism of Tpex cells.

In addition, the author has expanded the sample size and supplemented single-cell sequencing data, enhancing the reliability of the results. Although the study lacks dynamic monitoring of lymph node Tpex migration in animal models due to practical constraints, the author provides evidence of Tpex migration to some extent through multi-omics data and clonal correlation analysis, while explicitly acknowledging this limitation in the discussion, making the response scientifically sound and reasonable.

Finally, validating the prognostic value of the Tpex/Tex ratio in a larger cohort is of significant reference value. However, the author indicates that the difficulty in obtaining surgical samples from ICI-treated patients limits further validation of the correlation between the Tpex/Tex ratio and prognosis in a larger cohort, which is indeed an objective limitation.

In conclusion, this study systematically reveals the dynamic role of lymph node Tpex in ICI treatment, providing a new perspective for optimizing immunotherapy strategies. I recommend accepting the manuscript.

Version 2:

Reviewer comments:

Reviewer #2

(Remarks to the Author)

The revised manuscript of Nose et al has included some important updates.

The manuscript keeps most of the original figures intact and most of the updates is added into new supplementary figures. The new manuscript feels heavy and a little convoluted to my preference but I appreciate the efforts taken to address my concerns.

First, the Authors tested several methods of integration and demonstrated that T cells from different organs are different transcriptomically from one another (Response to Reviewer 2 comment #1). However, I noted that the integration (i.e. batch effect removal) is based on removing the effect of different donors. Indeed, when the batch effect removal is performed based on donors+site, much of the site specific differences disappear.

I agree that the Authors have freedom to choose to analyze the T cells without removing the organ-specific differences. However, if that is the Authors' choice, then when the Authors projected GSE173351 single cell data of lung cancer to their UMAP in Fig 1B, they have to check if the organ of origins of the mapped GSE173351 T cells should match the organ site of the Authors' own clusters.

For example, GSE173351's "CD8-effectors (and also CD8 TRMs)" seem to match with lymph node-enriched C05 (CD8-Tpex). Are these CD8 T effector/TRM cells from the lymph node in GSE173351? If yes - it is useful to highlight this. If the CD8 T effector/TRM cells in GSE173351 are mostly intratumoral, the Authors should compare them with C12, which is the intratumoral Tpex in Figure 1B and 1E.

Second, the discussion of CD4 Tfh is redundant (page 6). They are enriched in LNs and have well known function as B cell helper and have well defined markers that are different from naive CD4 T or CD4 Tcm. The fact that very little CD4 clones in the tumor (or PBMC) have TCR overlap with these Tfh suggest that they are NOT migratory. The Authors should consider removing this part to avoid confusion.

Third, following the same vein, the discussion on CD4 Tfh/Tex ratio in Fig 3g,h,i,j may not be relevant. The CD4 Tfh and CD4 Tex populations are not connected clonally and it is not clear why the ratio between them matters. Also, the log-rank p-values are all > 0.1. I also noted that Fig 3J still uses the "Tpex" label for the Tfh.

Responses to the reviewers' comments

Reviewer #1 :

This study aims to understand the dynamics of progenitor-exhausted T cells (Tpex) in cancer patients treated with immune checkpoint inhibitors (ICIs), addressing a fundamental question since Tpex are crucial for the effectiveness of ICIs. The key findings of the manuscript demonstrate that ICIs primarily target Tpex in lymph nodes, prompting their proliferation and migration to tumors. This has been suggested in recent studies in mice (https://www.science.org/doi/10.1126/sciimmunol.abg7836?url_ver=Z39.88-2003&rfr_id=ori:rid:crossref.org&rfr_dat=cr_pub 0pubmed) and human ([https://www.cell.com/cell/fulltext/S0092-8674\(23\)00164-2](https://www.cell.com/cell/fulltext/S0092-8674(23)00164-2)). However, overall, clarity needs enhancement, transitions should be smoother for better flow, and a more thorough analysis is necessary to provide additional context.

We thank the reviewer for constructive comments on our manuscript. The statement by the reviewer that “the paper reads nicely and is well written” was highly encouraging to us. We have responded to each of the comments one by one.

Major concerns:

1) While the paper reads nicely and is well written, it lacks the depth and detail required to provide the reader with a complete understanding of the experimental setup and the significance of the results. Additionally, the logic behind each cohort and experiment needs to be clarified, as well as the connections between the figures, to ensure the reader understands how each part of the study contributes to the overall findings.

Thank you for the thoughtful comment. According to the reviewer's suggestion, we have created an overview summarizing the samples and experimental methods for each figure to clarify the overall connections in the study (Supplementary Fig. 1). Additionally, we have added some introductory text at the beginning of each section in the results to enhance readability (page 4 lines 87-88, page 5 lines 128-131, page 5 line 147, page 6 lines 172-176, page 7 lines 197-198, page 8 lines 237-238, page 9 lines 289-290).

Furthermore, we have added a detailed description of the experimental setup, including the IgG4 detecting method in flow cytometry (page 16 lines 520-534, Supplementary Fig. 10) and single-cell RNA/TCR/CITE sequencing (pages 9-10 lines 294-297, Supplementary Fig. 12), and NMF analysis (page 18 lines 584-596). We expect that these revisions will contribute to improving the readability of the manuscript.

2) The authors seek to answer how ICI-bound CD8 respond in comparison to non-ICI bound CD8, a really important question that will be critical for understanding how to better improve/target new therapies. Nonetheless, the authors provide inadequate details about the method. Furthermore, there is no validation within the paper that confirms this method performs in a specific manner. It would be nice to have (1) more controls that corroborates the effectiveness of the method and (2) explanation of ICI-non-bound CD8 – the implication is that these cells are exhausted and ICI therapy is just not working for them, it would help to determine whether these cells are tumor specific, obviously this is challenging not knowing the specific antigen but at least marker of exhaustion could be helpful (e.g. CD39, TOX, TCF7, PD-1, Tim3)?

Appreciating the reviewer's comments to improve our manuscript, we have added the methodological details shown in the query 1.

In response to the reviewer's suggestion (1), we have performed experiments to
 validate the IgG4 methods with both flow cytometry and scRNA-seq analysis. A flow
 cytometry method was validated to detect ICI-bound T cells using an anti-IgG4 antibody
 with PBMCs from 29 newly added ICI-naive gastric cancer patients. We compared the
 positivity of anti-PD-1 and anti-IgG4 antibodies for PD-1⁺ T cells before and after in vitro
 culture with anti-PD-1 therapeutic antibodies (Supplementary Fig. 10). The results show
 that the anti-PD-1 antibody positivity before culture was consistent with the anti-IgG4
 antibody positivity after culture, clearly indicating that the IgG4 method with flow cytometry
 detected ICI-bound T cells (page 8 lines 239-244).

Furthermore, we additionally performed scRNA/TCR/CITE-seq using anti-IgG4 and
 anti-PD-1 antibodies in one new ICI-treated and two ICI-naive cancer patients to confirm
 the ability of the IgG4 methods to detect ICI-bound T cells in single-cell sequencing
 analysis (Supplementary Fig. 12b,c, 14). The anti-IgG4 antibody was detected in ICI-
 treated patients (Supplementary Fig. 12b-c right), whereas it was barely detected in ICI-
 naive patients (Supplementary Fig. 12b-c left). These experiments further validated the
 IgG4 method in scRNA-seq analysis. We have added the methodological description in the
 methods section (page 15 lines 477-484, page 16-17 lines 542-554) and described these
 results in the manuscript (page 9-10 lines 291-300).

In response to the reviewer's suggestion (2), regarding the characteristics of ICI-
 non-bound CD8⁺ T cells, we analyzed scRNA-seq data from tumor samples of three ICI-
 treated cancer patients to compare gene expression between ICI-bound and ICI-non-bound
 CD8⁺ T cells. Characteristic expressed genes of ICI-non-bound CD8⁺ T cells included *IL7R*,
 *TCF7*, and *LTB*, whereas ICI-bound CD8⁺ T cells expressed immune checkpoint-related
 genes such as *CXCL13*, *PDCD1*, *HAVCR2* (Tim-3), *TIGIT*, and *LAG3*, indicating that ICI-
 bound CD8⁺ T cells exhibit an exhausted phenotype while ICI-non-bound CD8⁺ T cells
 show a naive phenotype (Supplementary Fig. 13). Flow cytometry analysis further
 confirmed these findings, showing lower CD103 and higher TCF1 expression in ICI-non-
 bound CD8⁺ T cells compared with ICI-bound CD8⁺ T cells across all tissues
 (Supplementary Fig. 11a, b). These results collectively demonstrate that ICI-non-bound
 CD8⁺ T cells have a naive phenotype. We have described these results in the Results
 section (page 9 lines 264-269, page 10 lines 305-310).

**Figure legend: The frequency of CD103 (left) and TCF1 (right) among ICI-bound and**
 **ICI-non-bound CD8⁺ T cells in each tissue of ICI-treated cancer patients by flow**
 **cytometry.** The frequency of CD103 and TCF1 expression among ICI-bound and ICI-non-
 bound CD8⁺ T cells was shown in each tissue. Error bars indicate the median ±
 interquartile range. The significance of the differences between ICI-bound CD8 and ICI-
 non-bound CD8 in the same tissue was calculated using the nonparametric Wilcoxon
 matched-pairs signed-rank test (*P < 0.05, **P < 0.01, ***P < 0.001). Tumor: n=13, meta-
 LN: n=6, meta-free LN: n=55, PBMC: n=12.

**3) The major finding of the paper concludes that C5 T_{pex} in the LN have higher**
**expansion, migration, transition upon ICI treatment. Yet, we see in Figure 5e that C12**
**appear to have higher expansion, migration, transition but no ICI-bound CD8 T cells**
**(according to text). That said, figure 5H shows high IgG4 clones and they appear to**
**be located in cluster 12 T_{pex}, so it remains unclear how the authors claim with**
**certainty that ICI treatment is expanding T_{pex} in LN and not meta-LN. Also, cluster**
**12 has most T_{pex} within the meta-LN or tumor, where one would expect T_{pex} to**
**respond better. This reviewer feels that there is not evidence provided to determine**
**if ICI truly expands T_{pex} within LN instead of meta-LN. This point needs further**
**expansion. Do the authors have non-ICI treated patients they could compare cluster**
**12 to, to see whether the increased expansion, migration, transition pattern seen in**
**cluster 12 T_{pex} is truly independent of ICI?**

Thank you for the important comment. Since the cutoff value for IgG4-high or low in
the scRNA/TCR/CITE-seq analysis was not predetermined, we set it as the median of
1.079 (Fig. 5b, Supplementary Fig. 12b bottom), defining clusters with 50% or more IgG4-
positive cells as IgG4-high clusters. On the basis of this threshold, C12 (73.4%) was
classified as an IgG4-high cluster (page 10 lines 298-300)(Fig. 5b, Supplementary Fig.
12c). Accordingly, Fig. 5b was revised to reflect the updated IgG4-high clusters, and the
phrase “except for C12” was removed from the manuscript (page 10 line 329). As for
annotations of Fig. 1d, e, “Meta” indicates liver metastases and “LN” indicates metastatic
and metastasis-free LNs, as described in the legend of Fig. 1d, e. Therefore, CD8-T_{pex}
(C12) are derived from liver metastases and tumors, not metastatic LNs.

Appreciating the reviewer's suggestion, we have added two ICI-naive cancer
patients along with one new ICI-treated cancer patient for RNA/TCR/CITE-seq analysis. In
total, we performed single-cell RNA/TCR/CITE-seq in two ICI-naive cancer patients and
three ICI-treated cancer patients (Supplementary Fig. 12, 14). The STARTRAC algorithm
was used to compare clonal expansion between ICI-treated and ICI-naive samples. The
results showed that the expansion score increased after ICI treatment in C04, C05, C06.1,
C12, and C17.0; the migration score increased in C04, C05, C06.0, C06.1, C08, C14, and
C17.0; and the transition score decreased in C04, C06.0, C08, C12, C17.0, and C18
(Supplementary Fig. 14a). These changes—higher expansion and migration scores and
lower transition scores—were primarily observed in IgG4-high clusters. The C12 cluster
showed increased expansion and transition scores after ICI treatment, consistent with
trends observed in other IgG4-high clusters. The C08 cluster, the only IgG4-low cluster
exhibiting changes after ICI treatment, demonstrated an increased migration score and
decreased transition score, aligning with trends in the IgG4-high clusters. Regarding
migration potential, LN–tumor migration potential increased in C05, while LN–PBMC
migration potential increased and PBMC–tumor migration potential decreased in C08
(Supplementary Fig. 14b). Thus, STARTRAC analysis comparing the ICI-treated and ICI-
naive patients, confirmed the enhanced expansion and migration potential of ICI-bound T
cells after ICI treatment, particularly an increased LN–tumor migration potential in the LN-
T_{pex} population (C05).

Furthermore, to assess CD8⁺ T cell migration and differentiation within C05, C12,
and C04 clusters across organs, we conducted a pseudotime analysis using scRNA-seq
data from three ICI-treated patients. A flow from C05 (CD8-T_{pex})→C12 (CD8-T_{pex})→C04
(CD8-T_{ex}) was observed, suggesting that CD8-T_{pex} in C05 represents the initiation of
CD8⁺ T cell activation post-ICI treatment, with C12 as the intermediate stage in the tumor
(Supplementary Fig. 17).

Overall, these findings support the hypothesis that ICI treatment enhances the
expansion and migration potential of ICI-bound T cells and boosts LN–tumor migration
potential in the LN-T_{pex} population (C05). C12 was also an IgG4-high cluster, exhibiting
increased expansion scores after ICI treatment, but did not show a change in migration

potential. It likely represents the immediate stage of ICI-bound Tpex in the tumor. We
added these results and discussed the results in the revised manuscript (page 10 lines
322-329, page 11 lines 346-356, page 11 lines 364-367).

**Figure legend: Pseudotime analysis of CD8⁺ T cells in lymph nodes and tumor.**
Pseudotime analysis of T cells among clusters C04 (CD8-TEX), C05 (CD8-Tpex), and C12
(CD8-Tpex). Pseudotime was calculated using sc.tl.dpt.

**Minor concerns:**

**1) Figure 1: needs better explanation for cluster assignments. For example, authors**
**assigned effectors based on (*GrzmK*) expression, but in ext fig 1a – highest level of**
***GrzmK* was in Tpex. This should be explained in more depth and more markers**
**should be added to the text.**

We apologize for the complexity of the cluster annotations. To address this, we
added details of annotation for all clusters in Supplementary Fig. 2b. Additionally, we
included new canonical genes related to effector function, such as *TGFB1*, *GNLY*, *GZMH*,
and *FGFBP2*, to refine the annotations (Fig. 1c). In particular, *GNLY* and *FGFBP2* were
shown to be more highly expressed in CD8-effector cells (C08) than in CD8-Tpex cells
(C05). We have updated Fig. 1c and incorporated the corresponding description into the
Results section (page 4 lines 97-105).

**2) Why did the authors not exclude doublet clusters from their analysis? This could**
**compromise the interpretation of the results.**

Thank you for the reviewer's thoughtful comment. Doublet filtering generally
predicts a "doublet" population that is close to the simulated doublet cell. While this method
works well when applied to entire tissues and PBMCs, if it is applied only to T cells, the
doublet score for some intermediate cells will be high. To avoid this bias, we did not
perform automatic doublet filtering in this study. Re-embedding clusters other than cluster
18 may be useful, but we left the doublet in place to avoid making clusters harder to identify
due to the shuffle caused by re-embedding.

**3) Ext data fig 1A: This representation is quite confusing and not intuitive. Also**
**missing important markers like *Tcf7*.**

Thank you for raising this important question. The figure in question illustrates
automatic marker gene detection, which we have previously reported to be insufficient (ref
36). The key canonical marker genes are listed in Fig. 1c, while the full set of marker genes

identified by automatic marker gene detection is shown in Supplementary Fig. 2a.
Therefore, we believe that including both presentations is essential to provide detailed
annotations for clustering (Supplementary Fig. 2a, b).

**4) Fig2. it would significantly help if the authors provided more details about the**
**bioinformatics analysis and for example why they choose non-negative matrix**
**factorization (NMF) analysis for their analysis rather than saying Non-negative matrix**
**factorization (NMF) is useful for profiling T cells with ambiguous boundaries... what**
**does it mean ambiguous boundaries here ?**

Thank you for the important comment. We previously reported the robustness of
NMF analysis (ref 36). As we described in this paper, we used NMF to extract both cell-
type identity and cellular activity programs, thereby addressing the profiling of complex cell
populations. By “ambiguous boundaries,” we refer to the challenges in defining clear
distinctions between T cell subsets, as many subsets exhibit overlapping gene expression
profiles and functional states. Traditional clustering methods often assign cells to discrete
groups that may not fully capture the continuum of cellular states, especially in dynamic or
transitional populations. We have added the details about NMF analysis in the results and
methods section (page 6 lines 176-178, page 18 lines 584-596).

**5) Figure 2C: why is NMF4 missing?**

Thank you for your comment. NMF4 was excluded because no significant pathway
could be detected. We have added this explanation in the legend for Fig. 2c.

**6) Figure 3: Include gating tree**

Thank you for pointing this out. We have added a flow cytometry gating tree to the
Supplementary Fig. 7.

**7) Figure 4E-H: It would help significantly if we had number of samples used in 5E-H**
**(especially for Tumor and Meta LN samples that look like there are only 2 samples**
**there)**

Accepting reviewer’s suggestion, we have added a description of the number of
samples to the legend for Fig. 4e-h.

**8) Figure 5C: what is 1, 2, 3? Is this TCR sequences that were found more than 3**
**times?**

We regret not having provided a detailed description in the Figure legend. The right
panel in Fig. 5c shows the actual clone numbers, indicating how many cells share the same
TCR, while the left panel indicates clone sizes specifically labeled to represent categories
with clone size 1, 2, and greater than 3. We have added a detailed explanation of clonal
expansion in the legend for Fig. 5c.

**9) Figure 5G-H: explain better. Is this TCR sequences (alpha, beta chain), or is this**
**amino acid sequence specifying similar binding properties? How do they define a**
**clonotype – matching TCR sequence? How do they define clonotype matched to C5**
**cluster – T cells that share the exact same TCR sequence? Or similar amino acid**
**sequence?**

Thank you for the important questions. TCR analysis was performed using Scirpy
(0.10.1) (ref 64). Clonotypes were defined using `ir.tl.define_clonotypes`

(receptor_arms="all", dual_ir="any")
[https://scirpy.scverse.org/en/latest/generated/scirpy.tl.define_clonotypes.html]. The same
clone was defined as the one in which the DNA sequence of CDR3 of VD and VDJ
matched. For the assigned clones, STARTRAC (0.1.0) analysis (ref 42) was performed to
infer the expansion, migration, and transition status. For the clone network construction, we
used the ir.tl.repertoire_overlap function to calculate the Jaccard index, which was then
employed as a measure of clonotype similarity. "Jaccard Index" was used for the similarity
metric. We have added this explanation in the Methods section (page 18-19 lines 604-614)
and the legend for Fig. 5g.

**10) Authors may comment the result btw different figure for example why there is no**
**significant of Tpex with better prognosis in gastric cancer (Fig2) and why there is**
**better prognosis in gastric in fig 3f and J, this is because of they include LN...**

Thank you for the thoughtful comment. As the reviewer pointed out, intratumoral
Tpex was not correlated with prognosis from the TCGA database (Fig. 2d) because Tpex
was assessed in tumor, while an abundant Tpex infiltration in metastasis-free LNs was
correlated with a favorable prognosis in ICI-naive gastric cancer patients (Fig. 3f), since
Tpex was assessed in LNs. We have addressed this point in the Manuscript (page 7 lines
217-219, page 12 lines 388-391).

**Reviewer #2**

***In this manuscript, the authors aim to characterize the anti-tumor immunity of***
***progenitor exhausted T cells (CD8-Tpex) in gastrointestinal cancers. They also used***
***anti-IgG4 antibody combined with epitopes sequencing (CITE-seq) with scRNA/TCR-***
***seq (scRNA/TCR/CITE-seq) to study the ICI-bound T cells in tumors, lymph nodes***
***(LNs), and blood from one ICI-treated gastric cancer patient. They showed that there***
***was a clonal link between ICI-bound CD8-Tpex in LNs and intratumoral ICI-bound***
***CD8-Tex.***

***Although the mechanistic analysis of ICI-bound T cells in ICI-treated cancer patients***
***is interesting and important, I am confused by the flow of the manuscript. The***
***manuscript starts by analyzing the T cells from tumors, metastatic and metastasis-***
***free LNs, and blood from two esophagogastric junction (EGJ) cancer patients who***
***had received chemotherapy and anti-PD-1 antibody therapy. They then projected***
***published scRNAseq data of tumors from 15 ICI-treated lung cancer patients on to***
***their EGI dataset and claimed that a CD8-Tpex T cell subset from the LNs of the two***
***EGJ patients is associated with better therapy response. They redefined the T cell***
***subsets from the two EGJ patients using non-negative matrix factorization (NMF)***
***and tested their association with survival in the TCGA dataset of gastric***
***adenocarcinoma, finding no association between Tpex abundance and patient***
***survival.***

***Subsequently, they analyzed the LN Tpex from 55 ICI-naive gastric cancer patients***
***using flow cytometry and showed that the ratio of Tpex/Tex in metastasis-free LNs***
***was associated with better survival outcomes. They then compared the LN Tpex***
***from 12 ICI-treated gastrointestinal cancer patients with the ICI-naive gastric cancer***
***patients and found that ICI treatment potentially increases the proliferative activity of***
***the LN Tpex but not their cytotoxicity. Finally, they characterized the ICI-bound T***
***cells in two(?) ICI-treated gastric cancer patient and found that ICI-bound CD8-Tpex***
***had a higher cross-tissue migration potential and high clone overlaps with***
***exhausted CD8 T cells (CD8-Tex) in the tumor.***

**Overall, the observations of the scRNAseq and flow cytometry data from the ICI-**
**naive and ICI-treated gastric cancer patients are diverse and, many times, do not**
**converge on concrete conclusions. Please find the specific comments below.**

We thank the reviewer for constructive comments on our manuscript. The
statement by the reviewer that “the mechanistic analysis of ICI-bound T cells in ICI-treated
cancer patients is interesting and important” is highly encouraging to us. Accepting the
reviewer’s suggestion, we created an overview summarizing the samples and experimental
methods for each figure, to enhance the understanding of the overall connections of this
study (Supplementary Fig. 1).

**Comments:**

**1) As the authors noted, Figure 1b shows that all the CD3+ T cells sampled from**
**various locations/organs exhibit significant organ specificity. However, the authors**
**provided the same annotation for these cells from distinct clusters originating from**
**different organs (Fig 1b). This is confusing and raises doubts about the correctness**
**of the classification. For instance, the authors defined T_{pex} cells by the high**
**expression of genes TCF7, PDCD1, and CXCL13. However, among the two annotated**
**CD8-T_{pex} clusters (C05 and C12), C12 clearly lacks the expression of CXCL13.**

Thank you for your insightful comment. As shown in Fig. 1d, global tissue specificity
in cell characteristics was observed. However, we performed clustering in a unified
representation, as described in previous studies that simultaneously evaluated populations
common across tissues (ref 35, 42). Consequently, many clusters were derived from
multiple tissues, as demonstrated in Supplementary Fig. 3a. This approach enables
consistent cell classification and facilitates analyses, such as STARTRAC, by leveraging
the advantage of clustering across tissues.

To enhance the clarity of the annotation description, we have included detailed
explanations of all clusters in Supplementary Fig. 2b. According to this definition, CD8-T_{pex}
was characterized by the expression of *TCF7* and *PDCD1*. While C05 expressed *CXCL13*,
C12 did not. In response to the reviewer’s observation, we have removed *CXCL13* from the
definition of CD8-T_{pex} in the Manuscript (page 4 line 102).

**2) The authors later mentioned that "The transcriptomic similarity of clusters was**
**observed mainly according to tissue". Is this an actual difference (i.e. is it true that**
**the naive T cell populations in LN and PBMC are very distinct)? The UMAP and the**
**correlation analysis show that C2 and C10 are separated from C0 and C11 - I am not**
**sure if this is true difference or there is some batch effect that was not accounted for**
**here. Also, C05 and C08 are correlated in Fig 1E but they are much further in the**
**UMAP. Why is that?**

Accepting the reviewer’s suggestions, we investigated the differentially expressed
genes between C02+C10 (naive T cell in PBMC) and C00+C11 (naive T cell in LNs)
(Figure shown below). The results showed that the expression of *BATF*, which encodes a
tissue-specific transcription factor, was upregulated in C00+C11 (LNs), while the
expression of *CCR7* and *S1PR* genes involved in circulation was upregulated in C02+C10
(PBMC), indicating C02 and C10 are distinct from C00 and C11 in terms of gene
expression. We also analyzed the genes that were differentially expressed between C05
and C08 (Supplementary Fig. 3b). The results showed that the expression of AP1 family
genes (*CD69*) was upregulated in C05, while the expression of *S1PR1* and *TGFB1* was
upregulated in C08, indicating that C05 is genetically distinct from C08 (page 4 lines 113-

116). This is consistent with the gene expression program found to be elevated in tissues in
 our previous paper (ref 36).

Moreover, there is no sample bias shown in Fig. 1d, and this cannot be regarded as
 a batch effect. C05 and C08 are separated by reasonable distances because UMAP is a
 method to reduce the dimensions to two while maintaining the topology and does not
 completely maintain the gene expression distance. Therefore, there are cases where C05
 and C08 are separated by UMAP, as in this case. We included Fig.1e to clarify this issue.
 We believe this is not a batch effect but an actual difference between populations.

 **Figure legend: Volcano plot showing the differentially expressed genes between**
 **C02+C10 (PBMC) and C00+C11 (Lymph nodes).**

**3) C09 is the population of CD4 T follicular helper which is known to express CXCR5,**
 **PD1, TOX, and BCL6 - as the authors mentioned later in Fig 1F. There is no**
 **indication that these are CD4 T_{pex}. A part of C5 "T_{pex}" may have CD8+ CXCR5+**
 **population reported previously (PMID: 31028278) - this population resembles the**
 **CD4 T_{fh} in their markers and these seems to be found in the LN.**

We agree with the reviewer's opinion. A paper (ref 29) demonstrated that
 CXCL13⁺CH25H⁺IL-21⁺PD-1⁺CD4⁺ T cells (CXCL13⁺ T_{fh}) and PD-1⁺TCF1⁺CD8⁺ T cells
 (progenitor T cells) are related to the therapeutic effect of ICIs and interact to exert clinical
 effect. Among these, T_{fh} cells express CXCL13, BTLA, CD200, IL21, TCF7, and IL6ST,
 while progenitor T cells express CXCL13, XCL1, and T_{fh}-associated marker genes
 (CD200, BTLA, GMG4). Therefore, we consider C09 to be a population with characteristics
 similar to those of CD4⁺ T_{fh} cells, based on gene expression markers such as CXCL13 and
 TCF7, as mentioned in our manuscript. Furthermore, the study cited by the reviewer (ref
 31) indicates that CXCR5⁺CD8⁺ T cells interact with T_{fh} cells in LNs to enhance antitumor
 immunity. In our data, C05 and C09 exhibited high expression of CXCR5, PDCD1, TCF1,
 TOX, and BCL6 (Fig. 1c). Therefore, C05 includes the CXCR5⁺CD8⁺ T cell population and
 may interact with T_{fh} cells in LNs. However, in this study, we designated the CD4⁺ T_{fh}
 population as "CD4-T_{pex}" solely on the basis of the expression of CD4, PDCD1, and TCF7.
 We have described this point in the Results section (page 5 lines 120–122), and added the
 reference (ref 31).

**4) The authors stated that “a significantly higher proportion of CD8-Tpex (C05) and**
**CD8-effector (C08) in the MPR group than in the non-MPR group, indicating that**
**abundant intratumoral CD8-Tpex and CD8-effector were associated with a favorable**
**therapeutic effect of ICI (Fig. 1G).” However, neither C05 nor C08 were tumor-**
**infiltrating T cells in Fig 1B dataset: C05 were found mainly in lymph nodes (LNs)**
**and C08 in blood. Are the projected T cells from the lymph nodes of the lung cancer**
**patients (in GSE173351)? If they are not from the LN but from the tumor, how did the**
**authors correlate these non-tumor T cell clusters (C5 and C8) to the tumor-infiltrating**
**T cells from lung cancers and draw conclusions on their prognostic power?**
**Furthermore, in Figure 1g, no statistical significance of the comparison was**
**mentioned. What is the meaning of “Credible”?**

Thank you for your valuable comment. As the reviewer pointed out, the projected T
cells were derived from lung cancer tumors (GSE173351). We used this dataset because
we were unable to identify datasets other than that from the study (ref 35) that investigated
T cells in tumors after ICI treatment with scRNA-seq and the treatment efficacy of ICI. In
addition, only a few publications have evaluated Tpex in LNs after ICI treatment in human
samples (ref 25, 29), and these studies did not examine the correlation between the
abundance of the Tpex population in LNs and ICI treatment efficacy or prognosis. Thus,
there is currently no scRNA-seq dataset that directly evaluates Tpex in LNs after ICI
treatment in human samples and examines its association with ICI treatment efficacy or
prognosis. To infer this association, we used the dataset from the previous paper (ref 35)
and concluded that “a significantly higher proportion of CD8-Tpex (C05) and CD8-effector
(C08) in the MPR group than in the non-MPR group indicates that abundant intratumoral
CD8-Tpex and CD8-effector were associated with a favorable therapeutic effect of ICI.”
While we may hypothesize that a higher proportion of CD8-Tpex in tumors after ICI
treatment correlates with a favorable response to ICI from this result (Supplementary Fig.
4c), we acknowledge that it is difficult to draw conclusions regarding the correlation
between the Tpex population in LNs and treatment efficacy or prognosis. Therefore, in line
with the reviewer’s suggestion, we have decided to relocate this data to Supplementary Fig.
4c, temper our conclusions about this figure, and strengthen our logic focusing on Tpex in
LNs by citing several studies (page 5 lines 128-131).

“Credible” means significantly changed, since a cell population that is considered
significantly (FDR <0.05) different in the single-cell compositional data analysis (scCODA)
framework was described as “credible” in a previous report (ref 63) (Supplementary Fig.
4c). In scCODA, Bayesian estimation is used to calculate the posterior probability of cell
frequencies, and a qualitative assessment is made as to whether the difference is greater
than the significance level. We have added this explanation in the Methods section (page
18 lines 597-603).

**5) What is the data supporting this statement (page 5, line 154-155) “CD4-Tpex cells**
**have acquired migratory and proliferative functions in LNs, possibly working on**
**CD8-Tpex cells, while still maintaining progenitor cell characteristics”? The reviewer**
**still thinks that the CD4 Tpex is a population of Tfh that may or may not be related to**
**the CD4 Tex. Did the author check if the TCR of CD4 Tpex and CD4 Tex highly**
**overlap?**

Thank you for your valuable comment. As the reviewer pointed out, CD4-Tpex is
thought to overlap with the previously reported Tfh. The TCR similarity between C09 (CD4-
Tpex) and C06.1 (CD4-Tex) was not particularly high, as we have already shown in Fig. 5g.
Therefore, it remains unclear whether CD4-Tpex cells migrate and differentiate into CD4-
Tex cells in tumors, unlike CD8 populations. However, CD4-Tpex cells in LNs exhibited
high expression of genes encoding co-stimulatory molecules (*TIGIT*, *LAG3*), a gene related
to T-cell exhaustion (*TOX*), proliferation-associated genes (*DUSP4*, *DUSP6*), and

chemokine receptor genes (*CXCL13*, *CXCR3*) compared with CD4-naive/Tcm cells in LNs.
Additionally, they showed elevated expression of naive/progenitor-associated genes
(*TCF7*, *CCR7*) relative to CD4-Tex cells in tumors and liver metastases, suggesting that
CD4-Tpex cells in LNs possess migratory and proliferative functions while maintaining
progenitor-like characteristics. The referenced study (ref 31), which the reviewer
mentioned, demonstrated that *CXCR5*⁺*CD8*⁺ T cells (a subset of CD8-Tpex) interact with
CD4-Tfh to enhance anti-tumor immunity, suggesting a potential interplay between Tfh and
CD8-Tpex in promoting anti-tumor immunity. Accordingly, the phrase “possibly working on
CD8-Tpex cells” was revised to “interacting with CD8-Tpex” (page 6 lines 167-168), and the
reference (ref 31) was added.

**6) The NMF analysis of their scRNAseq dataset (Fig 2) appears inconsistent with**
**their clustering analysis (Fig 1b) and the similarity analysis of these clusters (Fig 1e).**
**For example, NMF4 includes C0, C11, C1, C2, and C10 (Fig 2a, Fig 1b). C0 and C11**
**are from lymph nodes (LNs), which do not share high similarity with C1, C2, and C10,**
**which are from blood (Fig 1e). This raises doubts about the robustness of the NMF**
**analysis or the initial clustering analysis in Fig 1.**

Thank you for the insightful comment. The robustness of NMF analysis was
reported previously (ref 36, PMID 39418984). NMF is extracted from the highly expressed
variation by factor. As the correlation plots are influenced by the large variation, local
features such as NMF4 can be underestimated. However, since this can be misleading as
the reviewer mentioned, we have added an explanation of NMF analysis in the Methods
section (page 18 lines 584-596).

**7) Furthermore, it is unclear how the biological processes represented by NMF1 in**
**Fig 2c are relevant to Tpex cell state and ICI response.**

Thank you for your thought-provoking question. NMF1 and NMF8 corresponded
with CD8-Tpex. NMF1 was associated with the naive population, defined by the expression
of *NR4A2* and *SLC2A3*, and correlated with interferon signaling (Fig. 2a-c). NMF8 was
associated with the Tpex population, defined by the expression of *GZMK*, *FGFBP2*, and
*CST7* and correlated with interferon signaling and immunoregulatory interactions between
lymphoid and non-lymphoid cells (Fig. 2a-c). As this dataset was derived from TCGA of
patients without ICI treatment, we are unable to evaluate ICI responses based on these
figures (page 6 lines 183-186).

**8) The absence of any significant correlation between the NMF1 or NMF8 gene**
**program enrichment in TCGA does not imply that the Tpex accumulation is not**
**important in the prognosis of ICI-naive gastric cancer. The authors has not**
**established that the enrichment of NMF1 or NMF8 onto a bulk RNAseq dataset of**
**TCGA is specific to the T cell and not other cell types in the tumor.**

Thank you for your perceptive question. Indeed, we acknowledge that the NMF-
defined gene programs in T cells are not necessarily T-cell specific. To verify the activity of
these gene programs across different cell types, we established a method to assess their
activation by calculating the explained variance (Evar) (ref 36). We evaluated the Evar of
the gene programs defined in this study across various cell populations using public
scRNA-seq data from gastric cancer samples (GSE206785, PMID 36550535), as shown
below. Our findings revealed that NMF1 exhibited high Evar in CD4⁺ T cells and mast cells,
while NMF8 had high Evar in CD8⁺ T cells and NK/innate lymphoid cells. These
observations suggest that NMF1 and NMF8 are, indeed, T cell-specific gene programs.

Figure Legend: Heatmap of the Explained Variance (Evar) for each gene program across different cell types in gastric cancer. Each column is normalized individually. The data used in this analysis are from GSE206785 (PMID 36550535).

**9) On Page 5, lines 184-188, why did the authors think that Extended Figures 5a and**
 **5b support their hypothesis that “anti-tumor immunity was induced in LNs close to**
 **the tumor”? Extended Figure 5 compares the frequency of IFN- γ + and TNF- α + CD8 T**
 **cells among CD8+PD1+ vs. CD8+PD1- cells across pathological tumor stages. It was**
 **expected that CD8+PD1+ T cells would have higher IFN- γ and TNF- α production than**
 **CD8+PD1- T cells, since PD1 itself is a T cell activation marker. However, I do not**
 **understand how this data is related to Figure 3b and the authors’ claim that “anti-**
 **tumor immunity was induced in LNs close to the tumor.” The authors should show**
 **that IFN γ and TNF expression data in the same grouping as Fig 3b and not by**
 **pathological tumor stage.**

Accepting the reviewer’s suggestion, we removed the previous figure for each
 pathological stage and evaluated the frequencies of IFN- γ and TNF- α expression in PD-1+
 and PD-1- T cells in metastatic LNs, proximal LNs, and distal LNs (Supplementary Fig. 8).
 Notably, there were no significant differences in cytokine production between PD-1+ T cells
 in proximal and distal LNs. This may be attributed to the cytokine production being
 assessed using forced T-cell stimulation with PMA and ionomycin, which is not antigen-
 specific. However, PD-1+ T cells exhibited significantly higher cytokine production
 compared with PD-1- T cells in both proximal and distal LNs. Moreover, checkpoint
 molecules, including PD-1 and CD103, were expressed at higher levels in pLNs than in
 dLNs. Several studies have reported that checkpoint molecules such as PD-1, CD103,
 CD39, and CD137 may serve as markers for tumor antigen-specific T cells (PMID
 31892342, 36792124, 36574773). Taken together, these findings suggest that anti-tumor
 immunity is induced in LNs proximal to the tumor. We have revised the description in the
 Results section (page 7 lines 203-211).

**10) “The frequency of CD8-Tpex in metastasis-free LNs was lower in advanced tumor**
 **stages (Fig. 3d).” Is this analysis only done in the proximal metastasis free LN?**

Thank you for your comment. This dataset includes all metastasis-free LNs (Fig.
 3d, right), encompassing both proximal and distal LNs. The same analysis was conducted

solely on proximal LNs, revealing that the frequency of CD8-Tpex among CD8⁺ T cells was
 significantly lower in Stage III, as shown below.

 **Figure legend: Percentage of TCF1⁺PD-1⁺, TCF1⁻PD-1⁺, and PD-1⁻ cell population**
 **among CD8⁺ T cells in proximal metastasis-free lymph nodes according to tumor**
 **stage.**

 **11) "The expression frequency of molecules among CD4⁺ T cells in each tissue was**
 **similar to that of CD8⁺ T cells (Fig. 3g)." The reviewer is unsure of the relevance of**
 **Tim3 and CD103 in CD4 T helper populations. Furthermore, Fig 3i also show an**
 **increase in GZMB expressing CD4 Tex - what is this population?**

 Thank you for your insightful comment. In peripheral blood, tumor-reactive T cells
 have been shown to be present in PD-1⁺CD4⁺ T cells but not in PD-1⁻CD4⁺ T cells (PMID
 31609250). Regarding Tim-3 expression in the CD4⁺ T cell population, studies on infectious
 diseases have demonstrated that Tim-3 is co-expressed with PD-1 in exhausted T cells
 during antimicrobial responses (PMID 21697013). In tumors that progress following an
 initial response to anti-PD-1 therapy, Tim-3 upregulation has been observed in PD-1
 antibody-bound T cells, and blockade of Tim-3 after PD-1 inhibition failure has shown a
 survival advantage (PMID 26883990). Therefore, PD-1 and Tim-3 are considered
 exhaustion markers in CD4⁺ T cells, similar to CD8⁺ T cells. Although CD103⁺CD4⁺ T cells
 in tumors have been reported to be a marker of high cytokine production and good
 prognosis (ref 40, 41), there are limited reports on their presence in LNs and peripheral
 blood. The results showed very low expression of CD103 in CD4⁺ T cells in pLN and dLN,
 with no significant differences (Fig. 3g). As the reviewer pointed out, GZMB expression in
 Tex cells within metastasis-free LNs was higher than in tumors; however, its expression in
 CD4⁺ T cells was markedly lower than in CD8⁺ T cells, suggesting that it may have limited
 biological significance. We have revised the relevant sentences in the Results section
 (page 7-8 lines 221-226, page 8 lines 229-231).

 **12) In Figure 3d, metastasis-free LNs from Stage III seem to have a lower Tpex/Tex**
 **ratio compared to Stages I and II. I suggest that the authors perform a multivariate**
 **Cox hazard analysis (as they did in Figure 2d) to determine if Tpex/Tex is an**
 **independent prognostic marker or if it is just reflecting tumor stages (the ratio**
 **correlates with stages in Fig 3d and 3h). Additionally, did the authors test the**

**survival association of Tpex, Tex frequency, or the Tpex/Tex ratio in tumors and**
**metastatic LNs?**

Appreciating the reviewer's suggestion, we conducted univariate and multivariate
analyses for recurrence-free survival (RFS) in 49 ICI-naive gastric cancer patients
(Supplementary Table 3). Univariate analysis identified sex, pStage, and Tpex/Tex ratio in
metastasis-free LNs as significant factors. Multivariate analysis revealed that a high CD8-
Tpex/Tex ratio in metastasis-free LNs tended to be a statistically significant prognostic
factor for RFS (P=0.081) (page 7 lines 219-221, page 14 lines 467-469).

Additionally, Kaplan–Meier survival curves for RFS and overall survival (OS) were
compared between patients with high and low Tpex/Tex ratios in tumor and metastatic LNs
(Supplementary Fig. 9). No significant differences in RFS or OS were observed between
these groups in tumor and metastatic LNs, although patients with high Tpex/Tex ratio had
better survival both in tumor and metastatic LNs. We have added Supplementary Fig. 9 and
have described the corresponding results in the Results section (page 7 lines 217-219).

**13) "On the other hand, cytotoxicity evaluated by GZMB was not significantly**
**different between each T-cell subset in patients with ICI-naive and ICI-treated**
**cancers (Fig. 4f)" - there seems to be one statistically significant difference in Fig 4f.**
**More puzzlingly, there is a uniform, statistically significant upregulation of GZMB**
**across all CD4+ T cell groups in meta free LNs in Fig 4h but the authors stated:**
**"These results were consistent with CD4 + T cells (Fig. 4g,h). Thus, ICI treatment**
**does not affect the cytotoxicity of Tpex in metastasis-free LNs, but greatly increases**
**their proliferative potential".**

We regret not having provided a more detailed description of these findings. As
noted, GZMB expression was lower in Tpex and PD-1⁻CD8⁺ T cells within metastasis-free
LNs among CD8⁺ T cells after ICI treatment. In CD4⁺ T cells, GZMB expression was
significantly higher in Tex of metastasis-free and PD-1⁻CD4⁺ T cells of metastatic LNs after
ICI treatment; however, its expression rate remained extremely low compared with that in
CD8⁺ T cells, suggesting that the significance of GZMB expression in CD4⁺ T cells is
limited. In contrast, Ki-67 expression was significantly elevated across CD8⁺ T cell subsets
within metastasis-free LNs and CD4⁺ T cell subsets within all LNs. Accordingly, we stated
that "ICI treatment does not affect the cytotoxicity of Tpex in metastasis-free LNs, but
greatly increases their proliferative potential." We have revised this sentence in the
manuscript (page 9 lines 274-284).

**14) Is it correct that both CD4 Tpex (of Tfh) and Tex are not clonally expanding as**
**shown in Fig 5c? The authors have been drawing parallel conclusions across the**
**CD8 and CD4 Tpex and Tex populations but most of the data, including this TCR**
**data, suggest that they are different.**

Thank you for the thoughtful comment. Though we did not describe the clonal
expansion of CD4⁺ T cells in detail, we fully agree with the reviewer's opinion. The
frequencies of clonally expanded cells in CD4-Tex (C06.1/C17.2) were 83% and 55%,
respectively, while that of CD4-Tpex (C09) was 36% (Fig. 5c, Supplementary Table 6). In
contrast, the frequencies of clonally expanded cells in CD8-Tex (C04/C06.0/C17.0/C17.1)
were 88%, 93%, 89%, and 71%, respectively, and in CD8-Tpex (C05/C12), it was 62% and
72%, respectively. Based on these results, it appears that CD8-Tpex and CD8-Tex were
clonally expanded following ICI treatment, whereas CD4-Tpex showed limited clonal
expansion after ICI treatment. We have described this point in the Results section (page 11
lines 333-334).

**15) The reviewer noted that most of the analysis in Fig 5 is based on only two**
**patients, which limits the ability to generalize any conclusions. Additionally, the**
**authors should provide some supporting data (maybe as supplementary) that the**
**clonal expansion, migration and transition across the organs and/or T cell clusters**
**are consistent between the two patients to ensure we are not looking at a pattern**
**dominated by a single patient.**

Appreciating the reviewer's suggestion, we have added one additional ICI-treated
(patient 3) and two ICI-naive gastric cancer patients (patient 4,5) to the previous
scRNA/TCR/CITE-seq analysis. In total, we performed single-cell RNA/TCR/CITE-seq in
two ICI-naive cancer patients and three ICI-treated cancer patients (Fig. 5, Supplementary
Fig. 12, Supplementary Fig. 14). Regarding the consistency in the clonality, each dot in the
STARTRAC analysis represents actual patient data, and high values were consistently
observed within each cluster (Fig. 5e, f, Supplementary Fig. 14). Thus, we confirmed that
the values in clonal expansion, migration, and transition were consistent across T-cell
clusters.

**16) "A high cross-tissue migration score was observed in CD8-Tpex (C05, C12) and**
**CD8-effector (C08) cells" - are the T cells in C08 clonally related to C05 and C12?**
**What is the authors opinion on the sequence of the migration of tumor infiltrating**
**Tex T cell clones? Is it starting in the proximal LN? In line 263, the authors stated**
**"Regarding migration, the most observed route of migration was between LN and**
**tumor (C04, C05, C06.0, C08) (Fig. 5f)." It is noted that C08 is from PBMC and not**
**from LN. The reviewer is also curious that the PBMC C08 clones are not discussed**
**as much as LN Tpex C05 despite a strong clonal overlap with tumors and a high**
**expansion, migration and transition indices.**

Thank you for your insightful comment. The percentage of TCR sharing of C08 cells
with CD8-Tex (C04/C06.0) and CD8-Tpex (C12) cells in tumor was 19.6% and 7.0% in
high- and low-IgG4 clones in patient 1, 64.9% and 50.9% in patient 2, and 7.7% and 16.9%
in patient 3 (Supplementary Fig. 15c). In contrast, the percentage of TCR sharing of C08
cells with CD8-Tpex (C05) cells in LNs was 75.3% and 79.1% in high- and low-IgG4 clones
in patient 1, 30.9% and 47.3% in patient 2, and 50.8% and 33.8% in patient 3
(Supplementary Fig. 15c). These results indicate that while C08 cells in patient 2 exhibited
notable TCR similarity with C04/06.0/12 cells, C08 cells generally shared a higher
proportion of TCR clones with C05 cells, regardless of IgG4 binding status.

As the reviewer suggested in query 21, we performed a STARTRAC analysis
based on ICI high- and low-bound T cells within the C08 cluster (Supplementary Fig. 16).
The results demonstrated that ICI low-bound T cells exhibited higher expansion, migration,
and transition scores than ICI high-bound T cells, while the pairwise migration index
remained similar between the two groups. This may be attributed to the fact that C08
clones contain recently activated effector memory (TEMRA) T cells (ref 42), which are
clonally expanded (Fig. 5c, Supplementary Table 6). A previous study on STARTRAC
analysis using T cells from tumors, adjacent normal tissues, and blood (ref 42) reported
that TEMRA cells express numerous effector molecules such as *PRF1*, *GZMB*, and *GZMH*,
but lack Tex cell markers such as *PDCD1* and *HAVCR2*. The study also demonstrated that
TEMRA cells are enriched in PBMCs and exhibit the highest expansion, migration, and
transition scores, as well as the highest migration potential between blood and tumor, blood
and normal tissue, and normal tissue and tumor. Thus, it is thought that TEMRA cells
match ICI low-bound T cells within C08 cells. Given that TEMRA cells do not express PD-1,
they are likely unaffected by ICI treatment. Therefore, the high expansion, migration, and
transition scores detected after ICI treatment in our data likely reflect the inherently high
proliferative and migratory potential of TEMRA cells within the C08 cluster.

Regarding the difference of TCR similarity to CD8-*Tex* and CD8-*Tpex* in tumor
between C05 and C08 cells, the percentages of TCR sharing of C05 cells with CD8-*Tex*
(C04/C06.0) and CD8-*Tpex* (C12) were 43.2% and 7.8% in high- and low-IgG4 clones,
respectively, in patient 1, 94.0% and 59.3% in patient 2, and 33.0% and 17.3% in patient 3
(Supplementary Fig. 15b). Compared with the percentage of TCR similarity between C08
cells and C04/06.0/12 cells (lines 651-654 in response letter) (Supplementary Fig. 15c),
C05 cells more frequently shared clones with CD8-*Tex* and CD8-*Tpex* cells in the tumor
than did C08 cells. Importantly, the clonotypes in blood TEMRA cells with low PD-1
expression were mutually exclusive with those in tumor *Tex* cells in the previous study (Ref
42). Thus, the high TCR similarity between C08 and C04/06.0/12 was likely a result of the
combined high TCR similarity between C08 and C05, as well as between C05 and
C04/06.0/12.

In summary, based on the TCR overlap with CD8-*Tex* and CD8-*Tpex* cells in tumor,
C05 cells were considered as the primary origin of CD8-*Tex* cells in tumor. The high
STARTRAC scores of C08 cells reflect the inherently high proliferative, migratory, and
transitional potential of TEMRA cells within the C08 cluster. We have described this
explanation for the C08 cluster in the manuscript (pages 10-11 lines 329-333, page 11 lines
356-364).

**17) The *Tex* clusters have high transition scores and the authors stated "intratumoral**
**CD8-*Tex* cells had a high potential to differentiate and expand"- To the reviewer's**
**understanding, the *Tex* are exhausted T cells that are not undergoing further state**
**transition.**

Thank you for the important question. The current study defines TCF1-PD-1⁺ T
cells as *Tex*, but this is a rather broad definition previously reported to include intermediate
and terminally exhausted types, with the former transitioning to the latter (PMID 32396847,
37968405, 37820583). It has also been reported that it is not *Tpex* but *Tex* (especially
intermediate *Tex*) that are involved in cytotoxicity or have an effector function (ref 19, PMID
37968405, 37820583). Thus, exhausted T cells (*Tex*) have different status which include
progenitor, intermediate, and terminally exhausted T cells. Among these, terminally
exhausted T cells do not undergo further state transitions, whereas other forms of
exhausted T cells have the potential to differentiate and expand.

**18) "All of these T cells were bound by ICIs except for C12, suggesting that ICI-**
**bound T cells play a leading role after ICI administration." I am not sure if this is what**
**I observe from Fig 5a - there are some IgG4 signal in C12 and the mRNA expression**
**in C12 is similar to C5 (Fig 5b). This seems to be a rather incoherent description of**
**the data. The implication "suggesting that ICI-bound T cells play a leading role after**
**ICI administration" is even more confusing. What is the basis of this statement?**

Thank you for the quite important suggestion. Since the cutoff value for IgG4-high
in the scRNA/TCR/CITE-seq analysis was not predetermined, we set it as the median of
1.079, defining clusters with 50% or more IgG4⁺ cells as IgG4-high clusters (Fig. 5b,
Supplementary Fig. 12b). On the basis of this threshold, C12 (73.4%) was classified as an
IgG4-high cluster (Fig. 5b, Supplementary Fig. 12c). Accordingly, we have revised Fig. 5b
to reflect the updated IgG4-high clusters, and removed the phrase "except for C12" from
the manuscript (page 10 line 329).

Moreover, we additionally performed scRNA/TCR/CITE-seq with three ICI-treated
patients and two ICI-naïve patients (Fig 5, Supplementary Fig. 12, Supplementary Fig. 14).
The STARTRAC algorithm was used to compare clonal expansion between ICI-treated and
ICI-naïve samples (Supplementary Fig. 14a). The expansion score increased after ICI
treatment in C04, C05, C06.1, C12, and C17.0; the migration score increased in C04, C05,
C06.0, C06.1, C08, C14, and C17.0; and transition scores decreased in C04, C06.0, C08,

C12, C17.0, and C18. These changes—higher expansion and migration scores and lower
transition scores—were primarily observed in IgG4-high clusters, with opposite trends
rarely detected. The C12 cluster showed increased expansion and transition scores after
ICI treatment, consistent with trends observed in other IgG4-high clusters. Regarding
migration potential, LN–tumor migration potential increased in C05, while LN–PBMC
migration potential increased and PBMC–tumor migration potential decreased in C08,
though C08 cells, compared with C05 cells, less frequently shared clones with CD8-Tex
and CD8-Tpex in the tumor, which was shown in query 16 (Supplementary Fig. 14b). Thus,
STARTRAC analysis confirmed enhanced expansion and migration potential of ICI-bound
T cells, and particularly increased LN–tumor migration potential in the LN-Tpex population
(C05).

Furthermore, we conducted pseudotime analysis using scRNA-seq data from three
ICI-treated patients to assess CD8⁺ T cell migration and differentiation within C05, C12, and
C04 clusters across organs. A flow from C05→C12→C04 was observed (Supplementary
Fig. 17), suggesting that CD8-Tpex in C05 represents the initiation of CD8⁺ T-cell activation
after ICI treatment, with C12 as the intermediate stage in the tumor.

Overall, these findings support the hypothesis that ICI treatment enhances the
expansion and migration potential of ICI-bound T cells and boosts LN–tumor migration
potential in the LN-Tpex population (C05). Although C12 exhibited increased expansion
and transition scores post-ICI treatment, it likely represents the intermediate stage of ICI-
bound Tpex in the tumor. We have added Supplementary Fig. 17 and have discussed the
results in the revised manuscript (page 11 lines 364-367).

**Figure legend: Pseudotime analysis of CD8⁺ T cells in lymph nodes and tumor.**

Pseudotime analysis of T cells among clusters C04 (CD8-Tex), C05 (CD8-Tpex), and C12
(CD8-Tpex). Pseudotime was calculated using sc.tl.dpt.

**19) "Although some cells in the C05 migrated between LNs and PBMCs, most cells**
**migrated between LNs and tumor, and clonotypes within this cluster were highly**
**shared with the tumor-enriched CD8-Tex clusters (C04, C06.0) (Fig. 5f,g)." This**
**qualitative approach comparing "some cells" with "most cells" without**
**quantification is concerning.**

Accepting the reviewer's suggestion, we quantified the TCR sharing of C05 cells with
other clusters in each patient (Supplementary Fig. 15a). The percentages of TCR sharing
of C05 cells with CD8-Tex (C04/C06.0), CD8-Tpex (C12) and CD8-effector (C08) cells
were 89.8% in patient 1, 97.9% in patient 2, and 53.1% in patient 3 (Supplementary Fig.
15a). Thus, TCR clonotypes within the C05 cluster were highly shared with the tumor-
enriched CD8-Tex clusters (C04, C06.0) and CD8-Tpex (C12) clusters and the blood-
enriched CD8-effector (C08) cells.

However, after categorizing clonotypes based on high and low IgG4 binding, we
 found that ICI high-bound T cells in the C05 cluster predominantly shared clones with the
 tumor-enriched CD8-*Tex* and CD8-*Tpex* (C04, C06.0, C12) cells. whereas, ICI low-bound
 T cells in the C05 cluster shared clones with the PBMC-enriched CD8-effector (C08) cells
 (Supplementary Fig.15b). Specifically, the percentages of TCR sharing of C05 cells with
 CD8-*Tex* (C04/C06.0) and CD8-*Tpex* (C12) were 43.2% and 7.8% in high- and low-IgG4
 clones, respectively, in patient 1, 94.0% and 59.3% in patient 2, and 33.0% and 17.3% in
 patient 3. Meanwhile, the percentages of TCR sharing of C05 cells with CD8-effector (C08)
 were 41.8% and 74.4% in high- and low-IgG4 clones, respectively, in patient 1, 3.6% and
 35.6% in patient 2, and 12.2% and 40.4% in patient 3. In summary, the ICI high-bound
 cells in the C05 cluster had a high degree of consistency with TCRs of intratumoral CD8-
 *Tex* and CD8-*Tpex*, while the ICI low-bound cells in the C05 cluster had a high degree of
 consistency with TCRs of CD8-effector cells in PBMC. In response to this result, we have
 incorporated the quantification results into Supplementary Fig. 15 and revised the
 corresponding sentences in the Results section (page 11 lines 346-356).

**Figure legend: TCR sharing of C05 cells with other clusters in the 3 ICI-treated**
 **patients.** The percentage of TCR sharing of high and low-IgG4 clones in C05 with the
 tumor-enriched CD8-*Tex* (C04, C06.0) and CD8-*Tpex* (C12) and the blood-enriched CD8-
 effector (C08) are shown below.

 **20) "Importantly, clonotypes with high IgG4 binding (ICI high-bound T cells) shared**
 **clones with the tumor-enriched CD8-*Tex* clusters (C04, C06.0), whereas clonotypes**
 **with low IgG4 binding (ICI low-bound T cells) shared clones with the PBMC-enriched**
 **CD8-effector (C08) (Fig. 5h), indicating the possibility that ICI high-bound *Tpex* in**
 **LN contains cancer antigen-specific T cells, whereas ICI low-bound *Tpex* in LN**
 **may have a different antigen specificity." This statement needs to be supported by a**
 **quantification of clone overlap between IgG4 high clones in C05 with the tumor (and**
 **met) compared to the overlap with C8 PBMC clones.**

Thank you for your thoughtful suggestion. We have provided the response to this
 query in query 19.

**21) Based on Fig 5f, C08 has a strong pairwise migration index from PBMC to the**
 **tumor (almost as strong as C05 in LN - Tumor comparison). Can the authors**
 **generate the same analysis focusing the clones in C08 and separating them by IgG4**
 **high vs. low? This is important to confirm the source of *Tex* in the tumor (whether**

**they all come from Tpex in LN or not). Either way these observations are important**
**to be reported.**

Thank you for your constructive suggestion. We have provided the response to this
query in query 16.

**22) "We also discovered that PDCD1+ TCF7+ Tpex presence within tumors correlates**
**with the successful outcome of ICI treatment. Interestingly, this correlation did not**
**extend to the prognosis of ICI-naive gastric cancers. In contrast, abundant Tpex**
**presence in LNs was linked to favorable prognosis in ICI-naive gastric cancer, and**
**the proliferation activity of Tpex in LNs significantly increased after ICI treatment." I**
**am not sure the projection of lung tumors on the scRNAseq of gastric cancer can**
**reliably define that the PDCD1+ TCF7+ T cell in the TIL is the same as the Tpex in the**
**LN. This convoluted sentence is probably unnecessary and is confusing to the**
**reader. It is adequate to state the positive correlation in gastrointestinal cancers.**

Thank you for the important comment. As we mentioned at query 4, we thought that
T cells present in LNs and blood other than tumors could also be projected, because we
analyzed T cells that migrate between tissues after ICI treatment and also infiltrate tumors.
Furthermore, since preoperative ICI treatment is not currently a standard approach for
gastrointestinal cancers, it is difficult to obtain surgical specimens from patients treated with
ICI in routine clinical practice, and reports using gastrointestinal cancers after ICI treatment
are limited. Therefore, we used the datasets of these lung cancer patients (ref 35),
investigating the correlation between T cell populations in tumors after ICI treatment,
analyzed by scRNA-seq, and the treatment efficacy of ICI. However, as the reviewer
mentioned, we acknowledge that our data included T cells in different types of cancer and
also in non-tumor LNs and PBMC. Thus, we have decided to relocate this data to
Supplementary Fig. 4c, temper our conclusions about this figure, and strengthen our logic
focusing on Tpex in LNs by citing several studies (page 5 lines 128-130, 142-144).

**23) In line 339, the authors stated "In our data, abundant intratumoral Tpex were**
**associated with better response to ICI treatment". Is this pointing to the lung cancer**
**data? It is confusing. I really suggest the authors to remove the lung cancer data**
**analysis altogether (GSE173351) - it is hard to see why it is related to this study.**

Accepting the reviewer's suggestions, we have removed this sentence from the
revised manuscript (page 14 line 442-443).

849 850 **Reviewer #3**

**The study conducted by Takuro Saito's team investigates the impact of therapeutic**
**PD-1 antibodies on TCF1 and PD1-expressing progenitor exhausted CD8 T cells**
**(Tpex) by analyzing tumor tissues, peripheral blood, and lymph node samples from**
**gastrointestinal cancer patients both before and after ICI treatment. The authors**
**found that Tpex cells are enriched in the proximal lymph nodes of non-metastatic**
**tumors and then proliferate and migrate to the tumor site post-ICI treatment,**
**possibly differentiating into exhausted CD8 T cells (Tex) to exert anti-tumor effects.**
**However, I have several major concerns listed as below.**

**1) The authors mention that several studies have already demonstrated the**
**association of TCF1 and PD1-expressing Tpex in tumors and peripheral blood with**
**the response and prognosis of cancer patients to ICI treatment. A study published in**
**Cell 2023 titled "Dynamic CD8+ T cell responses to cancer immunotherapy in human**

**regional lymph nodes are disrupted in metastatic lymph nodes" also reported a high**
**presence of TCF1+PD1+Tpex in the uninvolved lymph nodes of head and neck**
**squamous cell carcinoma patients, and verified the clonal correlation between these**
**Tpex and Tex in the tumor microenvironment. This study already provided a**
**potential explanation for the anti-tumor mechanism of Tpex post-ICI treatment.**
**Additionally, in another Cancer Cell paper published in 2023 entitled "Progenitor-like**
**exhausted SPRY1+CD8+ T cells potentiate responsiveness to neoadjuvant PD-1**
**blockade in esophageal squamous cell carcinoma." revealed that the abundance of**
**SPRY1+ CD8+ Tpex in the digestive tract tumor microenvironment can serve as a**
**marker for patient treatment response and prognosis to immunotherapy. Given the**
**existing research discussing the presence and functions of Tpex in improving ICI**
**treatment, the novelty and translational value of this study are limited.**

We appreciate the reviewer's insightful comments. As the reviewer pointed out,
several studies have already demonstrated the association between Tpex in tumors and
LNs and the response to and prognosis of cancer patients undergoing ICI treatment (Refs.
17–21). Accordingly, recent studies suggest that T-cell responses to ICIs may originate
outside the tumor and depend on the recruitment of peripheral T cells (Refs. 14, 43-44, 47–
48), with LNs potentially serving as a reservoir for ICI-responsive, tumor-reactive T cells
(Refs. 25, 32, 34, 49–50). However, a closer examination of the literature revealed that the
first paper cited by the reviewer suggested that Tpex may differentiate into Tex within
metastasis-free LNs following ICI treatment and subsequently migrate to the tumor via the
bloodstream. Notably, this study did not include matched LN and tumor samples. The
second paper, which analyzed paired esophageal cancer tumor samples before and after
ICI treatment, demonstrated that a subset of exhausted CD8⁺ T cells expressing *SPRY1*
exhibits a progenitor exhausted T-cell (Tpex) phenotype and correlates with a complete
response to ICI. Yet, this study did not examine LNs. Given these limitations, few studies
have analyzed the dynamic changes in T cells after ICI treatment using matched human
samples from tumors, LNs, and blood.

Moreover, a key strength of our study is that, by developing methods to identify ICI-
bound T cells across various organs using flow cytometry and scRNA/TCR/CITE-seq, we
were able to investigate the dynamics of both ICI-bound and ICI-non-bound T cells. Indeed,
ICI-bound Tpex were detected in LNs, and this migration pathway may be supported by
evidence showing that ICI-bound Tpex are activated in LNs, migrate to tumors, and
differentiate into Tex within tumors. Importantly, no previous study has investigated T-cell
dynamics based on the distinction between ICI-bound and ICI-non-bound cells. We have
emphasized this point in the manuscript (page 13 lines 434-436). We believe that our study
provides novel insights into the crucial role of ICI-bound T cells in determining ICI treatment
efficacy.

**2) The authors performed multi-sample single-cell sequencing on two cancer**
**patients who underwent chemotherapy combined with anti-PD-1 treatment, and**
**multi-sample flow cytometry analysis on 12 gastrointestinal tumor patients who**
**received ICI treatment. However, a larger number of samples from more patients is**
**required to provide more convincing evidence.**

Appreciating the reviewer's suggestion, we additionally performed single-cell
RNA/TCR/CITE-seq with 12 samples from two ICI-naïve patients along with one new ICI-
treated patient to enhance the data in Fig. 5 and compare ICI-treated and naïve patients.
Thus, in total, we performed single-cell RNA/TCR/CITE-seq in two ICI-naïve cancer
patients and three ICI-treated cancer patients (Fig. 5, Supplementary Fig. 12,
Supplementary Fig. 14). The STARTRAC algorithm was used to compare clonal expansion
between ICI-treated and ICI-naïve samples. The results showed that the expansion score
increased after ICI treatment in C04, C05, C06.1, C12, and C17.0; the migration score

increased in C04, C05, C06.0, C06.1, C08, C14, and C17.0; and the transition score
decreased in C04, C06.0, C08, C12, C17.0, and C18 (Supplementary Fig. 14a). These
changes—higher expansion and migration scores and lower transition scores—were
primarily observed in IgG4-high clusters, with opposite trends rarely detected (page 11
lines 334-337). The C12 cluster showed increased expansion and transition scores after ICI
treatment, consistent with trends observed in other IgG4-high clusters. Regarding migration
potential, LN–tumor migration potential increased in C05, while LN–PBMC migration
potential increased and PBMC–tumor migration potential decreased in C08
(Supplementary Fig. 14b). Thus, STARTRAC analysis confirmed the enhanced expansion
and migration potential of ICI-bound T cells, particularly increased LN–tumor migration
potential in the LN-Tpex population (C05).

We also performed flow cytometry analysis by adding two additional patients after
ICI treatment in Fig. 4. The total number of ICI-treated patients was 14 (Supplementary
Table. 4), and the revised data are shown in Fig. 4e-h. The results demonstrated that the
proliferation activity of Tpex in LNs was significantly increased after ICI treatment.

We have added these results to the Results section ("STARTRAC analysis unveiled
the clonal link between ICI-bound CD8-Tpex in LNs and intratumoral ICI-bound CD8-
Tex"). We believe that these additional experimental results strengthen our research findings.

**3) The authors utilized CITE-seq, TCR sequencing, and TCR tracking algorithms**
**(STARTRAC) to analyze the clonal correlation between lymph node CD8+ Tpex and**
**tumor CD8+ Tex, and their potential migration pathways at the single-cell level.**
**However, showing the analysis of bioinformatics is not enough and indirect. This**
**study lacks dynamic monitoring of lymph node Tpex migration in animal models,**
**failing to clearly elucidate the migration and differentiation process of Tpex.**
**Therefore, the study does not provide strong evidence for the hypothesis that lymph**
**node Tpex migrate to the tumor and differentiate into Tex.**

Thank you for your thoughtful suggestion. We acknowledge that analyzing paired
human tissue samples alone is insufficient to fully capture the true kinetics of immune cells.
Therefore, we completely agree that additional mouse studies would be valuable. A
previous study (ref 32) demonstrated that precursor-like TCF1⁺CD8⁺ T cells migrate from
LNs to tumors in mouse models treated with FTY720, which blocks the migration of T cells
from LNs to tumors. Such models could indeed provide important insights. However, even
with these models, it remains challenging to examine the dynamics of ICI-bound T cells
from LNs to tumors in mouse models. Thus, at present, we have not established a suitable
mouse model for verifying ICI-bound T cells. Given the time constraints of the revision
period, we were unable to conduct such studies and have decided to address this issue in
future research. Nevertheless, we believe the findings of this study, which comprehensively
analyzes ICI-bound T cells in humans, provide significant insights. This limitation is now
explicitly stated in the revised manuscript's limitations section (page 14 lines 450-453). We
would be glad if the reviewer kindly accepted our viewpoint.

**4) The authors analyzed the relationship between the Tpex/Tex ratio in the lymph**
**nodes of 12 patients without ICI treatment and prognosis, finding that a high**
**Tpex/Tex ratio correlates with better prognosis. This finding suggests that Tpex**
**could serve as a prognostic predictor and a novel cell therapy approach, holding**
**significant clinical translational value. However, the authors need to conduct**
**validation in large external cohorts to enhance the reliability of this finding.**

Thank you for this important suggestion. The data presented in Fig. 3f represents a
survival analysis of 49 ICI-naive gastric cancer patients with metastasis-free LNs, not ICI-
treated patients. Additionally, we demonstrated the significance of Tpex cells in metastasis-
free LNs in 14 ICI-treated cancer patients but were unable to establish a relationship

between these cells and prognosis. As the reviewer noted, we fully recognize the need to
confirm the correlation between Tpex cells in LNs and prognosis in ICI-treated cancer
patients using larger cohorts. However, since preoperative ICI treatment is not yet a
standard approach for gastrointestinal cancers, obtaining surgical samples, including LNs,
from ICI-treated patients remains challenging in routine clinical practice. Consequently,
opportunities to investigate the prognostic impact of the Tpex/Tex ratio in LNs in ICI-treated
patients are currently limited. Instead, we conducted analyses of T cells in tumors and LNs
after ICI treatment using flow cytometry and scRNA/TCR/CITE sequencing in a limited
number of cases. In the future, as opportunities for such investigations become more
feasible, we intend to further explore the prognostic significance of the Tpex/Tex ratio in
LNs in ICI-treated patients. This limitation has been explicitly stated in the revised
manuscript's limitations section (page 14 lines 456-458).

**Responses to the reviewers' comments**

**Reviewer #1:**

***The authors have completed a thorough revisions and have provided a detailed***
***response to my comments. This has significantly improved the manuscript. There***
***are just a few minor points I would suggest adding:***

We thank the reviewer for constructive comments on our manuscript.

***1) T_{pex} → T_{ex} transition: The pseudotime and clonal data are of high quality, but***
***lack functional validation. Acknowledging this point as a limitation in the discussion***
***would be beneficial.***

Thank you for your insightful comment. We have now acknowledged the lack of
functional validation as a limitation in the revised manuscript (page 14, lines 470-471).

***2) TOX in CD4-T_{pex}: These cells display both TOX and progenitor markers such as***
***TCF7 and IL7R. It might be useful to add a line in the discussion addressing the***
***potential plasticity of these cells, considering their dual marker expression.***

Thank you for your valuable feedback. Initially, we reannotated “C09” cluster as
follicular helper T cells (T_{fh}) based on its predominant derivation from lymph nodes and its
expression of *CD4*, *TCF7*, *PDCD1*, *CXCL13*, *TOX*, *CXCR5*, and *BCL6*, the master
transcriptional regulator of T_{fh} cells, and we have revised the annotation accordingly (page
4, line 99). Furthermore, we provided an explanation regarding the co-expression of *TOX*
and *TCF7* observed in the C09 cluster, consistent with a previous study (PMID: 37322116)
reporting that *TOX* and *TCF7* are co-expressed in T_{fh} cells, and another study (PMID:
31844658) demonstrating that *TOX* functions as a key transcriptional regulator in T_{fh} cells
by promoting the expression of multiple molecules essential for T_{fh} biology, including
*TCF1*, *LEF1*, and *PDCD1* (page 6, lines 160-164).

**Reviewer #2:**

***The revised manuscript by Nose et al is not addressing most of the concern of the***
***reviewer. In fact, Supp Fig 1 shows how complicated the flow of this paper is. Fig 1***
***start with scRNAseq analysis of T cells in blood, LN and tumor of ICI treated EGJ***
***cancer and the author applied the NMF-derived PBMC and blood gene signatures on***
***ICI naive TCGA gastric cancer (why?) and say that these signatures does not predict***
***survival (can it be the problem of the signature?), then they show using flow***
***cytometry that the T_{pex}/T_{ex} ratio in LN (a value not related to the NMF signatures) is***
***associated with RFS and OS. Then, the authors show that they can identify anti-PD1-***
***bound T cells using anti-IgG4 (the isotype of the therapeutic anti-PD1) and suggest***
***that the anti-PD-1 bound T cells in the LN is more proliferative. Then, they went back***
***into their original scRNAseq analysis and added additional samples with scTCRseq***
***and CITEseq (1 ICI-treated and 2 ICI naive) and this time added the layer of***
***scTCRseq and CITEseq analysis. The finding was that anti-PD-1 bound CD8 T_{pex}***
***cells are clonally expanding and share the same clones as the T_{ex} in the tumor.***

We appreciate this important comment. We performed a Cox hazard analysis
including stage as a covariate, treating NMF8/NMF3 as T_{pex}/T_{ex} ratio. No significant
association was observed between NMF8/NMF3 and prognosis. We have added a new
figure (Fig. 2d) and revised the manuscript accordingly (page 6, line 178/ pages 6-7, lines
186-190).

Figure legend. Forest plot of the Cox hazards analysis for survival duration in the TCGA dataset of gastric adenocarcinoma evaluated with NMF analysis. The analysis incorporates NMFs and cancer stages as covariates.

1) Pai et al (ref 25) and others have reported the clonal linkage between Tpex in tumor draining LN and the intratumoral CD8 T cell. The novel result that the authors reported here is the analysis of the anti-PD-1 bound T cells. The authors show the anti-PD-1 bound T cells seem to be proliferating more in the LN and have mostly exhausted phenotype in the tumor. Thus, instead of first showing scRNAseq analysis of 2 ICI-treated patients, the authors should directly start with analysis of all 5 scRNAseq/scTCRseq/CITEseq samples (3 ICI treated and 2 ICI naive).

Thank you for this important suggestion. We agree that analyzing all five cases together is valuable for assessing the generalizability of our findings. To address this, we performed an integrated single-cell RNA-seq data analysis of all five cases and identified distinct clusters using UMAP (Supplementary Fig.13). Based on gene expression profiles, clusters C5.0 and C5.1 were defined as CD8-Tpex cells, while C7 was classified as T follicular helper (Tfh) cells. IgG4 expression was observed in CD8-Tpex [LN] (C5.0, C5.1), CD8-Tex [tumor/metastases] (C6 + C10), and CD4-Tfh [LN] (C7), and was restricted to ICI-treated patients. Notably, clonal expansion was particularly enriched in C5.0, C5.1, C6, and C10 (Supplementary Fig.13h). These findings are consistent with our initial analyses (Fig. 1, Fig. 5). In contrast, the spatial distribution of clusters across tumor, LN, and blood did not differ substantially between ICI-treated and ICI-naive groups (Supplementary Fig. 13f). These results suggest that the therapeutic effect of ICIs may depend more on the functional state of T cells rather than on changes in their frequency or anatomical distribution.

Currently, ICIs are not established as a standard neoadjuvant approach in gastrointestinal (GI) cancers. In this study, we obtained tissue samples from two rare cases of initially unresectable gastroesophageal cancer with distant metastases that became eligible for conversion surgery after preoperative ICIs (patients 1 and 2). We then included one resectable gastric cancer case treated with ICI as part of neoadjuvant chemotherapy (patient 3), as well as two ICI-naive gastric cancer cases (patients 4 and 5). Because the clinical context of ICI treatment was different between patients 1/2 (conversion surgery) and patient 3 (neoadjuvant therapy), we designated the two rare ICI-treated conversion cases as the **discovery cohort**, and subsequently included the additional ICI-treated and two ICI-naive cases as the **validation cohort**. Reanalyzing the data by combining all five cases would require regenerating Figures 1, 2, 5, S2, S3, S4, S5, S6, S12, S13, S14, S15, S16, and S17, which is currently not feasible due to limitations in available personnel resources. We kindly ask for the reviewer's understanding in this regard and hope this rationale is acceptable.

To accommodate the suggestion, we have added a new figure (Supplementary Fig.
 13) and revised the manuscript accordingly (page 4, line 86; page 9, lines 282-289).

**2) The authors should present the phenotype(s) of the anti-PD-1-bound T cells and**
 **how they are different from the similar T cells from ICI naive patients.**

Thank you for the important comment. We compared the transcriptional features of ICI-bound PD-1⁺ CD8⁺ T cells from ICI-treated cases (Patients 1-3) with those of ICI-non-bound PD-1⁺ CD8⁺ T cells from ICI-naive cases (Patients 4,5), across each tissue type. In tumors, genes involved in antigen recognition and T cell activation, such as *HLA-DRB1* and *HLA-DQA1*, were upregulated, whereas in lymph nodes, genes related to cell migration and inflammatory signaling, including *CCL3L1* and *CCL4L2*, were elevated. In contrast, no significant differences in exhaustion-related genes were observed in the blood, suggesting that the effects of ICIs are most pronounced in lymph nodes and tumors. Based on these findings, we incorporated additional explanations into the main text (page 10, lines 305-312) and Supplementary Fig. 15a.

Figure legend. The characteristics of ICI-bound PD-1⁺CD8⁺ T cells in ICI-treated cases and PD-1⁺CD8⁺ T cells in ICI-naive cases. Volcano plot showing the differences in gene expression between ICI-bound PD-1⁺CD8⁺ T cells in ICI-treated cases (Patients 1-3) and ICI-non-bound PD-1⁺CD8⁺ T cells in ICI-naive patients (Patients 4,5) per tissue. Genes related to sex differences were excluded.

**3) The authors should separate analyses on intratumoral T cells vs T cells in LN vs. T**
**cells in PBMC unless they can be integrated into unified phenotypes across organs.**

Thank you for your insightful comment. As shown in the newly added density plot in
the Supplementary Fig. 3b, there is tissue-specific bias in the distribution of T cells. T cell
populations from LN are present in tumor, Meta, and blood, indicating that T cells from all
tissue sources are generally well represented. However, circulating and tissue-resident T
cells generally exhibit distinct transcriptional programs. For example, tissue-derived T cells
specifically express genes such as *CD69*, *NR4A1*, and *CD2*, which are known to be
upregulated in tissue contexts (Supplementary Fig. 3c). Therefore, the separation of tissue-
resident and circulating T cells into different clusters reflects true biological differences
rather than a batch effect.

Moreover, we consider cross-tissue clustering to be useful for achieving consistent
annotation, and similar approaches have been employed in previously published studies
(PMID: 32611125, 36423636) to investigate how the same types of immune cells function
across different tissues, to better understand the tumor immune microenvironment, and to
evaluate transitions in cell states. Conversely, without applying consistent clustering across
tissues, many downstream analyses—such as STARTRAC—cannot be effectively
performed. To demonstrate the robustness of cross-tissue analysis to the reader, we have
added the figure showing the tissue-wise distribution of clusters (Supplementary Fig. 3b, c).

**4) The authors highlighted the patterns of T cell clone sharing in the anti-PD-1 bound**
**T cells vs non-bound T cells (in ICI treated patients) vs. T cells from ICI-naive**
**patients across the tumor, LN and PBMC, which is novel and important. Is there any**
**clonal sharing between TFH in LN with the CD4 in the tumor?**

Thank you for your important comment. In Fig. 6c, TCR sharing between CD4-Tfh
in LN (C09) and other clusters, including CD4-Tex in tumor (C06.1 and C17.2), appears
limited. When we created a figure for CD4-Tfh in LNs, we observed some degree of TCR
similarity with C6.1 (CD4-Tex in tumor) (Supplementary Fig. 20a). However, this similarity
varied among individual cases, and the extent of clonal expansion was lower than that
observed in CD8⁺ T cells (Fig. 5e). Therefore, TCR similarity between CD4-Tfh in LN with
the CD4⁺ T cells in the tumor appears to be relatively low compared with that of CD8⁺ T
cells. This explanation has been incorporated into the main text (page 12, line 377-379),
and the corresponding figure has been added (Supplementary Fig. 20).

**Figure legend. UMAP plot showing the clones in the tumor and meta compartments**
**that are shared with CD4-Tfh in LNs (C09).** The dots represent only the tumor and meta
compartments. Clusters not shown indicate zero cells with TCR sharing. The percentage of
TCR sharing is shown between cells in the C09 cluster and those in the C06.1 cluster.

**5) Any intratumoral anti-PD-1 bound T cell population identified in the ICI treated**
**gastric tumor/mets samples can be compared to those found in lung tumor/mets**
**sample from GSE173351 to see if there are any similarity. T cell populations from the**

**LN of gastric tumor patients should be compared with those from LN of the lung**
 **tumor patients in the GSE173351 dataset (instead of what is done now where the**
 **authors are mixing the clusters from blood, LN and tumor and compare them against**
 **a mixture of T cells from lung tumor, adjacent normal lung, and LN).**

Thank you for the thoughtful comment. In the original study (PMID: 34290408), T
 cells from tumor, adjacent normal lung tissue, TDLN, and a resected brain metastasis, were
 jointly clustered (Fig. 1.2). As the reviewer pointed out, we additionally performed separate
 analyses for (a) LNs and (b) tumors (note that the GSE173351 dataset does not contain
 blood samples, and thus comparisons involving blood were not possible). In the LN
 comparison, cluster C05 (CD8-Tpex) from our dataset corresponded mainly to CD8-effector
 cells, whereas cluster C09 (CD4-Tfh) aligned with CD4-Tfh cells, indicating that C05 and
 C09 in our dataset corresponded well to the LN-derived subsets in the GSE173351
 dataset. Additionally, in the tumor comparison, CD8-Tex clusters (C04, C06.0, C17.0,
 C17.1) and CD4-Tex clusters (C06.1, C17.2) from our dataset showed the following
 associations: C04 and C06.0 corresponded primarily to CD8-TRM cells, C06.1 matched
 CD4-Tfh(2) cells, C17.0 and C17.1 corresponded to CD8-proliferating and CD8-TRM cells,
 and C17.2 matched CD4-Treg cells. Thus, consistent similarities were observed in tissue-
 specific comparative analyses.

**(a) Lymph nodes (LNs)**

(b) Tumor

**Figure legend. Comparison of our single-cell RNA-seq data and GSE173351 data in**
**lymph nodes and tumor.** Uniform manifold approximation and projection (UMAP) plot of T
cells in single-cell RNA-seq colored by patients, clusters, and marker genes. Heatmap
comparing clusters per tissue.

**6) The flow cytometry analysis should be focused on that in Figure 4 where the**
**authors compare the effect of ICI in the T_{pex} and other T cell populations. The effect**
**of anti-PD-1 treatment in increasing the proliferation of TCF1⁺ PD1⁺ T_{pex} in LN. The**
**authors should demonstrate if the fraction of Ki67⁺ TCF1⁺ PD1⁺ T_{pex} in LN is**
**associated with survival in patients with gastric cancer. The importance of anti-PD-1**
**bound TFH in LN should also be supported with a similar analysis.**

Thank you for the insightful comment. In ICI-treated cases, although not statistically
significant, patients with higher Ki-67 expression in CD8-T_{pex} within metastasis-free LNs
tended to show better RFS. (There was no difference in the frequency of Ki-67 expression
in CD4-Tfh cells within metastasis-free LNs) These findings suggest that Ki-67 expression
after ICI treatment may be associated with subsequent prognosis. However, as this
analysis included only a small number of cases and various cancer types, future studies will
require a larger cohort with a uniform cancer type. We have added Supplementary Fig. 12
and described this point in the main text (page 9, lines 273-275).

**Figure legend. The association between Ki-67 expression levels in CD8-T_{pex} or CD4-**
**Tfh cells within metastasis-free lymph nodes of ICI-treated patients and clinical**
**outcomes.** Kaplan–Meier curves for recurrence-free survival rate (RFS) were compared
between high and low groups of the Ki-67 percentage in CD8-T_{pex} (a) or CD4-Tfh (b)

among metastasis-free lymph nodes (n=13). The two groups were divided based on the
median value. P-values were calculated by using the log-rank test.

**7) Finally, the Reviewer could not see any significant implication or conclusion from**
**the TCGA analysis.**

As shown in Supplementary Fig. 1, FACS analysis enabled us to assess immune
cell function and its association with prognosis. However, scRNA-seq analysis could not be
used for prognostic evaluation in either ICI-treated or ICI-naive cases, due to the lack of
available prognostic data. In ICI-naive cases, TCGA data allowed for prognostic analysis,
but because bulk RNA-seq data were used, NMF analysis was used in this study. This
approach demonstrated that intratumoral CD8-Tpex cells did not contribute to prognosis. In
the future, as scRNA-seq datasets of lymphocytes from tumors or LNs become publicly
available, more direct analyses of infiltrating lymphocytes and their relationship to
prognosis will be possible. We kindly ask for the reviewer's understanding in this regard.

**1. The single cell RNAseq analysis in Fig 1b shows a strong organ-driven batch**
**effect (not patient driven batch effect). The authors should really try to integrate the**
**data using data integration method like CCA, RPCA, bbknn etc and observe whether**
**such organ effect can be mitigated. Looking at the method description of the**
**scRNAseq analysis in line 579-580 of the main text, the reviewer noticed that the**
**authors did not integrate the profiles from different samples (different organs and**
**different patients). For an example of multi sample integration using scanpy, please**
**see [https://scanpy-tutorials.readthedocs.io/en/latest/integrating-data-using-](https://scanpy-tutorials.readthedocs.io/en/latest/integrating-data-using-ingest.html)**
**[https://nbisweden.github.io/workshop-archive/workshop-](https://nbisweden.github.io/workshop-archive/workshop-scRNAseq/2020-01-27/labs/compiled/scanpy/scanpy_03_integration.html)**
**[scRNAseq/2020-01-27/labs/compiled/scanpy/scanpy_03_integration.html](https://nbisweden.github.io/workshop-archive/workshop-scRNAseq/2020-01-27/labs/compiled/scanpy/scanpy_03_integration.html).**
**For instance, the integration may be able to cluster together the naive T cell**
**populations in the LN (C0 and C11) and PBMC (C2 and C10).**

Thank you for raising this important question. Our approach (Harmony) is a well-
established method for batch correction, as previously reported (PMID: 31948481). As the
reviewer suggested, we also evaluated other commonly used methods such as BBKNN,
MNN, and Scanorama, and all yielded broadly consistent results (Figures only for
reviewer). As mentioned in Supplementary Fig. 3b,c, although naive T cells from LNs and
PBMCs are both clearly classified as naive T cells, they are separated in the embedding
due to tissue-specific transcriptional programs, including the expression of genes such as
*CD69*, *NR4A1*, and *CD2*.

Figure Legend. UMAP plot of T cells in single-cell RNA-seq by each integration method (BBKNN, MNN, and Scanorama).

Currently, batch correction has been applied at the donor level; however, applying it
at the donor + site level would further remove tissue-specific differences (Figures only for
reviewer). In that case, LN (C0 and C11) and PBMC (C2 and C10) would be represented
as a single population. That said, there is no clear consensus on how strictly to separate
cell populations and cell states in the context of batch correction, and treating naive T cells

from PBMC and LNs as distinct clusters does not pose a problem for downstream
 analyses. We kindly ask for the reviewer's understanding, as updating the embedding and
 clustering to reflect these changes would require regenerating all related figures, which is
 not feasible given our current personnel capacity.

 **Figure legend. UMAP plot of T cells in single-cell RNA-seq at the donor + site level**
 **colored by Leiden cluster (left) and tissue of origin (right).**

**2. The scRNAseq analysis should be done on all 5 scRNAseq samples right from the**
 **start rather than analyzing two sample first (pt 1 and 2) and projecting the scRNAseq**
 **of Pt 3,4,5 on to the clustering of Pt 1 and 2 using symphony.**

Thank you for the thoughtful comment. We have posted our reply to the comment
 1).

 **3. If the authors still find separate CD4 and CD8 T cell populations after integration**
 **step, then the authors must ensure that analysis of different T cell populations are**
 **grouped by organ. For instance, to analyze the correlation between intratumoral T**
 **cell populations in TCGA gastric cancer dataset, the authors should only test gene**
 **signatures representing intratumoral T cell populations (e.g. C4, C6.0, C6.1, C7, C12,**
 **C14, C17.0, C17.1, C19 - based on Fig 1d). If the authors must do NMF analysis, then**
 **please only test NMF signatures that included intratumoral T cells (NMF0,3,5,6,7).**
 **NMF8 is clearly not strongly related to intratumoral T cells clusters (see Fig 1d and**
 **Fig 2b).**

Thank you for your perceptive comment. Previous studies (PMID: 28622514,
 38981439, 34450029) have not specifically separated CD8⁺ and CD4⁺ T cells, and we
 believe it is not necessary in our analysis either. In NMF analysis, it is important to project a
 wide range of data without making arbitrary selections, as irrelevant factors will simply have
 values close to zero. Therefore, we projected clusters from all tissues, not just tumor data.
 It is generally difficult to assign heterogeneous populations, except for clearly defined
 subsets such as Tregs, to a single NMF component. In particular, Tpex cells represent an
 intermediate state, and as the reviewer pointed out, it is indeed difficult to conclude that
 NMF1 and NMF8 are specific to the Tpex population. Therefore, we have added the
 limitation about this point in the discussion part (page 14, lines 461-465).

**4. (related to response to Reviewer 1 minor comment #5, Reviewer 2 comment #6 #7**
 **#8) The gene signatures of each NMF group need not be enriching for some**
 **reactome gene set (reactome is a pathway database with groups of proteins in a**
 **pathway, not a gene signature database with list of genes changing in expression in**
 **response to perturbation like the MsigDB). NMF0 clearly enriches for Treg markers**
 **(Fig 2b). NMF4 enriches naive or central memory T cell genes (Fig 2b). Looking at the**
 **genes in NMF1 and NMF8 (Fig 2b), there is no indication that they are related to CD8**

***Tpex. Please provide paper reference(s) showing that SLC2A3 (GLUT2), NR4A2,***
***GZMK, FGFBP2, CST7 and interferon signaling are in any way related to Tpex. Also,***
***are TCF7, SELL, TOX, PDCD1 included in the gene signatures of NMF1 and 8?***

We thank the reviewer for the important comment. Fig. 2a indicates that NMF8 and
NMF1 are associated with CD8-Tpex cells. As the reviewer noted, examination of the
weights of *TCF7*, *SELL*, *TOX*, and *PDCD1* expression in each NMF program
(Supplementary Table 2) shows that NMF2, which corresponds to CD4-Tfh cells,
successfully captured key Tfh features such as *TCF7* and *PDCD1*. However, marker genes
of CD8-Tpex cells were not well represented in NMF1 or NMF8. This may be explained by
the nature of NMF, which linearly decomposes gene expression data and is therefore
particularly well suited for identifying terminal and well-defined gene programs such as
those of Treg, Th1, or Th17 cells (PMID: 38359792). In contrast, Tpex cells possess an
intermediate phenotype and may exhibit behaviors that deviate from expectations in this
context. Accordingly, in the main text, we moderated the description specifically regarding
NMF1 (page 6, lines 180-182). The association between CD8-Tpex and GZMK has been
reported (PMID: 38510246). Furthermore, rather than CD8-Tpex cells intrinsically exhibiting
a strong interferon signaling signature, it is possible that they are influenced by other cell
populations included in NMF1 and NMF8.

Nevertheless, compared to pathway-based approaches such as MsigDB,
Reactome, or GO analyses—which often lack well-defined gene programs for T cell
subsets—NMF proved highly effective in extracting biologically meaningful transcriptional
programs in our TIL dataset, as previously demonstrated (PMID: 38359792). Given its
strengths in integrating across datasets and generating reusable gene programs, NMF
remains an indispensable tool for our analysis. While no single tool offers a perfect solution
at present, we hope this limitation is understood in the appropriate context. These points
have been added as a limitation (page 14, lines 461-465).

***5. (related to response to Reviewer 2 comment #4) In Suppl Fig 4, the authors test if***
***any of the T cell clusters in Fig 1b is enriched in the T cells in a neoadjuvant PD-1***
***inhibitor-treated lung cancer patients. The lung cancer patient-derived T cells are***
***coming from lung tumor/mets or adjacent normal lung or the LN (this is described in***
***the Figure caption). The authors are lumping all of them together into their existing***
***clustering, which is also a mixture of T cells from tumor, LN and PBMC of gastric***
***cancer patients. The authors then reported that higher proportion of T cells in C5***
***(Tpex in LN) and C8 (T eff in PBMC) is associated with major pathological response***
***(MPR) in the lung cancer patients receiving neoadjuvant anti-PD1. Are the T cells***
***projected to C5 from the lung cancer dataset from the LN T cells (there are 3 patients***
***with T cells from LN in the dataset)? Are the T cells projected to C8 from the lung***
***cancer dataset from the blood T cells (there is only 1 patient with T cells from***
***blood)? In their response, the authors stated that "there is currently no scRNA-seq***
***dataset that directly evaluates Tpex in LNs after ICI treatment in human samples and***
***examines its association with ICI treatment efficacy or prognosis." This does not***
***justify checking the Tpex cluster in LN across T cells from multiple organs. What the***
***authors can do is to test if any of the intratumoral T cell population in Fig 1b is***
***enriched in the intratumoral T cell population of lung cancer patients with MPR after***
***neoadjuvant anti-PD1.***

Thank you for your perceptive question. As described above (the question 5)), we
separated the tumor and LN samples in the *GSE173351* dataset and compared them with
our dataset. Accordingly, our data showed that LN-derived C05 (CD8-Tpex) and C09 (CD4-
Tfh) clusters corresponded well to the LN-derived CD8-effector and CD4-Tfh populations in
the GSE dataset, respectively. In response to the question of whether the T cells projected

to C8 in the lung cancer dataset originate from blood, we are unable to make this
 determination due to the absence of blood sample data in the *GSE173351* dataset. Finally,
 in the comparison across tumors, we observed that among the tumor-infiltrating exhausted
 T cell subsets—CD8-Tex (C04, C06.0, C17.0, C17.1) and CD4-Tex (C06.1, C17.2)—C04
 and C06.0 primarily corresponded to CD8-TRM cells, while C06.1 aligned with CD4-Tfh(2)
 cells. In addition, C17.0 and C17.1 matched CD8-proliferating and CD8-TRM cells, and
 C17.2 corresponded to CD4-Treg cells.

 Furthermore, to address the reviewers' comments, we extracted only the
 tumor/metastasis data from our reference dataset and performed reference mapping with
 Symphony followed by quantification with scCODA. As shown in the graph below, only
 clusters C04 and C05 exhibited statistically significant differences (FDR < 0.05), with higher
 proportions of these clusters being associated with favorable major pathological response
 (MPR). Taken together, both the analysis of all tissues and that restricted to
 tumor/metastasis samples suggest that the C05 cluster plays a role in mediating MPR.

 **Figure legend. Comparison of the proportion of each cluster between patients with**
 **the major pathologic response (MPR) and the non-major pathologic response (non-**
 **MPR).** The data from a previous study (*GSE173351*) were used for the analysis. The
 single-cell RNA-seq data of tumor/metastases were projected on our reference using
 Symphony. A cell population that is considered significantly (FDR < 0.05) different in the
 single-cell compositional data analysis (scCODA) framework is described as credible.

**6. (related to response to Reviewer 2 comment #3 and #5) The whole discussion of**
 **"CD4 T_{pex}" in this paper totally ignores the known fact that LNs enriches for CD4**
 **T_{fh} and that C09 matches the phenotype of T_{fh} very clearly (including the**
 **expression of TCF7, TOX, TOX2, PDCD1, TIGIT, CXCL13, CXCR5, BCL6 shown in**
 **Supp Fig 5) and that C09 is the ONLY cluster matching CD4 T_{fh} in this whole dataset.**
 **So C09 is T_{fh} and there is no need to invent a new CD4 T_{pex} population here. What**
 **is NOT known is whether aPD-1 binds to T_{fh} in the tumor draining LN (answer: yes**
 **based on Fig 5a), whether they are activated by aPD-1 (yes, their proliferation is**
 **induced in Fig 4g) and whether they migrate into the tumor (answer: probably no**
 **based on Fig 5d,e,g; more specific analysis is needed).**

 Thank you for your valuable feedback. We agree with the reviewer's opinion, and
 we annotated this cluster as follicular helper T cells (T_{fh}) (page 4, line 99). The C09
 population is bound by ICI (Fig. 5c, d), and increases Ki-67 expression with ICI treatment
 (Fig. 4g). However, STARTRAC analysis revealed that these cells exhibit low levels of
 migration and transition (Fig. 6a,b), and TCR sharing between CD4-T_{fh} in LN (C09) and
 other clusters appears limited (Fig. 6c). This explanation has been cooperated in the main
 text (page 11, lines 349-350).

**7. Analysis in Fig 3 is to support the importance of CD8 T_{pex} and T_{fh} fraction in**
 **proximal (presumably tumor draining) LNs. The authors used a ratio between T_{pex}**

**and Tex to correlate with relapse free or overall survival. Does the fraction of Tpex in**
 **the LN not correlated with RFS/OS? Also, the difference in p-value significance**
 **between CD8 Tpex/ Tex ratio in LN vs. CD8 Tpex/ Tex ratio in tumor may simply reflect**
 **the difference in the sample numbers (LN n=49, tumor n=23) - the KM plots look quite**
 **comparable between the two.**

 Thank you for the valuable comment. As shown below, the percentages of Tpex in
 CD8⁺ T cells and Tfh in CD4⁺ T cells were not significantly associated with either RFS or
 OS. These findings suggest that the balance between Tpex and Tex populations, rather
 than the proportion of Tpex alone, may more accurately reflect patient survival. As the
 reviewer pointed out, it is possible that a significant difference could emerge in tumors if the
 sample size were increased. Therefore, we have removed the statement in the manuscript
 indicating that “no difference was observed in the tumor or metastatic LNs (page 7, lines
 214-215).

**Figure legend. Kaplan-Meier curves for recurrence-free survival (RFS: left) and**
 **overall survival (OS: right) among ICI-naive gastric cancer patients.** The upper panel
 shows the comparison of the percentage of Tpex in CD8⁺ T cells from metastasis-free
 lymph nodes between the two groups. The lower panel shows the percentage of Tfh in
 CD4⁺ T cells from metastasis-free lymph nodes between the same two groups. The two
 groups were divided based on the median value. P-values were calculated by using the
 log-rank test.

**8. (related to response to Reviewer 2 comment #9) "First, The comparison between**
 **ICI-bound and ICI-non-bound CD8+ T cells revealed that ICI-bound CD8+ T cells**
 **exhibited higher CD103 expression and lower TCF1 expression than did ICI-non-**
 **bound CD8+ T cells across all tissues, suggesting that ICI-bound CD8+ T cells**
 **possess an exhausted phenotype, whereas ICI-non-bound CD8+ T cells retain a**

**naive phenotype (Supplementary Fig. 11a, b)." CD103 is not a known marker of**
**exhaustion - it is reported as a marker of tissue resident T cells that can also be**
**expressed on intratumoral T cells (see for example**
**[https://rupress.org/jem/article/218/4/e20201605/211911/Tissue-resident-memory-T-](https://rupress.org/jem/article/218/4/e20201605/211911/Tissue-resident-memory-T-cells-in-tumor-immunity)**
**cells-in-tumor-immunity). The authors should include citation of supporting papers**
**defining CD103 as marker of exhaustion, especially for T_{pex} in the LN?**

Thank you for the thoughtful comment. As correctly noted, CD103 is not regarded
as an exhaustion marker. Accordingly, we have revised the phrase "exhausted" phenotype"
to "memory" phenotype" in the manuscript (page 9, line 261).

**9. (Fig 4e-h) are the samples from ICI-treated T cells stained for PD-1 using the**
**indirect method (i.e. using anti-IgG4) while the ICI-naive stained using direct method**
**(i.e. using anti-PD1)? If yes, cells from the same T cell subsets from the same organ**
**in the ICI-treated and ICI-naive may not be comparable. This is because the**
**antibodies used to define the subsets are not exactly the same and hence may bind**
**the PD1⁺ T cells at different rates. Indeed, Supp Fig 10 shows that although the**
**correlation between PD-1 staining and IgG4 staining is highly positive, the slope of**
**the fitted linear curve is not equal to 1 (the %IgG4⁺ is smaller than %PD-1⁺). What**
**the authors can do is comparing the relative abundance of TCF1⁺ within PD1⁺ subset**
**(for ICI naive) or IgG4⁺ subset (for ICI-treated).**

Thank you for this important comment. In ICI-treated cases, detection was
performed using both anti-IgG4 and anti-PD-1 antibodies. The method employing the IgG4
antibody is considered equivalent to a secondary detection method using a biotinylated
antibody and streptavidin, and we believe that nearly all cells bound by therapeutic anti-
PD-1 antibodies are detected. In in vitro experiments using anti-PD-1 antibodies, PD-1
expression is unlikely to change markedly over a short period, suggesting that the anti-PD-
1 and anti-IgG4 antibodies detect nearly identical PD-1⁺ cell populations (Supplementary
Fig. 10a). The slope not reaching 1 is likely due to the absence of a strict cutoff for both
PD-1 and IgG4 signals in FACS analysis, which limits the accuracy of percentage
determination (Supplementary Fig. 10b).

Furthermore, ex vivo data have shown that detachment of therapeutic anti-PD-1
antibodies takes approximately 30–40 weeks (Osa, JCI Insight 2018). Therefore, in the
early treatment phase analyzed in this study, we consider that nearly all PD-1⁺ cells in any
tissue remain bound by therapeutic anti-PD-1 antibodies and are consequently detected by
the anti-IgG4 antibody as well (Fig. 4b). In addition, ex vivo scRNA-seq data demonstrated
that the expression levels of *PDCD1* mRNA and anti-IgG4 were well correlated, indicating
concordance between PD-1 expression and anti-IgG4 detection (Fig. 5b).

Taken together, these findings suggest that, in the early phase following anti-PD-1
antibody treatment in patients, nearly all PD-1⁺ cells across tissues are bound by anti-IgG4
antibodies. Although the inability to trace PD-1⁺ cells before and after treatment is a
limitation of this study, no such method currently exists. Considering the observed
concordance between *PDCD1* mRNA expression and anti-IgG4 detection, classification of
T_{pex} and T_{ex} cells based on IgG4–PD-1 expression appears reasonably valid. We
therefore hope the reviewer understands that, given this limitation, our current approach
represents the most feasible method for flow cytometric comparison between ICI-naive and
ICI-treated cases.

**10. The STARTRAC analysis in the main Figure 5 should highlight the differences**
**between the ICI-naive vs. ICI-treated patients (now Supp Fig 14). The authors should**
**highlight the subsets with differences. The important observations include ICI**

**increases expansion in T_{pex} (C5 and C12) and increased migration from LN to tumor**
**only in C5 (which may make sense since the T_{pex} in C12 is already in the tumor). It**
**is important to highlight that with ICI, the transition of intratumor Tex and T_{pex} are**
**lower -- does this mean they slow their transition to terminally exhausted**
**phenotype? These are important points to highlight. Finally, the STARTRAC analysis**
**should be performed only on the IgG4⁺ subset of the ICI-treated samples to show the**
**patterns of expansion, migration and transition in the anti-PD-1 bound T cells.**

**To aid the visualization of the TCR overlap between different pairs of clusters, the**
**author should visualize the percentage overlap in a heatmap format rather than**
**listing them in the text. This should be done in ICI treated vs ICI naive (and perhaps**
**in IgG4⁺ T cell subset).**

Thank you for this important comment. To better emphasize the comparison
between ICI-treated and ICI-naive cases in the STARTRAC analysis, we have restructured
the original Figure 5 into new Figures 5 and 6. As noted by the reviewer, the transition
scores of CD8-T_{pex} (C12) and CD8-T_{ex} subsets (C04, C06.0, C17.0, C17.2) within tumors
were lower in ICI-treated cases compared with ICI-naive cases. This may suggest that ICI
treatment does not promote the differentiation from T_{pex} to T_{ex}. We have added a
discussion on this point in the revised manuscript (page 11, lines 333-336).

Since each STARTRAC score presented in Supplementary Fig. 14b is calculated
independently for each cluster, extracting only the IgG4⁺ clusters would not alter the
results. For further details, please refer to the STARTRAC methodology (PMID: 30479382).
As the reviewer pointed out, STARTRAC analysis indeed shows that the IgG4⁺ clusters
exhibit higher scores than the IgG4⁻ clusters, indicating their importance. However, as
explained in the main text, we believe that performing STARTRAC analysis and
comparisons across all clusters, rather than focusing solely on IgG4⁺ clusters, is essential
to highlight the significance of IgG4-bound clusters after ICI treatment in this study.

Finally, in response to the reviewer's suggestion, we generated a heatmap of TCR
overlap across clusters, separately for ICI-treated and ICI-naive cases. As expected, this
analysis further demonstrates that, in ICI-treated cases compared with ICI-naive cases,
C05 showed high TCR similarity with C04, C06.0, C08, and C12. These findings indicate
that ICI treatment enhances TCR similarity between T_{pex} cells in LNs and T_{ex} cells in
tumors. We have added this as Supplementary Fig. 16 and described the corresponding
results in the main text (page 11, lines 356-357).

Figure legend. Heatmap illustrating the TCR overlap among clusters in ICI-treated
 and ICI-naive cases.

Reviewer #3

*After carefully evaluating the author's revised manuscript and supplementary data in*
 *response to the review comments, I find that the author has provided detailed and*
 *substantive responses to the key concerns, including research innovation, sample*
 *size, limitations in animal model validation and prognostic analysis. While certain*
 *constraints remain due to practical limitations, the author has made notable*
 *improvements to the paper, addressing several critical issues and enhancing its*
 *overall rigor. The authors has integrated matched tumor, lymph node, and blood*
 *samples from patients after ICI treatment and utilized scRNA/TCR/CITE-seq*
 *technologies to distinguish ICI-bound and non-bound T cells, demonstrating a*
 *certain level of technical innovation. Furthermore, the study focuses on the*
 *activation of T_{pex} cells in non-metastatic LNs, their migration into tumors, and their*
 *association with prognosis, providing unique insights into the dynamic migration*
 *mechanism of T_{pex} cells. In addition, the author has expanded the sample size and*
 *supplemented single-cell sequencing data, enhancing the reliability of the results.*
 *Although the study lacks dynamic monitoring of lymph node T_{pex} migration in*
 *animal models due to practical constraints, the author provides evidence of T_{pex}*
 *migration to some extent through multi-omics data and clonal correlation analysis,*
 *while explicitly acknowledging this limitation in the discussion, making the response*
 *scientifically sound and reasonable. Finally, validating the prognostic value of the*
 *T_{pex}/Tex ratio in a larger cohort is of significant reference value. However, the*
 *author indicates that the difficulty in obtaining surgical samples from ICI-treated*
 *patients limits further validation of the correlation between the T_{pex}/Tex ratio and*
 *prognosis in a larger cohort, which is indeed an objective limitation. In conclusion,*
 *this study systematically reveals the dynamic role of lymph node T_{pex} in ICI*
 *treatment, providing a new perspective for optimizing immunotherapy strategies. I*
 *recommend accepting the manuscript.*

We are deeply grateful to the reviewer for their constructive and insightful comments, which
 have greatly helped us to improve our manuscript.

Responses to the reviewers' comments

Reviewer #2

The revised manuscript of Nose et al has included some important updates. The manuscript keeps most of the original figures intact and most of the updates is added into new supplementary figures. The new manuscript feels heavy and a little convoluted to my preference but I appreciate the efforts taken to address my concerns.

1, First, the Authors tested several methods of integration and demonstrated that T cells from different organs are different transcriptomically from one another (Response to Reviewer 2 comment #1). However, I noted that the integration (i.e. batch effect removal) is based on removing the effect of different donors. Indeed, when the batch effect removal is performed based on donors+site, much of the site specific differences disappear. I agree that the Authors have freedom to choose to analyze the T cells without removing the organ-specific differences. However, if that is the Authors' choice, then when the Authors projected GSE173351 single cell data of lung cancer to their UMAP in Fig 1B, they have to check if the organ of origins of the mapped GSE173351 T cells should match the organ site of the Authors' own clusters. For example, GSE173351's "CD8-effectors (and also CD8 TRMs)" seem to match with lymph node-enriched C05 (CD8-Tpex). Are these CD8 T effector/TRM cells from the lymph node in GSE173351? If yes - it is useful to highlight this. If the CD8 T effector/TRM cells in GSE173351 are mostly intratumoral, the Authors should compare them with C12, which is the intratumoral Tpex in Figure 1B and 1E.

→We thank the reviewer for the constructive comments. As noted in our previous response (Response to Reviewer 2, comment #1), we explicitly evaluated the effect of batch correction using Harmony with donor and site as covariates. Even under this more stringent correction, tissue-specific differences among T cells were not eliminated but remained evident (**Figure for Reviewer #2**).

Importantly, these residual differences were not arbitrary or driven by technical distortion. Across integration strategies, clusters retained highly consistent marker gene expression profiles, and each cluster could be defined by essentially the same canonical markers as described in the main text. This indicates that the integration did not collapse biologically distinct T cell states nor artificially generate site-specific clusters.

Rather, tissue-specific effects manifested primarily as differences in the relative abundance of shared transcriptional states across organs, with substantial overlap in embedding space. Taken together, these analyses support that organ-associated transcriptional differences reflect genuine biological variation rather than incomplete batch correction. Consequently, our donor-based integration strategy, which yields highly similar clustering and marker patterns, provides a valid and biologically interpretable framework for downstream analyses.

Harmony corrected by Sample and Site

Figure for Reviewer #2

(Top) UMAP plot showing tissues after batch correction by donors and sites. (Bottom) UMAP plots showing the marker genes.

Furthermore, as described in our previous response to Reviewer 2 (comment #5), we performed comparative analyses between our dataset and the GSE173351 dataset separately for lymph nodes and tumor (Supplementary Fig. 5). In the LN comparison, cluster C05 (CD8-Tpex) in our dataset mainly corresponded to CD8-effector cells, whereas cluster C09 (CD4-Tfh) aligned with CD4-Tfh cells, indicating that C05 and C09 in our dataset closely matched the LN-derived subsets in the GSE173351 dataset (Supplementary Fig. 5a). Additionally, in the tumor comparison, CD8-TRM clusters (C04, C06.0, C17.0, C17.1) and CD4-TRM clusters (C06.1, C17.2) from our dataset showed the following associations: C04 and C06.0 primarily corresponded to CD8-TRM cells, C06.1 matched CD4-Tfh(2) cells, C17.0 and C17.1 corresponded to CD8-proliferating and CD8-TRM cells, and C17.2 matched CD4-Treg cells (Supplementary Fig. 5b). Thus, consistent similarities were observed in tissue-specific comparative analyses.

2, Second, the discussion of CD4 Tfh is redundant (page 6). They are enriched in LNs and have well known function as B cell helper and have well defined markers that are different from naive CD4 T or CD4 Tcm. The fact that very little CD4 clones in the tumor (or PBMC) have TCR overlap with these Tfh suggest that they are NOT migratory. The Authors should consider removing this part to avoid confusion.

→We thank the reviewer for this valuable comment. Indeed, as shown in our data, TCR similarity between CD4-Tfh cells in lymph nodes and CD4-TRM cells in tumors is not high (Figure 6c, Supplementary Fig. 21). These findings suggest that lymph node-resident Tfh cells are unlikely to migrate into the tumor microenvironment, but rather exert their functions primarily within the

lymph nodes. According to the reviewer's advice, we have revised and partly removed the related descriptions of CD4-Tfh cells (page 6, lines 168-172).

3, Third, following the same vein, the discussion on CD4 Tfh/Tex ratio in Fig 3g,h,i,j may not be relevant. The CD4 Tfh and CD4 Tex populations are not connected clonally and it is not clear why the ratio between them matters. Also, the log-rank p-values are all > 0.1. I also noted that Fig 3J still uses the "Tpex" label for the Tfh.

→Thank you for this important comment. As the reviewer pointed out, CD4-Tfh and CD4-Tex populations are not clonally connected, and no significant prognostic difference was observed based on the CD4-Tfh/Tex ratio. Accordingly, we have moderated our discussion on the CD4-Tfh/CD4-Tex ratio and instead emphasized the importance of CD8-Tpex/Tex populations (pages 7-8, lines 226-228, 231-232). In addition, as suggested, we have corrected the labeling in Figure 3J and its legend.